# ALT: Adaptive Low-Rank Transformation for Learngene-based Transformer Initialization

## Abstract

Parameter initialization plays a critical role when building and training diverse models under different scenarios. The recently proposed *Learngene* framework firstly learns one compact parameter set, termed **learngene**, from a large well-trained Ancestry model (Ans-Net), which is then inherited and transformed to initialize diverse descendant models (Des-Nets). One central goal in this framework is the pursuit of maximal inheritable efficiency, which entails learning a parameter set that is *dramatically more compact* than the Ans-Net. However, existing methods typically fall short in this part, thus limiting the portability of these parameters across diverse initialization scenarios. To diagnose this limitation, we rethink one state-of-the-art method LeTs, and reveal that its transformation matrices, which account for the majority of inherited parameters, are substantially overparameterized. Inspired by this insight, we introduce **ALT** (*Adaptive Low-rank Transformation*) to fundamentally improve inheritable-parameter efficiency. Specifically, we propose one novel SVD-inspired metric termed Gated Importance Score, building on which two distinct adaptation strategies, namely Flat Global Adaptation and Hierarchical Component Adaptation, dynamically refine the transformation matrices while preserving the initialization quality for Des-Nets. Comprehensive experiments across both vision and language domains demonstrate ALT's state-of-the-art inheritable-parameter efficiency and superior downstream performance.

## 1 Introduction

Transformer architectures have attracted substantial attention across multiple research fields, such as computer vision (Pan et al. (2023); Kirillov et al. (2023); Li et al. (2024)), natural language processing (Grattafiori et al. (2024); Liu et al. (2024); Guo et al. (2025)), and multimodal learning (Radford et al. (2021); Alayrac et al. (2022); Peng et al. (2025)). In practical scenarios, it is often necessary to build and train models of varying scales to meet diverse task requirements and resource constraints. Prior to model training, effective parameter initialization is essential for both training efficiency and final model performance (Glorot & Bengio (2010); He et al. (2015); Mishkin & Matas (2015); Dauphin & Schoenholz (2019); Zhu et al. (2021); Xu et al. (2024)). Nowadays, parameters obtained through pretraining on large-scale datasets provide an excellent initialization for downstream models, facilitating fine-tuning across a wide range of downstream tasks (He et al. (2020); Oquab et al. (2023); Team et al. (2024)). Nevertheless, such method typically involves transferring the entire pretrained parameters for initialization, which obviously neglects the diverse resource constraints of different downstream tasks. Although one could pretrain target models under specific constraints, such process requires repeated pretraining process and the access to pretrained datasets, which is inflexible, time-consuming and computationally expensive. *Therefore, how to flexibly initialize variable-sized models while considering the initialization efficiency?*

Recently, a novel learning framework termed *Learngene* (Wang et al. (2022; 2023b)) has been proposed, which firstly learns one compact parameter set, termed **learngene**, from a large well-trained ancestry model (Ans-Net). Then learngene is inherited and transformed to initialize variable-sized descendant models (Des-Net), after which they are fine-tuned on different downstream tasks, as shown in Fig.1(a). Grad-LG (Wang et al. (2022)) employs gradient-guided layer selection from the Ans-Net, and then stacks these layers with randomly-initialized ones to construct Des-Nets. Recent studies (Xia et al. (2024a;b); Shi et al. (2024); Xia et al. (2024c)) propose different parameter transformation strategies to construct one auxiliary model (Aux-Net) from learngene, which is trained through

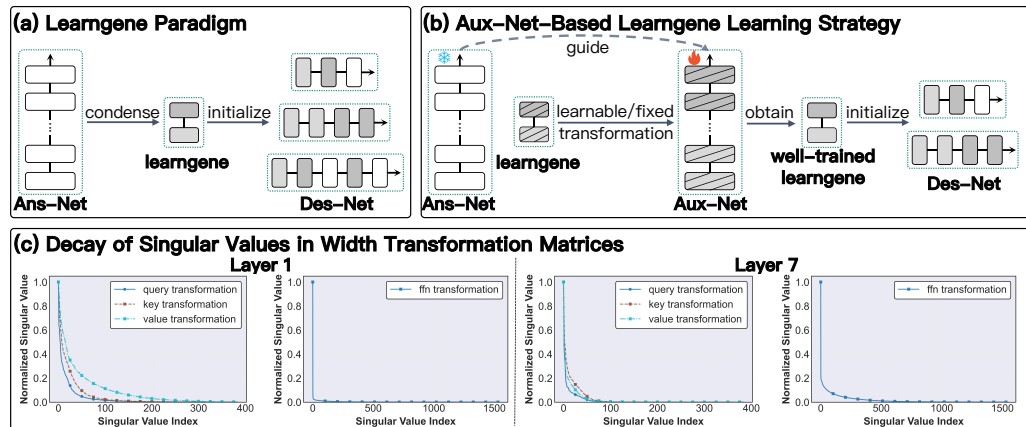

Figure 1: (a) Learngene paradigm. (b) Learngene learning strategy base on Aux-Net. (c) Singular values of well-trained width transformation matrices.

knowledge transfer from the Ans-Net. Then the well-trained learngene is inherited and transformed to initialize Des-Nets, as shown in Fig.1(b). For example, LeTs (Xia et al. (2024c)) transforms the learngene matrices along both width and depth dimension with learnable transformation matrices. In the Learngene framework, one central goal is the pursuit of maximal inheritable efficiency, which entails learning a parameter set that is *dramatically more compact* than the Ans-Net (further discussed in Sec.A.7). However, existing methods have paid insufficient attention to this, mostly achieving a modest 3–6× reduction in inherited parameters over the Ans-Net (see Fig.3(a)). Such inefficiency limits the inheritable-parameter portability across diverse Des-Net initialization scenarios, highlighting the need for methods that can substantially boost the inheritable-parameter efficiency.

In this paper, we rethink one state-of-the-art Learngene method LeTs (Xia et al. (2024c)) from the perspective of inheritable-parameter efficiency and observe that the majority of inherited parameters resides in the transformation matrices. For example, transformation matrices account for $\sim$ 38M parameters (over 70%) of the total 53M parameters learned and inherited from one 304M Ans-Net. This indicates that the transformation matrices are the primary bottleneck limiting the inheritable-parameter efficiency. Therefore, could the efficiency be improved by reducing the parameters of these transformation matrices? To answer this question, we utilize Singular Value Decomposition (SVD) (Klema & Laub (1980); Wall et al. (2003)) to analyze the low-rank properties of these well-trained transformation matrices, motivated by prior observations that weight matrices whose singular values decay rapidly could be well approximated by parameter-efficient low-rank matrices (Golub et al. (1987); Cai et al. (2010)). Fig.1(c) presents the singular values of well-trained width transformation matrices, yielding *two key insights*: (1) The singular values of these matrices decay rapidly among the leading terms, thereby enabling their effective approximation via low-rank parameterization with significantly reduced parameters; (2) The decay patterns vary not only across layers but also across different modules within the same layer. Please see more discussions in Sec.A.1.

Inspired by the above findings, we propose a novel Learngene approach termed **ALT** (*Adaptive Low-rank Transformation*), which substantially enhances inheritable-parameter efficiency while preserving initialization effectiveness for Des-Nets. Specifically, inspired by the rapid singular-value decay of these transformation matrices, we explicitly parameterize each matrix in the form of SVD as $U \Sigma V$, which naturally enables us to define the triplet of a learnable value ($\sigma_i$) and its associated vectors ($U_{:,i}, V_{i,:}$) as a cohesive learnable component, forming the basis for our subsequent adaptation. Next, exploiting the diverse decay patterns, we dynamically adapt the number of active components within each parameterized matrix. To guide this process, we first propose a novel, SVD-inspired importance metric termed Gated Importance Score (GIS), which formulates a component's importance as a gated combination of its magnitude (learnable value) and its orientation (learnable vectors). Building upon GIS, we then design and explore two distinct strategies for component adaptation, each motivated by a different principle: (1) Flat Global Adaptation (FGA) strategy, which follows a principle of pure global elitism by retaining only the globally top-scoring components; (2) Hierarchical Component Adaptation (HCA) strategy, which introduces a two-level decision hierarchy to first apportion the total number of active components among different matrices at a macro level, before performing

component selection at a micro level within each matrix. Overall, ALT proceeds in two stages. In Stage 1, we train one Aux-Net containing learngene and parameterized transformation matrices, during which our dynamic adaptation strategy progressively refines the transformation matrix. In Stage 2, we inherit both these matrices to initialize Des-Nets for downstream fine-tuning.

Through comprehensive experiments, we demonstrate the effectiveness of ALT: (1) Significant improvement in inheritable-parameter efficiency. For vision tasks, ALT achieves a $20\times$ reduction in inherited parameters over the 304M ViT-Large (Dosovitskiy et al. (2021)) Ans-Net. For language tasks, ALT achieves a $25\times$ reduction over the 1.5B GPT2-XLarge (Radford et al. (2019)) Ans-Net. (2) Superior performance across diverse downstream vision tasks. For instance, ALT outperforms standard pre-training and fine-tuning method by $2.0\%$ on ADE20K (Zhou et al. (2019)) with Des-H12-L12 (86M) as the backbone. (3) Training efficiency improvement on language tasks. For example, ALT achieves around $2\times$ reduction in training steps compared to training from scratch on GPT2-based Des-H16-L24 (354M). Our key **contributions** are as follows: (1) We are the first to pursue one critical yet underexplored goal in Learngene framework: achieving dramatic inheritable-parameter efficiency. (2) We introduce ALT, a novel Learngene approach featuring two key innovations: one new SVD-inspired metric termed Gated Importance Score, and two distinct adaptation strategies built upon it, namely Flat Global Adaptation and Hierarchical Component Adaptation. (3) Comprehensive experiments across both vision and language domains demonstrate ALT's state-of-the-art inheritable-parameter efficiency and superior downstream performance.

## 2 RELATED WORK

**Parameter initialization** is a critical step before model training, significantly influencing both training stability and the final model performance (Mishkin & Matas (2015); Dauphin & Schoenholz (2019); Zhu et al. (2021); Xu et al. (2024); Wu et al. (2024); Arpit et al. (2019)). Numerous initialization strategies have been proposed, such as the default methods provided by the timm library (Paszke et al. (2019)), Kaiming initialization (He et al. (2015)), and Xavier initialization (Glorot & Bengio (2010)). Currently, parameters acquired through pretraining on large-scale datasets offer excellent initialization for downstream models, which are then fine-tuned on various downstream tasks without altering the model architecture (He et al. (2020); Oquab et al. (2023); Team et al. (2024)). For downstream tasks subject to diverse resource constraints, one could pretrain variable-sized models under each specific constraint, which necessitates access to pretrained datasets and repeated pretraining process. Recently, there has been growing interest in leveraging the parameters of large pretrained models to initialize smaller models (Samragh et al. (2023); Xu et al. (2024); Zhmoginov et al. (2024); Wang et al. (2024b)). Additionally, Matformer (Devvrit et al. (2023)) introduces a universal training approach in which a single model is trained to support multiple sub-models. In contrast, we propose to learn one *highly-compact* and *reusable* parameter set that can flexibly initialize an entire family of models, thereby maximizing the portability and efficiency of the initialization parameters.

**Learngene** (Wang et al. (2022; 2023b); Xia et al. (2024b;a); Shi et al. (2024); Xia et al. (2024c); Wang et al. (2024c); Zu et al. (2025)) first learns one compact parameter set, termed learngene, from a large well-trained ancestry model (Ans-Net), and then transforms the learngene to initialize variable-sized descendant models (Des-Nets), which are subsequently fine-tuned, as depicted in Fig.1(a). Some prior works propose diverse parameter transformation strategies to convert the learngene into one auxiliary model (Aux-Net), which is trained under Ans-Net's guidance (Xia et al. (2024b;a); Shi et al. (2024); Xia et al. (2024c); Zu et al. (2025)). Once trained, the learngene and its associated transformation parameters are inherited to initialize Des-Nets, as shown in Fig.1(b). TLEG (Xia et al. (2024b)) introduces a linear expansion strategy while SWS (Xia et al. (2024a)) adopts a multi-stage weight sharing mechanism. LeTs (Xia et al. (2024c)) applies a series of learnable transformation matrices to convert learngene matrices. While these methods advance the transformation mechanism, we are the first to systematically address the core—yet previously underexplored—challenge of maximizing the inheritable-parameter efficiency. Building on the insight that the transformation matrices in LeTs are highly overparameterized, we introduce a novel approach to learn and dynamically adapt a significantly more compact, low-rank transformation structure.

**Low-rank decomposition** is a fundamental technique for diverse research fields such as model compression (Hsu et al. (2022); Li et al. (2023); Wang et al. (2024d)) and dimensionality reduction of word embeddings (Tanwar et al. (2018)). With the rapid development of pretrained models, low-rank

methods, particularly SVD, have emerged as a promising strategy for parameter-efficient fine-tuning (PEFT) (Hu et al. (2022); Zhang et al. (2023); Meng et al. (2024); Wang et al. (2024a); Gu et al. (2024)). While inspired by this line of work, our approach diverges fundamentally in both its core objective and technical design. Our goal is not to adapt a single large model to various tasks, but to learn and inherit one *reusable*, *highly-compact* parameter set for initializing a family of models. To this end, we propose one novel metric termed Gated Importance Score motivated by the multiplicative nature in SVD, and leverage it within two distinct adaptation strategies termed Flat Global Adaptation and Hierarchical Component Adaptation. Please see more discussions in Sec.A.9.

## 3 PROPOSED APPROACH

Our proposed ALT comprises two main parts: SVD-based parameterization and dynamic component adaptation for transformation matrices. For the former, we parameterize each transformation matrix in the form of Singular Value Decomposition (SVD). For the latter, we dynamically adapt the active learnable components within each transformation matrix during training. Next, we briefly review Learnable Transformation (LeTs) upon which our approach is built.

### 3.1 LEARNABLE TRANSFORMATION

For clarity, we retain the original notations from LeTs (Xia et al. (2024c)). Let the learngene module be denoted as $\Theta^{\mathrm{lg}} = [\boldsymbol{W}_1, \ldots, \boldsymbol{W}_L]^\top$, which consists of a sequence of $L$ learngene matrices, where each $\boldsymbol{W}_l$ is a learngene matrix. LeTs transforms learngene using a set of width transformation matrices and one depth transformation matrix $\boldsymbol{G}$.

**Width Transformation.** For each learngene matrix $\boldsymbol{W}_l$, LeTs introduces $\boldsymbol{F}_l^{\mathrm{in}}$ and $\boldsymbol{F}_l^{\mathrm{out}}$ to perform in-dimension and out-dimension transformation respectively. Specifically, the in-dimension trans-formed learngene $\boldsymbol{W}_l^{'}$ is given by $\mathrm{Concat}\big(\boldsymbol{W}_l, \boldsymbol{F}_l^{\mathrm{in}} \boldsymbol{W}_l\big)$, where $\mathrm{Concat}(\cdot)$ denotes the concatenation operation. Then the final width-transformed learngene $\boldsymbol{W}_l^{\mathrm{wt}}$ is given by $\mathrm{Concat}\big(\boldsymbol{W}_l', \boldsymbol{W}_l' \boldsymbol{F}_l^{\mathrm{out}\,\top}\big)$.

**Depth Transformation.** After width transformation, the set $\big\{\boldsymbol{W}_l^{\mathrm{wt}}\big\}_{l=1}^L$ is partitioned into groups, and the element of $\boldsymbol{G}$ is used to linearly combine the learngene matrices of each group into new ones. Since $\boldsymbol{G}$ accounts for only a negligible fraction of the total transformation parameters (on the order of $10^{-4}$), we focus on applying SVD-based parameterization to width transformation matrices.

### 3.2 SVD-BASED PARAMETERIZATION

Guided by the rapid singular-value decay observed in the width transformation matrices (Fig.1(c)), we explicitly parameterize each such matrix, denoted generically as $\boldsymbol{F}$, in the form of SVD:

$$\boldsymbol{F} = \boldsymbol{U}\,\boldsymbol{\Sigma}\,\boldsymbol{V}, \tag{1}$$

where the diagonal matrix $\boldsymbol{\Sigma} \in \mathbb{R}^{r \times r}$ holds the learnable values, $\boldsymbol{U} \in \mathbb{R}^{d^{\mathrm{new}} \times r}$ and $\boldsymbol{V} \in \mathbb{R}^{r \times d}$ contain the corresponding learnable vectors, with $r \ll \min(d^{\mathrm{new}}, d)$. This parameterization is applied to all $K$ width transformation matrices, indexed by $k = 1, \ldots, K$. The initial $r^{(k)}$ for each matrix $\boldsymbol{F}^{(k)}$ is the same. These SVD-parameterized matrices replace the original dense ones in the Aux-Net and are trained jointly with learngene matrices. To ensure such parameterization remains a valid approximation of SVD throughout training, we enforce the orthogonality of $\boldsymbol{U}^{(k)}$ and $\boldsymbol{V}^{(k)}$ by adding a regularization term to the training objective (see Sec.A.22.1 for details). While one could apply SVD for dense transformation matrices and truncate singular values after training, this process is computationally prohibitive for many high-dimensional matrices and fails to incorporate the SVD-based structure into the learning process. Our explicit parameterization, by contrast, enables these matrices to be learned jointly with the learngene matrices in a structurally aware manner.

### 3.3 DYNAMIC COMPONENT ADAPTATION

As shown in Fig.1(c), the singular-value decay patterns of these matrices vary by position and module type. Motivated by this observation, instead of using a static structure, we propose to *dynamically adapt* the internal structure of each transformation matrix during training. Our SVD-based parameterization naturally enables this adaptation by decomposing each matrix into a series

of distinct, learnable components. Specifically, within each $\boldsymbol{F}^{(k)}$, we partition its parameters into components $\{\boldsymbol{U}_{:,i}^{(k)}, \sigma_i^{(k)}, \boldsymbol{V}_{i,:}^{(k)}\}$, where $\boldsymbol{U}_{:,i}^{(k)}$ is the $i$-th column of $\boldsymbol{U}^{(k)}$, $\sigma_i^{(k)}$ is the $i$-th learnable diagonal entry of $\boldsymbol{\Sigma}^{(k)}$, and $\boldsymbol{V}_{i,:}^{(k)}$ is the $i$-th row of $\boldsymbol{V}^{(k)}$. To guide the adaptation process, we first introduce a novel metric to quantify the importance of each component.

**Gated Importance Score.** To measure the importance of each component, we propose the Gated Importance Score (GIS). This metric is motivated by the multiplicative nature of SVD, where the singular value acts as a scalar gateway for its corresponding singular vectors. Thus the GIS for the $i$-th component in the $k$-th matrix is defined as a multiplicative combination:

$$S_i^{(k)} = s(\sigma_i^{(k)}) \odot \left( w_U \cdot \bar{s}(\boldsymbol{U}_{:,i}^{(k)}) + w_V \cdot \bar{s}(\boldsymbol{V}_{i,:}^{(k)}) \right), \tag{2}$$

where $\bar{s}(\cdot)$ denotes the mean importance score over the elements of a vector, and $w_U, w_V$ are trade-offs. $s(\cdot)$ is a sensitivity-based importance function for a single parameter element, calculated as $s^{(t)} = \overline{I}^{(t)} \odot \overline{U}^{(t)}$, where $\overline{I}^{(t)}$ and $\overline{U}^{(t)}$ are the exponential moving averages of the sensitivity $I^{(t)}$ and its uncertainty estimation $U^{(t)}$ at training iteration $t$ (Zhang et al. (2022b)). Our GIS formulation ensures a component is deemed important only if both its magnitude, analogous to energy (learnable value), and its orientation, analogous to structural direction (learnable vectors), are significant.

**Dynamic Adaptation Strategy.** Building upon the GIS, we design and explore two distinct strategies for dynamically adapting components within each transformation matrix. Let $H^{(t)}$ be the total number of active components to keep across all transformation matrices at training iteration $t$. The loss function that drives the dynamic adaption process is the training objective of Aux-Net, detailed in Sec.A.22.1.

(1) Flat Global Adaptation (FGA). This strategy is motivated by a principle of pure global elitism. Specifically, it gathers the GIS values of components from all transformation matrices into a single global pool and retains only the top-$H^{(t)}$ components. Formally, we define a global binary mask vector $\boldsymbol{m} \in \{0,1\}^{N_{\text{total}}}$, where $N_{\text{total}} = \sum_k r^{(k)}$ is the total number of initial components. The mask for the $i$-th component of the $k$-th matrix is determined by $m_i^{(k,t)} = \mathbf{1}\left(S_i^{(k,t)} \geq \tau^{(t)}\right)$, where $\tau^{(t)}$ is the threshold value corresponding to the $H^{(t)}$-th highest score in the global pool, and $\mathbf{1}(\cdot)$ is the indicator function. The active components are those for which $m_i^{(k,t)} = 1$. This straightforward strategy ensures that only the globally most important components are retained.

(2) Hierarchical Component Adaptation (HCA). In contrast, HCA introduces a two-level decision hierarchy that decouples the adaptation process into a global, inter-matrix phase and a local, intra-matrix phase, motivated by the structured and modular nature of deep neural networks. Specifically, HCA adopts a two-step process:

*Step 1: Inter-Matrix Importance Apportionment.* At the macro level, HCA first assesses the overall importance of each transformation matrix $\boldsymbol{F}^{(k)}$ by computing an aggregated score, $\hat{S}^{(k,t)}$, from the GIS values of its constituent components by $\hat{S}^{(k,t)} = \text{Aggregate}\left(\{S_i^{(k,t)}\}_{i=1}^{r^{(k)}}\right)$, where the Aggregate function can be the mean or quantile. The total number of active components $H^{(t)}$ is then proportionally apportioned among all matrices to yield a target number of active components for each matrix: $H^{(k,t)} = \mathcal{A}\left(H^{(t)}, \left[\hat{S}^{(1,t)}, \ldots, \hat{S}^{(K,t)}\right], \left[r^{(1)}, \ldots, r^{(K)}\right]\right)$, where $\mathcal{A}$ denotes our capacity-aware iterative apportionment mechanism (detailed in Sec.A.3). This stage ensures that structurally critical matrices receive an adequate number of components.

*Step 2: Intra-Matrix Component Selection.* At the micro level, each matrix $\boldsymbol{F}^{(k)}$ conducts an internal selection based on its assigned $H^{(k,t)}$. A local binary mask $\boldsymbol{m}^{(k,t)}$ is computed as $m_i^{(k,t)} = \mathbf{1}\left(S_i^{(k,t)} \geq \tau^{(k,t)}\right)$, where $\tau^{(k,t)}$ is the threshold corresponding to the $H^{(k,t)}$-th highest score within the *local* set $\{S_i^{(k,t)}\}_{i=1}^{r^{(k)}}$. This step optimizes the internal structure of each matrix.

Here, the total number of active components $H^{(t)}$ is gradually decreased over iterations from an initial value $H^{(0)}$ to a final target $H^{(T)}$ using a commonly used cubically decreasing function (Sanh et al. (2020); Liang et al. (2023)) (detailed in Sec.A.4). By default, we adopt HCA in our main experiments. The pseudo code of our adaptation strategies is presented in Sec.A.2.

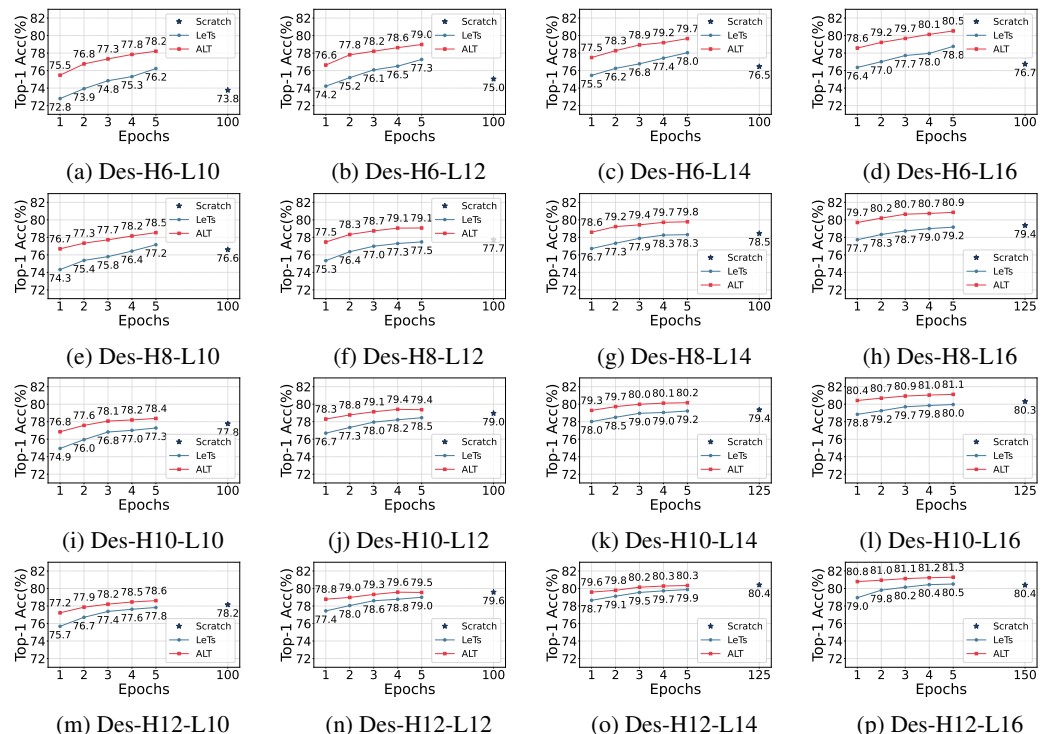

Figure 2: Performance comparison between Scratch, LeTs and ALT on ImageNet-1K classification.

### 3.4 Overall Pipeline of ALT

Building on the above two key parts, ALT proceeds in two stages:

**Stage 1: Training Inheritable Parameters.** We construct the Aux-Net by applying the SVD-parameterized transformation matrices to the learngene module. The Aux-Net is then trained with a joint objective that minimizes both the task loss, the output discrepancy between Ans-Net and Aux-Net, and one orthogonality regularization loss. During training, our dynamic component adaptation mechanism progressively refines the transformation matrices. Once trained, we obtain well-trained learngene and compact transformation parameters together as inheritable parameters.

**Stage 2: Model Initialization with Inheritable Parameters.** The well-trained transformation matrices are applied to the learngene module for initializing variable-sized Des-Nets. For initialization along width, we select continuous rows or columns from reconstructed $\boldsymbol{F}^{(k)}$ to satisfy each Des-Net's dimensional requirements. For initialization along depth, we select learngene groups in a predefined order and then pick the corresponding elements of the depth transformation matrix $\boldsymbol{G}$ to linearly combine learngene matrices within each group for initialization. Then the initialized Des-Net is fine-tuned on different downstream tasks. We present more details in Sec.A.8.

## 4 Experiments

### 4.1 Experimental Settings

**Vision experiments** are conducted on ImageNet-1K (Deng et al. (2009)), several downstream image classification datasets including CIFAR-10, CIFAR-100 (Krizhevsky et al. (2009)), Food-101 (Bossard et al. (2014)) and Cars-196 (Krause et al. (2013)), several semantic segmentation datasets including ADE20K (Zhou et al. (2019)), Pascal Context (Mottaghi et al. (2014)) and Cityscapes (Cordts et al. (2016)), and object detection dataset COCO2017 (Lin et al. (2014)). We report Top-1 classification accuracy (Top-1(%)) for classification tasks, mean Intersection over Union (mIoU(%)) for segmentation tasks following (Strudel et al. (2021)), and mean Average Precision (mAP(%)) for COCO2017 val following (Fang et al. (2021)). We report Params(M) as the

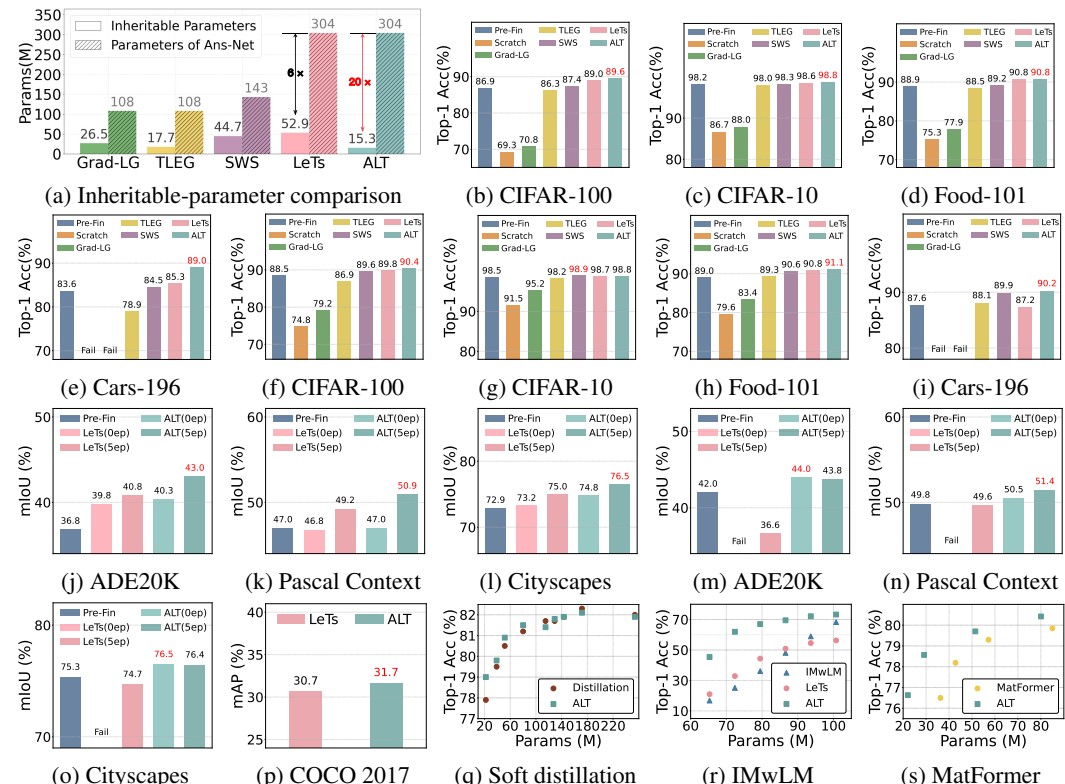

Figure 3: (a): Inheritable-parameter comparison. (b)–(i): Downstream image classification ((b)-(e):Des-H6-L12, (f)–(i):Des-H12-L12). (j)–(o): Semantic segmentation ((j)–(l):Des-H6-L12, (m)–(o):Des-H12-L12). (p): Object detection (Des-H6-L12). (q)–(s): Performance comparison with Soft distillation (Hinton et al. (2015)), IMwLM (Xu et al. (2024)) and MatFormer (Devvrit et al. (2023)). "Des-H6-L12" and "Des-H12-L12" means the backbone.

number of model parameters. **Language experiments** are conducted on English Wikipedia corpus following (Tan & Bansal (2020); Wang et al. (2023a)) using BERT (Devlin et al. (2019)), and on the OpenWebText dataset (Gao et al. (2020); Gokaslan et al. (2019)) following (Karpathy (2022)) using GPT2 (Radford et al. (2019)). We also use GLUE (Wang et al. (2018)) and SQuADv1.1 (Rajpurkar et al. (2016)) for downstream task evaluation. In Stage 2, we generally name the Des-Net with 12 heads and 12 layers as Des-H12-L12, and so on. Please see more details in Sec.A.22.

## 4.2 MAIN RESULTS

**Compared to previous Learngene methods, ALT significantly enhances the inheritable-parameter efficiency.** As shown in Fig.3a, for vision tasks, ALT inherits about 15M parameters, which achieves a **20×** reduction over the 304M ViT-Large Ans-Net. LeTs transfers substantially larger transformation parameters, while ALT significantly reduces this overhead. Moreover, the inheritable parameters of ALT grow only modestly with the transformation dimension increasing. The computational cost analysis of Aux-Net training are presented in Sec.A.6.

**Compared to training from scratch on ImageNet-1K, ALT performs better while reducing total training costs.** We evaluate ALT against: (1) Scratch that training models from scratch with default initialization from timm library (Paszke et al. (2019)) for 100 epochs (more epochs for larger models); (2) LeTs (Xia et al. (2024c)). As shown in Fig.2 and Tab.4, ALT consistently surpasses Scratch after 5 epoch tuning (in some cases after 1 epoch), while achieving around **2×** reduction in training costs for 19 models (∼ 947 GPU hours *vs.* ∼ 472 GPU hours). Notably, of ALT's 472 GPU hours, 445 GPU hours are devoted to training the inheritable parameters. Therefore, the efficiency of ALT becomes increasingly obvious as the number of Des-Nets grows, since we only need to train the inheritable parameters *once*. Compared to LeTs, ALT presents competitive performance while

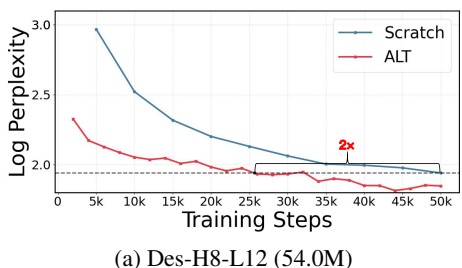 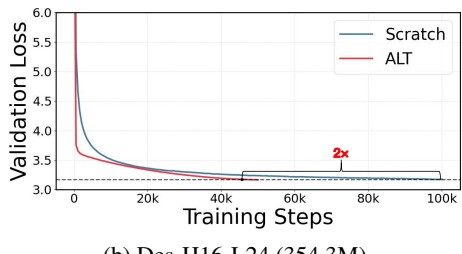

(a) Des-H8-L12 (54.0M)  (b) Des-H16-L24 (354.3M)

Figure 4: Comparison of ALT versus training from scratch. Left (a): Validation log perplexity for BERT on English Wikipedia corpus. Right (b): Validation loss for GPT-2 on OpenWebText.

Table 1: Comparison for BERT-based Des-Nets on downstream task GLUE and SQuADv1.1.

| Model | Method | SST-2 (Acc.) | MNLI (Acc.) | MRPC (F1/Acc.) | QNLI (Acc.) | QQP (Acc./F1) | STS-B (P./S. Corr.) | SQuADv1.1 (F1/EM) |
|---|---|---|---|---|---|---|---|---|
| Des-H8 -L12 | Pre-Fin | 88.6 | 79.0 | 88.7/83.8 | 85.8 | **89.8**/**86.2** | 82.1/81.9 | 80.0/70.6 |
| | ALT | **90.1** | **80.4** | **89.5**/**85.5** | **88.0** | **89.8**/86.0 | **86.1**/**85.7** | **80.9**/**71.2** |
| Des-H12 -L12 | Pre-Fin | 89.8 | 80.8 | 87.5/82.6 | 87.9 | 90.1/86.6 | 86.3/86.0 | **81.3**/71.1 |
| | ALT | **90.5** | **81.0** | **89.6**/**85.3** | **89.0** | **90.4**/**86.9** | **87.0**/**86.6** | **81.3**/**71.4** |

improving the inheritable-parameter efficiency. We conjecture that the superiority stems from the knowledge is mostly condensed in learngene matrices rather than in transformation ones. Compared to MatFormer (Devvrit et al. (2023)), ALT presents better performance (Fig.3(s)). Comparisons with soft distillation (Hinton et al. (2015)) are shown in Fig.3(q) and discussed in Sec.A.10. More comparisons with other initialization and training strategies are presented in Sec.A.5.

**ALT demonstrates better performance and efficiency across diverse downstream vision tasks.** Specifically, we compare ALT against: (1) Pre-Fin, which pre-trains on ImageNet-1K and fine-tunes on each downstream dataset; (2) Scratch; (3) Grad-LG (Wang et al. (2022)); (4) TLEG (Xia et al. (2024b)); (5) SWS (Xia et al. (2024a)); (6) LeTs (Xia et al. (2024c)). When evaluating ALT on downstream *image classification* datasets, as shown in Fig.3(b)-(i), ALT outperforms these baselines in most cases. Moreover, on Food-101, ALT surpasses Scratch after only 12 epochs within a 300-epoch schedule, which is about **25×** faster. We also report the results on downstream *semantic segmentation* and *object detection* tasks. For *semantic segmentation*, we follow the setting in (Strudel et al. (2021)), where we adopt Des-H6-L12 and Des-H12-L12 as the backbone and mask transformer as the decoder. "ALT(0ep)" and "ALT(5ep)" denote direct evaluation after initialization and evaluation after an additional 5-epoch training on ImageNet-1K, respectively. As shown in Fig.3(j)-(o), ALT(0ep) consistently outperforms Pre-Fin on all semantic segmentation tasks. For *object detection*, we employ YOLOS (Fang et al. (2021)) with Des-H6-L12 as the backbone and utilize learngene to initialize the backbone. From Fig.3(p), ALT outperforms LeTs on COCO2017 val, demonstrating its effectiveness in localization tasks. More details and discussions of Fig.3 are presented in Sec.A.22.3.

**Evaluation on ImageNet-1K without any tuning after initialization confirms ALT's superior initialization quality.** Specifically, we compare ALT against: (1) IMwLM (Xu et al. (2024)) which uses a 300-epoch pretrained model (129.1M parameters) for initialization; (2) LeTs (Xia et al. (2024c)). In Fig.3(r), ALT substantially outperforms all baselines. These results underscore ALT's ability to provide a superior initialization point for variable-sized models.

**ALT enhances training efficiency when pretraining BERT and GPT-2, and demonstrates competitive performance on downstream language tasks.** For BERT, we train and inherit about 39M parameters to initialize BERT-based Des-Nets. This corresponds to an $8\times$ reduction over the 335M Ans-Net (and **20×** when excluding embedding and prediction decoder parameters). As shown in Fig.4(a), ALT achieves around **2×** reduction in training steps compared to training from scratch. For GPT-2, we train and inherit about $62$ M parameters to initialize GPT2-based Des-Nets, achieving a $25\times$ reduction over the 1.5B Ans-Net (and **60×** when excluding embedding and head parameters). Fig.4(b) shows that ALT achieves around **2×** reduction in training steps compared to training from scratch. Besides, we conduct downstream evaluation on GLUE and SQuADv1.1, using the Des-H8-L12 (BERT-Small) and Des-H12-L12 (BERT-Base) configuration, and compare ALT against a standard Pretrain-Finetune (Pre-Fin) protocol. As shown in Tab.1, ALT consistently

Table 2: Performance of several variants on ImageNet-1K. "IS" means importance score function, "AS" means dynamic adaptation strategy and "EP" means the training epochs of Aux-Net.

| Method | IS | AS | EP | Top-1(Aux-Net) | Des-Net | ALT(Add/300) | ALT |
|---|---|---|---|---|---|---|---|
| ALT (Fixed) | - | - | 100 | 78.2 | Des-H6-L16 | 80.2 | **80.5** |
| ALT (Add) | Add | FGA | 100 | 78.7 | Des-H8-L16 | 80.5 | **80.9** |
| ALT (GIS) | GIS | FGA | 100 | 79.0 | Des-H10-L16 | 80.9 | **81.1** |
| ALT (HCA) | GIS | HCA | 100 | **79.1** | Des-H12-L16 | 81.1 | **81.3** |
| ALT | GIS | HCA | 300 | 81.9 | Des-H12-L24 | 81.2 | **81.5** |

Table 3: Evaluation on ImageNet-1K across varying parameter selection strategies, alternative learngene group configurations, and without initializing specific modules.

| Model | Selection | Top-1(%) | Learngene Group | Top-1(%) | Module | Top-1(%) |
|---|---|---|---|---|---|---|
| | continuous | 79.1 | 1,1,1,1,2,2,2,2,3,3,4,4 | **79.1** | MSA | 66.9 |
| Des-H8-L12 | $L_2$-norm $\uparrow$ | 79.1 | 1,1,1,2,2,2,3,3,3,4,4,4 | 78.7 | MLP | 50.3 |
| | $L_2$-norm $\downarrow$ | 78.8 | 1,1,2,2,3,3,3,3,4,4,4,4 | 77.2 | LN | 78.4 |

matches or outperforms Pre-Fin. The above results demonstrate that leveraging ALT for initialization provides a substantially better starting point, which not only accelerates the pre-training process but also boosts the final performance on downstream tasks. More details and discussions of Fig.4 and Tab.1 are presented in Sec.A.22.4 and A.22.5, respectively. We also present the discussion on applying ALT to Large Language Models (LLMs) in Sec.A.11.

All above vision and language experiments demonstrate that ALT-based initialization improves both the model performance and training efficiency across diverse Transformer architectures and sizes.

### 4.3 ABLATION AND ANALYSIS

**Dynamic Component Adaptation.** To investigate the effectiveness of our core contributions: dynamic adaptation process itself, GIS, FGA and HCA. We establish two key baselines for comparison: (1) ALT(Fixed), a static approach where each transformation matrix maintains a fixed number of components throughout training, and (2) ALT(Add), which replaces our GIS with the additive importance metric from AdaLoRA (Zhang et al. (2023)). Our analysis, with results in Tab.2, begins with the performance of Aux-Net. First, the performance gap between ALT(Add) and ALT(Fixed) underscores the importance of *dynamically* adapting components during training. Second, ALT(GIS) outperforms ALT(Add), validating the superiority of our proposed GIS over the additive metric. Furthermore, the strong performance of ALT(GIS) and ALT(HCA) demonstrates that both FGA and HCA effectively leverage the GIS for adaptation. When comparing the initialized Des-Nets, we observe that our ALT, benefiting from GIS and HCA, consistently outperforms the ALT(Add/300) baseline.

**Parameter Selection Strategies and Initialization Modules.** We further evaluate ALT using various parameter selection strategies following LeTs under 5 epoch tuning of Des-H8-L12. Specifically, we consider several magnitude-wise selection strategies including $L_2$-norm $\uparrow$, $L_2$-norm $\downarrow$, where $L_2$-norm $\uparrow$ means selecting rows/columns whose $L_2$-norm ranks top, and so on. As reported in Tab.3, we observe that these selection strategies achieve similar performance. We also compare different learngene group selection schemes. Besides, we ablate individual modules during ALT initialization: Multi-Head Self-Attention (MSA), Multi-Layer Perceptron (MLP), Layer Normalization (LN). Tab.3 reveals that omitting MLP causes severe performance degradation. More details and discussions of Tab.2 and Tab.3 are presented in Sec.A.22.6 and A.22.7.

## 5 CONCLUSION

In this paper, we pinpointed one critical yet underexplored goal in Learngene framework: achieving dramatic inheritable-parameter efficiency. We introduced ALT, a novel Learngene approach featuring two key innovations: one new SVD-inspired metric termed Gated Importance Score, and two distinct adaptation strategies termed Flat Global Adaptation and Hierarchical Component Adaptation. Comprehensive experiments across both vision and language domains demonstrated ALT's state-of-the-art inheritable-parameter efficiency and superior downstream performance.

**Ethics Statement.** The research presented in this paper strictly adheres to the ICLR Code of Ethics. Our work focuses on the fundamental challenge of parameter-efficient model initialization. The datasets used in our experiments, including ImageNet-1K, English Wikipedia corpus, OpenWebText, and *etc.*, are all publicly available benchmarks widely used by the research community. Our proposed method, ALT, is a general-purpose technique for improving computational and parameter efficiency. We do not foresee any direct negative societal impacts or ethical concerns arising from our methodology or its results.

**Reproducibility Statement.** We are committed to ensuring the reproducibility of our work. To this end, we provide the following resources and details:

- **Source Code:** The complete source code for ALT (including the implementations of GIS, FGA, and HCA) will be made publicly available upon acceptance of this paper.
- **Algorithmic Details:** The core algorithms for our adaptation strategies, Flat Global Adaptation (FGA) and Hierarchical Component Adaptation (HCA), are formally described with detailed pseudo code in Sec.A.2. Further implementation details, including our capacity-aware allocation mechanism, are provided in Sec.A.3 and Sec.A.22.
- **Experimental Configuration:** Important hyperparameters, including learning rates, batch sizes, training schedules, the specific configurations for our dynamic adaptation strategy, and *etc.*, are detailed in Sec.A.22.
- **Datasets and Models:** We use publicly available datasets (ImageNet-1K, OpenWebText, English Wikipedia, and *etc.*) and popular models (ViT, BERT, GPT-2) as the foundation for our experiments. Detailed references are provided in the experimental setup sections of our main paper and appendix.

We believe these resources provide a clear and comprehensive pathway for the research community to reproduce our results and build upon our work.

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

# A APPENDIX

In the appendix, we present more details about the proposed ALT for learngene-based transformer initialization in this paper, including:

- In Subsection A.1, we present more details and results about adopting SVD to transformation matrices.
- In Subsection A.2, we present pseudo code of our proposed adaptation strategies.
- In Subsection A.3, we present Capacity-Aware Iterative Apportionment for HCA.
- In Subsection A.4, we present the exact definition about the cubically decreasing function.
- In Subsection A.5, we present the comparison with other initialization and training strategies.
- In Subsection A.6, we present the computational cost analysis of Aux-Net training.
- In Subsection A.7, we present the discussion on the necessity of inheritable parameter reduction in the Learngene paradigm.
- In Subsection A.8, we present the strategy for Des-Net initialization.
- In Subsection A.9, we present more detailed comparison between ALT and low-rank methods.
- In Subsection A.10, we discuss the comparison between ALT and model compression.
- In Subsection A.11, we present the discussion on applying ALT to Large Language Models (LLMs).
- In Subsection A.12, we present the discussion on use of Large Language Models (LLMs).
- In Subsection A.13, we discuss the rationale and ablation for the initial Learngene structure.
- In Subsection A.14, we discuss the hyperparameter sensitivity analysis.
- In Subsection A.15, we discuss the detailed workflow of Learngene and the LeTs method.
- In Subsection A.16, we discuss the comparison between SVD-based parameterization and a Low-Rank product.
- In Subsection A.17, we discuss incorporating finer-grained knowledge transfer.
- In Subsection A.18, we discuss the break-even analysis of ALT's computational cost.
- In Subsection A.19, we discuss the analysis of training-time footprint.
- In Subsection A.20, we discuss the robustness under low-data scenarios.
- In Subsection A.21, we discuss the parameter counting conventions and storage footprint.
- In Subsection A.22, we provide more experimental details.
- In Subsection A.23, we provide the comparison between ALT, online models, meta learning and existing Learngene methods.
- In Subsection A.24, we discuss the limitations and future work of this paper.
- In Subsection A.25, we present the broader impacts of this paper.

## A.1 DETAILS AND MORE RESULTS ABOUT ADOPTING SVD TO TRANSFORMATION MATRICES

In this paper, we rethink one state-of-the-art Learngene method LeTs (Xia et al. (2024c)) from the perspective of inheritable-parameter efficiency and observe that the majority of inherited parameters resides in the width transformation matrices. Specifically, for LeTs, we configure the learngene module with 6 heads (head dimension is 64) and 8 layers, and transform it with *dense* width transformation matrices and depth transformation matrix to construct Aux-Net with 16 heads and 24 layers (Aux-H16-L24). We clarify that the dimensions of the width transformation matrices are not uniform; rather, they are determined by the specific architectural configuration of the module (*e.g.*, Attention versus FFN) where they are applied. To illustrate this adaptation, we provide the specific dimensions for the matrices within the first layer as a concrete example. For attention module, the

---

**Algorithm 1** Flat Global Adaptation (FGA) at training iteration $t$

---

**Input:**

1: Set of all GIS values from all $K$ matrices, $\mathcal{S}^{(t)} = \{S_i^{(k,t)}\}_{k=1..K, i=1..r^{(k)}}$.

2: Total number of active components to keep, $H^{(t)}$.

**Output:**

3: Global binary mask $\boldsymbol{m}^{(t)} \in \{0,1\}^{N_{\text{total}}}$, where $N_{\text{total}} = \sum_k r^{(k)}$.

4: **function** FGA($\mathcal{S}^{(t)}, H^{(t)}$)

5:      $\boldsymbol{\Phi}_{\text{global}} \leftarrow \text{Flatten}(\mathcal{S}^{(t)})$          ▷ Gather all scores into a single global pool

6:      $\tau^{(t)} \leftarrow \text{FindKthLargest}(\boldsymbol{\Phi}_{\text{global}}, H^{(t)})$          ▷ Find the threshold

7:      **for** $k \leftarrow 1$ **to** $K$ **do**

8:          **for** $i \leftarrow 1$ **to** $r^{(k)}$ **do**

9:              $m_i^{(k,t)} \leftarrow \mathbf{1}(S_i^{(k,t)} \geq \tau^{(t)})$

10:          **end for**

11:      **end for**

12:      **return** $\boldsymbol{m}^{(t)}$

13: **end function**

---

---

**Algorithm 2** Hierarchical Component Adaptation (HCA) at training iteration $t$

---

**Input:**

1: Set of all GIS values, $\mathcal{S}^{(t)} = \{S_i^{(k,t)}\}_{k=1..K, i=1..r^{(k)}}$.

2: Total number of active components to keep, $H^{(t)}$.

3: Vector of maximum component capacities, $\boldsymbol{c} = [r^{(1)}, \ldots, r^{(K)}]$.

**Output:**

4: Global binary mask $\boldsymbol{m}^{(t)} \in \{0,1\}^{N_{\text{total}}}$, where $N_{\text{total}} = \sum_k r^{(k)}$.

5: **function** HCA($\mathcal{S}^{(t)}, H^{(t)}, \boldsymbol{c}$)

6:      *// Stage 1: Inter-Matrix Importance Apportionment*

7:      Initialize aggregated score vector: $\hat{\boldsymbol{S}}^{(t)} \in \mathbb{R}^K$.

8:      **for** $k \leftarrow 1$ **to** $K$ **do**

9:          $\hat{S}^{(k,t)} \leftarrow \text{Aggregate}\left(\{S_i^{(k,t)}\}_{i=1}^{r^{(k)}}\right)$          ▷ e.g., mean or quantile

10:      **end for**

11:      $[H^{(1,t)}, \ldots, H^{(K,t)}] \leftarrow \mathcal{A}\left(H^{(t)}, \hat{\boldsymbol{S}}^{(t)}, \boldsymbol{c}\right)$          ▷ Capacity-Aware Iterative Apportionment

12:      *// Stage 2: Intra-Matrix Component Selection*

13:      **for** $k \leftarrow 1$ **to** $K$ **do**

14:          $\tau^{(k,t)} \leftarrow \text{FindKthLargest}\left(\{S_i^{(k,t)}\}_{i=1}^{r^{(k)}}, H^{(k,t)}\right)$          ▷ Find local threshold

15:          **for** $i \leftarrow 1$ **to** $r^{(k)}$ **do**

16:              $m_i^{(k,t)} \leftarrow \mathbf{1}\left(S_i^{(k,t)} \geq \tau^{(k,t)}\right)$

17:          **end for**

18:      **end for**

19:      **return** $\boldsymbol{m}^{(t)}$

20: **end function**

---

width transformation matrix has dimensions of $640 \times 384$. For ffn module, the width transformation matrix has dimensions of $2560 \times 1536$. Then we train the Aux-H16-L24 on ImageNet-1K (Deng et al. (2009)) for 300 epochs under the guidance of 304M ViT-Large (Dosovitskiy et al. (2021)) Ans-Net. After training, $\sim 38$M transformation matrices and $\sim 15$M learngene matrices are inherited, where transformation matrices account for over 70% of the total 53M inherited parameters. Since depth transformation matrix accounts for only a negligible fraction of the total transformation parameters (on the order of $10^{-4}$), we focus on analyzing width transformation matrices. The above results

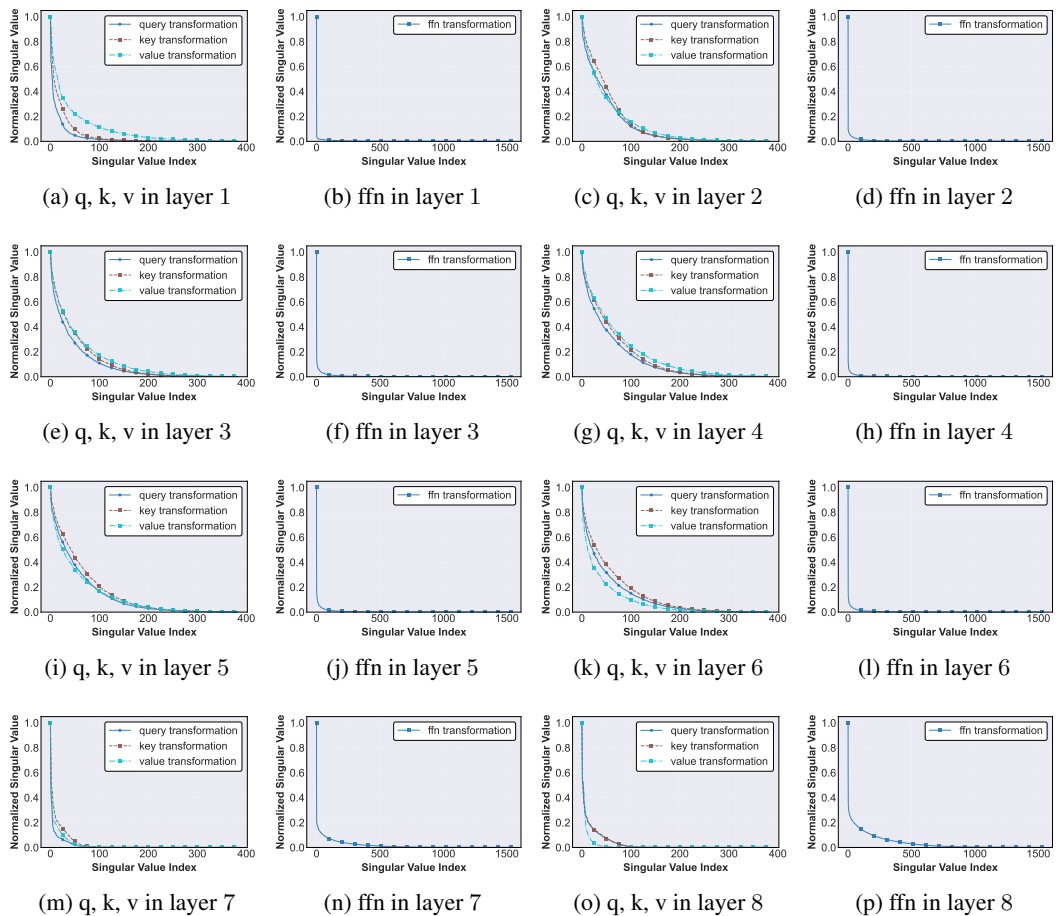

Figure 5: Singular values of well-trained width transformation matrices using LeTs Xia et al. (2024c). Here, "q, k, v in layer 1" means that the singular values of width transformation matrix for attention query, key and value projection in learngene 1, respectively. "ffn in layer 1" means that the singular values of width transformation matrix for FFN in learngene 1.

indicate that the width transformation matrices are the primary bottleneck limiting the inheritable-parameter efficiency. Therefore, could the efficiency be improved by reducing the parameters of width transformation matrices?

To answer this question, we utilize singular value decomposition (SVD) (Klema & Laub (1980); Wall et al. (2003)) to analyze the low-rank properties of these well-trained width transformation matrices, motivated by prior observations that weight matrices whose singular values decay rapidly could be well approximated by parameter-efficient low-rank matrices (Eckart & Young (1936); Golub et al. (1987); Cai et al. (2010)). Concretely, we perform SVD on well-trained width transformation matrices in Python via $U, S, V = np.linalg.svd(matrix)$, where $matrix$ is one well-trained transformation matrix, $S$ is the diagonal matrix containing singular values, $U$ and $V$ contain the corresponding left and right singular vectors. We then normalize the singular values following (Van den Berg et al. (2006)). Fig.5 and Fig.6 present the singular values and cumulative energy fraction of well-trained width transformation matrices, respectively, where "cumulative energy fraction" denotes the partial sum $\sum_{i=1}^{k} \sigma_i^2$ normalized by the total sum $\sum_{i=1}^{r} \sigma_i^2$, indicating how much of the matrix's spectral "energy" is captured by its top $k$ singular values. Fig.5 and Fig.6 yield *two key insights*:

- **Sharp initial decay.** The singular values of these matrices decay rapidly among the leading terms and the top singular values account for most energy, thereby enabling their effective approximation via low-rank parameterization with significantly reduced parameters. For

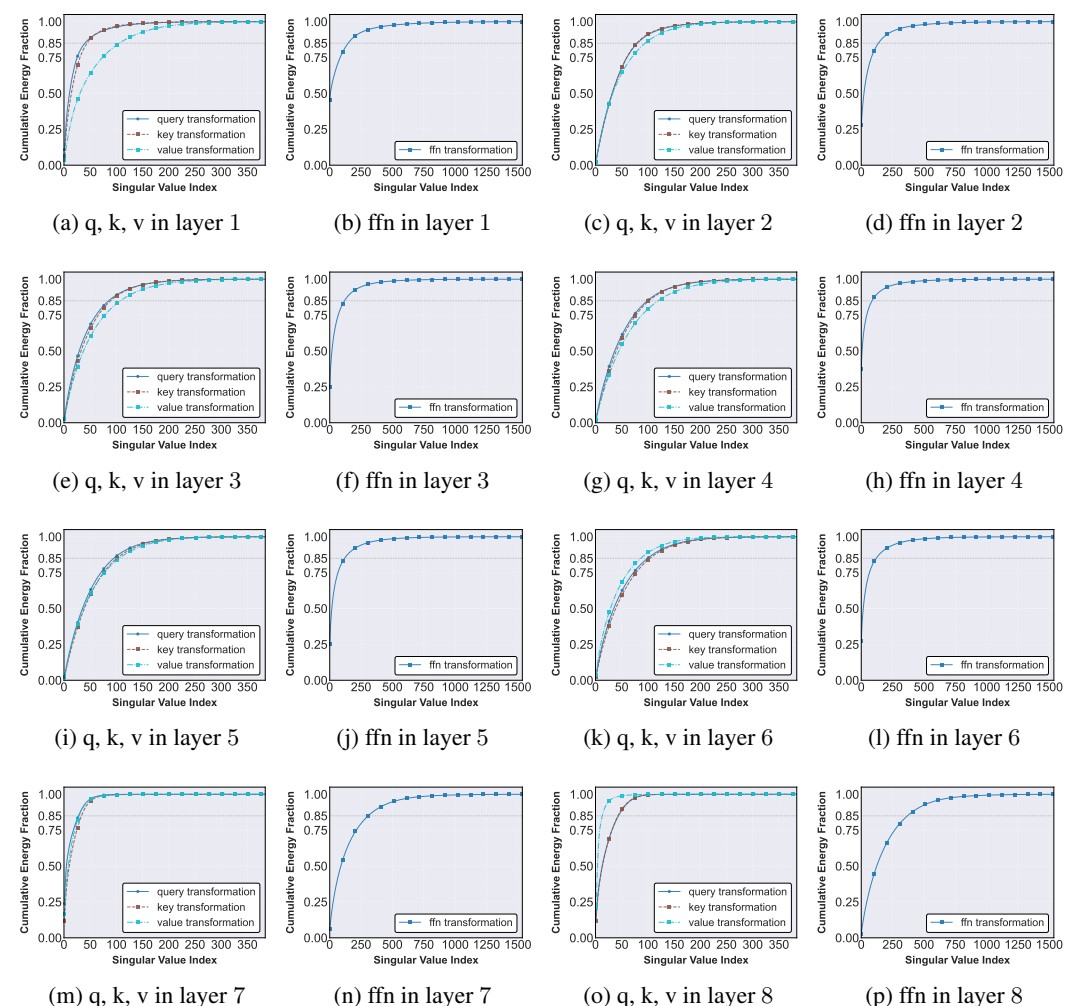

Figure 6: Cumulative energy fraction of well-trained width transformation matrices using LeTs Xia et al. (2024c). Here, "Cumulative energy fraction" denotes the partial sum $\sum_{i=1}^{k} \sigma_i^2$ normalized by the total sum $\sum_{i=1}^{r} \sigma_i^2$, indicating how much of the matrix's spectral "energy" is captured by its top $k$ singular values. "q, k, v in layer 1" means that the cumulative energy fraction of width transformation matrix for attention query, key and value projection in learngene 1. "ffn in layer 1" means that the cumulative energy fraction of width transformation matrix for FFN in learngene 1.

example, as shown in Fig.6j, in the width transformation matrix for FFN module of layer 5, the top 15% singular values account for more than 85% energy.

- **Module- and layer-dependent patterns.** The decay patterns vary not only across different layers but also across modules within the same layer. For example, the well-trained transformation matrix for the FFN module shows a different decay pattern compared to that for the attention query projection. Besides, the transformation matrices for FFN at layer 1 and layer 7 exhibit different decay behaviors.

Here, Fig.7 and Fig.8 show the number of retained learnable components within each transformation matrix. We observe that the transformation matrices for attention query, key and value projection and last layers retain more learnable components than those for others in BERT-based Aux-Net.

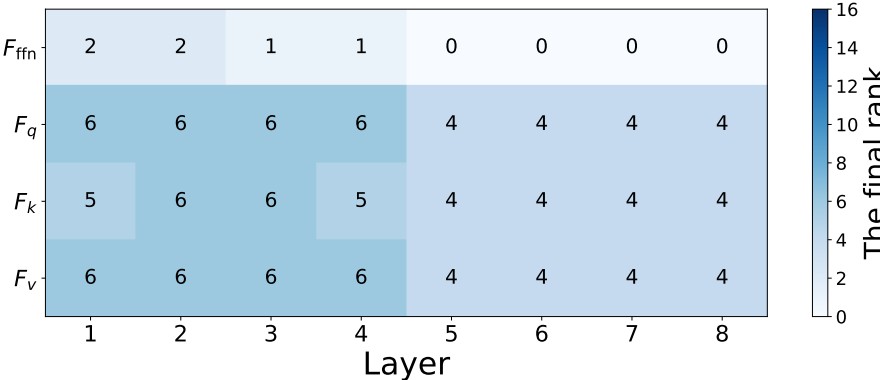

Figure 7: The number of retained learnable components of each parameterized transformation matrix when training ViT-based Aux-Net with ALT. The $x$-axis is the layer index and the $y$-axis represents different types of transformation matrices. "$F_{ffn}$", "$F_q$", "$F_k$" and "$F_v$" in the $y$-axis denote the transformation matrix for FFN, attention query, key and value projection, respectively.

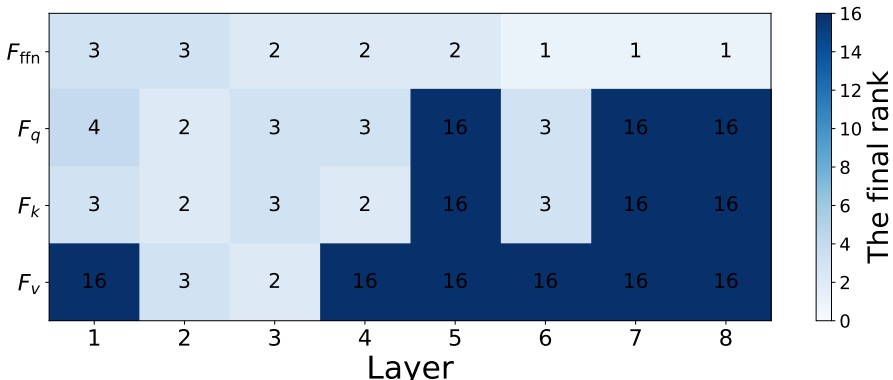

Figure 8: The number of retained learnable components of each parameterized transformation matrix when training BERT-based Aux-Net with ALT. The $x$-axis is the layer index and the $y$-axis represents different types of transformation matrices. "$F_{ffn}$", "$F_q$", "$F_k$" and "$F_v$" in the $y$-axis denote the transformation matrix for FFN, attention query, key and value projection, respectively.

## A.2 PSEUDO CODE OF OUR PROPOSED ADAPTATION STRATEGIES

To provide a concrete and formal description of our two dynamic adaptation strategies, we present their detailed procedures in this section. As discussed previously, while both strategies leverage the same Gated Importance Score (GIS) to quantify component importance, they differ fundamentally in their mechanism for applying the total component count, $H^{(t)}$.

Algorithm 1 details the **Flat Global Adaptation (FGA)** strategy. It operates on a principle of pure global elitism, where all components from all matrices compete in a single global pool. A single threshold, $\tau^{(t)}$, is computed based on the global top-$H^{(t)}$ scores, and this threshold is uniformly applied to adapt all components across the transformation matrices. The adaptation process is thus a monolithic competition where only the globally highest-scoring components survive, irrespective of their originating matrix. This strategy maximizes the retention of individually strong components but may risk sacrificing the functional diversity across the architecture.

Algorithm 2 outlines the **Hierarchical Component Adaptation (HCA)** strategy. This approach respects the model's modular structure by first executing an *Inter-Matrix Importance Apportionment* stage. It aggregates scores at the matrix level to determine a specific target number of components, $H^{(k,t)}$, for each matrix $\boldsymbol{F}^{(k)}$. Subsequently, in the *Intra-Matrix Component Selection* stage, a unique local threshold, $\tau^{(k,t)}$, is computed for each matrix to perform the final selection. This two-stage

Table 4: The numerical results of Fig.2 in our original paper. The number of epochs is indicated in brackets within the "Scratch" column, with the default being 100 epochs when no brackets are present. FLOPs(G) is the indicator of model complexity.

| Model | Params(M) | FLOPs(G) | Scratch | LeTs 5 ep | ALT 1 ep | ALT 5 ep |
|-------|-----------|----------|---------|-----------|----------|----------|
| Des-H6-L10 | 18.5 | 3.8 | 73.8 | 76.2 | 75.5 | **78.2** |
| Des-H6-L12 | 22.1 | 4.6 | 75.0 | 77.3 | 76.7 | **79.0** |
| Des-H6-L14 | 25.6 | 5.3 | 76.5 | 78.0 | 77.5 | **79.7** |
| Des-H6-L16 | 29.2 | 6.1 | 76.7 | 78.8 | 78.6 | **80.5** |
| Des-H8-L10 | 32.5 | 6.6 | 76.6 | 77.2 | 76.7 | **78.5** |
| Des-H8-L12 | 38.8 | 8.0 | 77.7 | 77.5 | 77.5 | **79.1** |
| Des-H8-L14 | 45.1 | 9.3 | 78.5 | 78.3 | 78.6 | **79.8** |
| Des-H8-L16 | 51.5 | 10.6 | 79.4 (125ep) | 79.2 | 79.7 | **80.9** |
| Des-H10-L10 | 50.5 | 10.2 | 77.8 | 77.3 | 76.9 | **78.4** |
| Des-H10-L12 | 60.3 | 12.3 | 79.0 | 78.5 | 78.3 | **79.4** |
| Des-H10-L14 | 70.2 | 14.3 | 79.4 (125ep) | 79.2 | 79.3 | **80.2** |
| Des-H10-L16 | 80.0 | 16.3 | 80.3 (125ep) | 80.0 | 80.4 | **81.1** |
| Des-H12-L10 | 72.4 | 14.6 | 78.2 | 77.8 | 77.2 | **78.6** |
| Des-H12-L12 | 86.6 | 17.5 | 79.6 | 79.0 | 78.8 | **79.6** |
| Des-H12-L14 | 100.7 | 20.4 | 80.4 (125ep) | 79.9 | 79.6 | **80.3** |
| Des-H12-L16 | 114.9 | 23.3 | 80.4 (150ep) | 80.5 | 80.8 | **81.3** |
| Des-H12-L18 | 129.1 | 26.2 | 79.1 (300ep) | - | 81.2 | - |
| Des-H12-L24 | 171.6 | 34.9 | 79.5 (200ep) | - | 81.5 | - |
| Des-H16-L24 | 304.3 | 61.3 | 78.3 (200ep) | - | 81.6 | - |

Table 5: Initialization performance comparison on ImageNet-1K. We report Top-1 accuracy (%). "Direct Evaluation (0 epoch)" denotes direct evaluation after initialization without any fine-tuning, while "Fine-tuning (1 epoch)" represents performance after one epoch of fine-tuning.

| Model | Direct Evaluation (0 epoch) | | | Fine-tuning (1 epoch) | | | |
|-------|-----------|--------|------------|-----------|-----------|-----------|-----------|
| | Params(M) | IMwLM | ALT (ours) | Params(M) | Matformer | Params(M) | ALT(ours) |
| Des-H12-L9 | 65.3 | 16.8 | **45.4** | 36.3 | 76.5 | 22.1 | **76.6** |
| Des-H12-L10 | 72.4 | 25.1 | **62.0** | 42.8 | 78.2 | 29.2 | **78.6** |
| Des-H12-L11 | 79.5 | 36.1 | **67.0** | 57.2 | 79.3 | 51.5 | **79.7** |
| Des-H12-L12 | 86.6 | 48.1 | **69.5** | 85.0 | 79.9 | 80.0 | **80.4** |

process ensures a structured and balanced adaptation. HCA thus prioritizes the overall health and functional completeness of the entire matrix over the singular prowess of individual components.

## A.3 Capacity-Aware Iterative Apportionment for HCA

A critical implementation detail of our Hierarchical Component Adaptation (HCA) strategy lies in the *Inter-Matrix Importance Apportionment* stage. A naive proportional distribution of the total active components, $H^{(t)}$, could assign a target number of components, $H^{(k,t)}$, to a transformation matrix $\boldsymbol{F}^{(k)}$ that exceeds its intrinsic maximum rank, $r^{(k)}$. Such an over-allocation would lead to an inefficient use of the adaptation capacity, as the surplus components cannot be realized.

To address this challenge and ensure a lossless distribution, we have designed and implemented a **Capacity-Aware Iterative Apportionment** mechanism. This algorithm guarantees that the entire component count $H^{(t)}$ is distributed efficiently by intelligently reallocating any "overflow" capacity from saturated matrices to those that can still accommodate additional components.

The mechanism unfolds in iterative rounds.

- **Initial Round:** The process begins by proportionally distributing the total component count $H^{(t)}$ among all transformation matrices based on their aggregated Gated Importance Scores. We then identify any matrix for which its assigned component count meets or exceeds its

Table 6: Comparison with other initialization or training strategies on ImageNet-1K. We report Top-1 accuracy (%).

| Model | Params(M) | SAM-5 | LookSAM-5 | SAM-10 | LookSAM-10 | Look-LayerSAM | ALT(ours) |
|---|---|---|---|---|---|---|---|
| Des-H6-L12 | 22.1 | 75.5 | 77.6 | 74.9 | 77.1 | - | **79.0** |
| Des-H12-L12 | 86.6 | 75.7 | **79.8** | 75.1 | 78.7 | 79.7 | **79.6** |

Table 7: Comparison with Soft distillation (Hinton et al. (2015)) on ImageNet-1K. We report Top-1 accuracy (%) of 100 epoch soft distillation and the corresponding training time (GPU hours) for Soft distillation. We also report Top-1 accuracy (%) of 5-15 epoch finetuning and the corresponding training time (GPU hours) for ALT. Our ALT achieves comparable or better performance with significantly less training time.

| Model | Params(M) | Soft distillation | | ALT(ours) | |
|---|---|---|---|---|---|
| | | Training Hours(h) | Top-1(%) | Training Hours(h) | Top-1(%) |
| Des-H6-L12 | 22.1 | 37.2 | 77.9 | **1.2** | **79.0** |
| Des-H8-L12 | 38.8 | 41.0 | 79.5 | **3.4** | **79.8** |
| Des-H8-L16 | 51.5 | 49.8 | 80.5 | **1.4** | **80.9** |
| Des-H10-L16 | 80.0 | 55.8 | 81.2 | **5.8** | **81.5** |
| Des-H12-L16 | 114.9 | 61.6 | **81.7** | **4.5** | 81.4 |
| Des-H12-L18 | 129.1 | 66.8 | 81.7 | **7.5** | **81.8** |
| Des-H12-L20 | 143.3 | 70.5 | **81.9** | **5.5** | 81.9 |
| Des-H12-L24 | 171.6 | 79.7 | **82.3** | **6.3** | 82.1 |
| Des-H16-L20 | 253.9 | 86.8 | **82.0** | **6.5** | 81.9 |

maximum rank $r^{(k)}$. Such a matrix is deemed "saturated," and its allocation is capped at its capacity, $r^{(k)}$.

- **Subsequent Rounds:** The "overflow" component count—defined as the sum of excesses from all newly saturated matrices—constitutes the budget for the next round. This remaining count is then proportionally re-apportioned, but critically, only among the subset of matrices that are not yet saturated.

- **Termination:** This iterative process of apportionment, saturation checking, and redistribution continues until either the entire component count $H^{(t)}$ has been assigned, or all matrices have reached their maximum capacity.

This iterative procedure provides a dual guarantee: first, that no component is wasted on a matrix that cannot physically accommodate it, and second, that the full adaptation capacity is utilized by distributing it among the matrices that can most benefit from additional structural complexity.

### A.4 EXACT DEFINITION ABOUT THE CUBICALLY DECREASING FUNCTION

As mentioned in Sec.3.3 of our original paper, we adopt a commonly used cubically decreasing function (Sanh et al. (2020); Zhang et al. (2023); Zhu & Gupta (2017); Zafrir et al. (2021); Liang et al. (2023)) to ensure that the number of active components is gradually decreasing. The detailed function is defined as follows:

$$
H^{(t)} = \begin{cases} H^{(0)}, & 0 \leq t < T_s, \\ H^{(T)} + \left(H^{(0)} - H^{(T)}\right)\left(1 - \dfrac{t - T_s}{T_f - T_s}\right)^3, & T_s \leq t \leq T_f, \\ H^{(T)}, & T_f < t < T. \end{cases} \tag{3}
$$

where $H^{(0)}$ and $H^{(T)}$ is the initial and final total number of components across all width transformation matrices, $T$ is the number of total training iterations and $0 < T_s < T_f < T$ are hyperparameters. Here, we set $T_s$ as $0.1T$ and $T_f$ as $0.9T$.

Table 8: Comparison under linear probing protocol on CIFAR-100, CIFAR-10 and Food-101. We report Top-1 accuracy (%).

| Method | CIFAR-100 | CIFAR-10 | Food-101 |
|--------|-----------|----------|----------|
| LeTs   | 50.0      | 72.3     | 61.1     |
| ALT    | **56.4**  | **78.3** | **68.5** |

## A.5 Comparison with Other Initialization and Training Strategies

This section provides a broader comparative analysis of ALT against other paradigms for improving model performance, specifically addressing the trade-offs between initialization costs and downstream training regimens.

**Comparison with Resource-Intensive Initialization Strategies.** A key advantage of our ALT framework is its "*learn once, apply many times*" paradigm. It focuses on learning a single, compact yet effective set of inheritable parameters that can be repeatedly used to initialize a wide variety of variable-sized models. This one-time upfront cost is central to ALT's long-term efficiency.

We compare ALT against alternative approaches that also invest significant resources, such as IMwLM (Xu et al. (2024)) and MatFormer (Devvrit et al. (2023)). As shown in our main paper (see Fig.3r and Fig.3s), these methods typically rely on pre-training a much larger or universal model to support the initialization of smaller target models. For instance, IMwLM utilizes a 129M-parameter model pre-trained for 300 epochs, while MatFormer employs an 85M-parameter universal model.

Tab.5 highlights ALT's distinct advantages in both efficiency and flexibility. First, direct evaluation on ImageNet-1K (without any fine-tuning) reveals that ALT provides a significantly higher initialization quality compared to IMwLM. Second, when fine-tuned, ALT often achieves superior performance even with smaller descendant models compared to MatFormer's supported models. Most notably, ALT accomplishes this by inheriting and utilizing a mere **15**M parameters. This dramatic reduction in the size of the inheritable parameter set underscores ALT's superior portability and flexibility, as its benefits are not contingent on storing or accessing a large source model.

**Orthogonality to Advanced Training Strategies.** It is crucial to distinguish between the model *initialization* and the subsequent *fine-tuning strategy* applied to it. Techniques such as strong data augmentations (*e.g.*, RandAug (Cubuk et al. (2020)), Mixup (Zhang et al. (2017))), extended training schedules, and advanced strategies like Sharpness-Aware Minimization (SAM) (Foret et al. (2020)) are powerful tools for enhancing downstream performance.

These techniques, however, are entirely **orthogonal** to ALT's contribution. ALT provides a high-quality starting point (initialization), upon which any of these advanced training strategies can be subsequently applied. The benefits are expected to be additive.

To isolate and demonstrate the standalone efficacy of ALT, we conducted further comparisons against reported results from SAM, LookSAM (Liu et al. (2022)), and Look-LayerSAM (Liu et al. (2022)). As noted in their original paper (Tab.2 and Tab.3), these methods achieve their strong performance through significantly heavier training regimens, often involving 300-epoch schedules, extremely large batch sizes (*e.g.*, 4096 to 32k on hundreds of TPUs), and strong augmentations.

In stark contrast, the results for ALT presented in Tab.6 are achieved with an extremely lightweight fine-tuning setup: **a mere 5-epoch tuning schedule**, a standard batch size of 128 on two GPUs, and **without any complex training modifications**. Despite this remarkably efficient fine-tuning process, ALT achieves performance that is competitive with, and in some cases superior to, these much heavier training paradigms. This result clearly demonstrates the profound impact of a high-quality initialization in accelerating convergence and achieving strong performance with minimal downstream training effort. We anticipate that combining ALT's superior initialization with such extended and robust training strategies would yield even greater performance gains.

## A.6   COMPUTATIONAL COST ANALYSIS OF AUX-NET TRAINING

This section provides a direct comparison of the computational cost required to train the auxiliary model (Aux-Net) for our proposed ALT and the baseline LeTs (Xia et al. (2024c)).

**Experimental Configuration.** To ensure a fair comparison, we standardize the experimental setup for training the Aux-Net on the ImageNet-1K dataset. For both ALT and LeTs, the objective is to train an Aux-H16-L24 (16 heads, 24 layers) model derived from learngene module (6 heads, 8 layers). The distinction between the two setups lies in the transformation matrices: ALT employs our SVD-parameterized matrices and dynamic component adaptation, whereas LeTs utilizes standard dense matrices. Both auxiliary models were trained for 300 epochs.

**Results and Trade-off Analysis.** The total computational cost for training the Aux-Net, measured in GPU hours, was approximately 390 hours for LeTs and 445 hours for ALT. This represents a marginal increase of only 14% in training time for our approach.

We argue that this modest additional computational overhead constitutes a highly favorable trade-off. In exchange for this slight increase in one-time training cost, ALT delivers a substantial and fundamental benefit: the resulting set of inheritable parameters is over **3× more compact** than that produced by LeTs (15M vs. 53M). This dramatic reduction in parameter size is not merely a numerical improvement; it directly addresses the central challenge of the Learngene framework—learning a maximally efficient parameter set for broad and flexible reuse.

**One-Time Investment for Long-Term Gain.** Furthermore, it is critical to contextualize this computational cost as a **one-time, upfront investment**. The Aux-Net is trained only once to generate the set of inheritable parameters. This modest additional training time is thus amortized over a potentially vast number of applications, as the resulting parameter set—being significantly more compact—offers superior portability and versatility for initializing a multitude of diverse, variable-sized descendant models.

## A.7   ON THE NECESSITY OF INHERITABLE PARAMETER REDUCTION IN THE LEARNGENE PARADIGM

This section elaborates on the core motivation of our work: the argument that achieving a drastic reduction in the number of inheritable parameters is not merely an incremental improvement, but a fundamental necessity for realizing the full potential of the Learngene paradigm. Our rationale is twofold, resting on both the conceptual advancement of the paradigm and the enablement of its practical applicability.

**Conceptual Advancement: Pursuing Maximal Inheritable Efficiency.** At a conceptual level, a primary—yet previously underexplored—goal of the Learngene paradigm is to achieve maximal inheritable efficiency. The central question is how compactly a large ancestry model's knowledge can be condensed into a reusable, inheritable form. While prior methods have made valuable progress, typically achieving a 3–6× reduction in inherited parameters over the ancestry model, our work demonstrates that this frontier can be pushed significantly further. ALT establishes a new state-of-the-art by achieving a **20–25× parameter reduction**. This represents a substantial leap towards the paradigm's foundational vision of extreme compactness and efficiency.

**Practical Imperative: Enabling Real-World Applicability.** This conceptual leap has direct and critical practical consequences, primarily by enhancing the portability and flexibility essential for real-world deployment. Many practical scenarios operate under strict resource constraints (*e.g.*, on-device deployment) where the size of the inheritable parameters can be a prohibitive factor.

To illustrate this with a concrete example from our vision experiments: consider a common resource-constrained scenario with a hard limit of less than 40M total parameters for the final descendant model. The state-of-the-art LeTs method, with its 53M inheritable parameters, is fundamentally unusable in this context; its inheritable parameter set alone exceeds the constraint, making initialization of any viable model impossible. In stark contrast, ALT's remarkably compact 15M parameter set provides the essential flexibility to initialize a wide range of models that comfortably satisfy this constraint.

Therefore, the drastic parameter reduction achieved by ALT is *not simply a quantitative improvement but a qualitative one*. It acts as an enabling factor, unlocking the Learngene paradigm for a vast new range of practical, resource-sensitive applications that were previously inaccessible.

### A.8    STRATEGY FOR DES-NET INITIALIZATION

The focus of our work is on learning an efficient set of inheritable parameters (the learngene and its associated transformation matrices). For the subsequent step of utilizing these parameters to initialize variable-sized descendant models (Des-Nets), we directly adopt the initialization strategies proposed in LeTs (Xia et al. (2024c)), as developing new initialization strategies is outside the scope of our core contributions. The two key strategies for width and depth initialization are summarized below.

**Continuous Selection for Width Initialization.** For initialization along width, we first reconstruct the full transformation matrix from its SVD components, *i.e.*, $\boldsymbol{F}^{(k)} = \boldsymbol{U}^{(k)}\boldsymbol{\Sigma}^{(k)}\boldsymbol{V}^{(k)}$. For the initialization along width, the required transformation sub-matrices are formed by selecting a *continuous* block from these reconstructed transformation matrices to satisfy each Des-Net's dimensional requirements. For consistency, our implementation defaults to selecting the first $n$ rows, where $n$ is determined by the target dimension. This continuous selection strategy is applied consistently across different modules to preserve the learned connectivity patterns between neurons.

**Sequential Selection for Depth Initialization.** When constructing the Aux-Net, learngene matrices have been partitioned into multiple groups. For the initialization along depth, the selection operates on the level of learngene groups. These groups are selected sequentially according to a predefined order, which can be configured to emphasize shallower or deeper learngene matrices. Once the learngene groups are selected, the corresponding elements from the depth transformation matrix $\boldsymbol{G}$ are used to linearly combine them into the final layers of the Des-Net. Our default implementation allocates a slightly greater number of coefficient groups (rows of $\boldsymbol{G}$) to shallower learngene groups.

### A.9    MORE COMPARISON BETWEEN ALT AND LOW-RANK METHODS

Low-rank decomposition underpins many areas—from model compression (Chen et al. (2021); Hsu et al. (2022); Li et al. (2023); Wang et al. (2024d)) to word-embedding reduction (Tanwar et al. (2018)). LoSparse (Li et al. (2023)) combines low-rank approximation with sparsity to compress Transformer weight matrices. With the rapid development of pretrained models, low-rank methods, particularly singular value decomposition (SVD), have emerged as a promising strategy for parameter-efficient fine-tuning (PEFT) (Hu et al. (2022); Zhang et al. (2023); Meng et al. (2024); Wang et al. (2024a)). SARA (Gu et al. (2024)) applies SVD at initialization to determine the optimal rank for pretrained weights. PiSSA (Meng et al. (2024)) and MiLoRA (Wang et al. (2024a))refine LoRA's initialization through SVD-based methods. Among these, AdaLoRA (Zhang et al. (2023)) is a notable work that dynamically adapts the rank of LoRA modules by pruning SVD components based on an additive importance score.

While our work draws inspiration from this line of works, it differs fundamentally in both its **objective**, its **methodology** and its **effectiveness** in our case.

- **Objective.** First, our primary objective is to address a critical challenge within the Learngene framework: learning and inheriting a single, highly-compact, and reusable set of parameters for initializing a *family* of downstream models. This contrasts with the standard PEFT goal, which focuses on adapting a large pretrained model to different downstream tasks.
- **Methodology.** Second, our methodology introduces several key innovations that diverge significantly from prior approaches. Instead of a simple additive metric, we propose the **Gated Importance Score (GIS)**, a novel, SVD-inspired metric that captures the multiplicative interplay between the energy and structural direction of each low-rank component. Furthermore, we systematically design and explore two distinct adaptation strategies: one robust **Flat Global Adaptation (FGA)** and one **Hierarchical Component Adaptation (HCA)** strategy which introduces a two-level, capacity-aware mechanism to respect the model's modular structure. In essence, our work focuses on *learning a structured, adaptable low-rank transformation* for a broader initialization purpose.
- **Effectiveness.** Finally, the empirical results presented in Tab.2 validate the superiority of our proposed methodology for our goal of learning efficient inheritable parameters. Specifically,

the performance gap between our approach using GIS and the baseline using an additive score provides clear evidence that our **Gated Importance Score** is better suited for our dynamic component adaptation. Compared to the additive score, our GIS appears to provide a more effective signal for identifying truly essential components, leading to a superior set of inherited parameters and consistently better downstream performance.

In essence, while PEFT methods use low-rank adaptation as a tool for *task-specific fine-tuning*, our work leverages it to forge a structured, adaptable, and highly efficient *initializer*. This conceptual shift and the novel methodological components (GIS and HCA) represent a distinct contribution to the field of efficient model initialization.

Table 9: Comparison between LoRA (Hu et al. (2022)), GPS (Zhang et al. (2024b)), SNF (Wang et al. (2023c)) and ALT on CIFAR100. We refer the results reported in Tab.4 of (Xin et al. (2024)).

| Model | LoRA | GPS | SNF | ALT |
|---|---|---|---|---|
| Des-H12-L12 | 67.1 | 81.1 | 84.0 | **90.4** |

**Empirical Comparison with PEFT.** Experimentally, we compare ALT against LoRA (Hu et al. (2022)), GPS (Zhang et al. (2024b)) and SNF (Wang et al. (2023c)) on CIFAR100 with Des-H12-L12 model (86M) and refer the results reported in Tab.4 of (Xin et al. (2024)). As shown in Tab.9, ALT achieves better performance. Notably, LoRA requires the entire 86M pretrained parameters to initialize the downstream model. In contrast, ALT only requires its highly compact parameter set (15.3M) to initialize.

Crucially, **ALT and PEFT methods are orthogonal and can be used in sequence**. After initializing a model via ALT, one can still apply any PEFT method on the ALT-initialized Des-Net.

## A.10 ALT VERSUS MODEL COMPRESSION

### A.10.1 ALT VERSUS KNOWLEDGE DISTILLATION

While our ALT framework incorporates a knowledge distillation step during the training of its auxiliary model, it is crucial to distinguish our overall paradigm from conventional Knowledge Distillation (KD) (Hinton et al. (2015); Jiao et al. (2020); Wang et al. (2020); Chen et al. (2022); Zhang et al. (2022a); Ren et al. (2023); Bai et al. (2023); Zhang et al. (2024a); Cui et al. (2025)). Although both approaches transfer knowledge from a larger model, they differ fundamentally in their core **objectives**, **methodologies**, and resulting **deployment paradigms**.

**Contrasting Objectives and Paradigms.** The primary goal of conventional KD is typically to *compress* a single large teacher model into a specific, smaller student model, or to improve that student's performance by leveraging the teacher's "dark knowledge" throughout its training. This process necessitates a direct, coupled teacher-student interaction for *every new student model* one wishes to train.

In stark contrast, ALT addresses a different, more general challenge within the Learngene framework: to *decouple* the knowledge extraction from its subsequent application. Our objective is not to train a single student, but to learn a **single and highly-compact set of inheritable parameters** that can act as a powerful initializer for a whole *family* of diverse, variable-sized descendant models. This embodies a **"learn once, initialize many"** paradigm. This one-time knowledge extraction completely eliminates the need to store or repeatedly access the large, computationally expensive ancestry model for subsequent initializations, a capability that lies outside the primary scope of conventional KD methods.

**Methodological Distinctions.** This difference in objective necessitates a distinct methodology. ALT distills knowledge from the ancestry model to an auxiliary model **only once**. This auxiliary model is uniquely structured to produce a compact learngene module and a set of efficient, SVD-parameterized transformation matrices. Once this initial learning phase is complete, the large ancestry model is no longer needed. Any number of descendant models can then be initialized directly and efficiently from this compact, portable set of inheritable parameters. In essence, while KD typically involves

an ongoing teacher-student interaction for each training run, ALT front-loads this interaction into a single knowledge consolidation step.

**Empirical Comparison and Efficiency Gains.** To empirically validate these conceptual distinctions, we conduct a direct and fair comparison between ALT and a standard soft distillation method (Soft-distillation) (Hinton et al. (2015)), because we also select soft distillation (Hinton et al. (2015)) to train our Aux-Net. This choice is motivated by fairness, as our own ALT framework also utilizes soft distillation during the training of its auxiliary model (Aux-Net) with the same 304M ViT-Large Dosovitskiy et al. (2021) teacher. We train 9 different student models on ImageNet-1K, with sizes ranging from 22M to 254M parameters.

The results, summarized in Tab.7 and Fig.3q, demonstrate that ALT is not only highly efficient but also highly effective. Despite utilizing a remarkably lightweight fine-tuning schedule (5-15 epochs vs. Soft distillation's 100 epochs), ALT achieves **competitive, and in many cases superior performance**. More strikingly, this performance is achieved with a fraction of the computational cost. Considering only the student fine-tuning phase, ALT reduces the required GPU hours from 549 (for Soft distillation) to just 43—a **12.8× reduction**. Even when accounting for the one-time, 445 GPU-hour cost of training ALT's inheritable parameter set, the total computational costs for training all 9 models remains lower for ALT (488 hours vs. 549 hours).

This highlights the core advantage of our paradigm: the upfront cost of learning the inheritable set is **amortized** over the number of descendant models. The more models one needs to initialize, the greater ALT's efficiency advantage becomes. This confirms that ALT provides a highly effective and computationally superior alternative to conventional knowledge distillation for the specific, yet critical, task of deploying families of variable-sized models under diverse constraints.

### A.10.2 ALT versus Pruning

Conceptually, pruning aims to obtain a single compact model as an endpoint. In contrast, ALT, embodying a "learn once, initialize many" paradigm, learns a highly-compact, reusable parameter set for initializing variable-sized models. Furthermore, pruning often results in irregular model architectures (*e.g.*, varying widths per layer), which may complicate deployment, whereas ALT always initializes standard, regular Transformer architectures.

Table 10: Comparison between WDPruning (Yu et al. (2022)), VBP (Berisha et al. (2025)) and ALT. "wo KD" and "w/ KD" mean without and with knowledge distillation respectively.

| Method | Params(M) | Main Process | Top-1 Acc.(%) |
| --- | --- | --- | --- |
| WDPruning | 55.3 | 100 epoch pruning (wo KD) | 80.8 |
| VBP | 55.4 | 10 epoch fine-tuning (w/ KD) | 80.7 |
| ALT | 51.5 | 5 epoch fine-tuning (wo KD) | **80.9** |

Experimentally, we compare ALT against WDPruning (Yu et al. (2022)) and VBP (Berisha et al. (2025)): pruning ImageNet-1K pretrained Des-Nets to a target size (results are referred in their original paper). As shown in Tab.10, ALT achieves **better performance and obtains smaller model**. Both WDPruning and VBP require the entire pretrained parameters to initialize the pre-pruning model and then prune the model to target ones. Besides, VBP needs 10-epoch fine-tuning with knowledge distillation. In contrast, ALT only requires its reusable and compact parameter set to initialize target model and simply fine-tunes it for 5 epochs.

### A.11 Discussion on Applying ALT to Large Language Models (LLMs)

This section discusses the applicability, scalability, and performance characteristics of our ALT in the context of Large Language Models (LLMs).

**Existing Validation on LLMs.** Our primary validation of ALT on LLMs was conducted using the GPT-2 (1.5B) as an ancestry model (Ans-Net). As detailed in the main paper, we successfully learned a set of approximately 62M inheritable parameters, achieving a remarkable **25×** parameter reduction (or **60×** when excluding embedding and head parameters). The practical benefit of this compact parameter set was demonstrated by a significant enhancement in training efficiency: as shown in

Fig.4b, a 354.3M-parameter GPT-2-based descendant model (Des-H16-L24) initialized with ALT achieved its target performance with roughly a $2\times$ reduction in required pre-training steps compared to training from scratch.

**Scalability to Larger LLMs and Data Dependency.** ALT is designed to be directly scalable to much larger models, such as the LLaMA (Touvron et al. (2023)) series (7B, 65B, *etc.*). It is estimated that, one could employ LLaMA-7B as an Ans-Net to train a corresponding Aux-Net, from which a highly efficient inheritable parameter set (*e.g.*, 130M) could be derived.

However, a critical prerequisite for ALT's Stage 1 (Training Inheritable Parameters) is access to the original pre-training dataset, or a representative equivalent. This is necessary to train the Aux-Net under the guidance of the Ans-Net. This data dependency presented a practical obstacle during our exploration of scaling to the latest models like the Qwen3 series (Yang et al. (2025)), as their pre-training corpora are not publicly available. **It is crucial to emphasize that this limitation is not inherent to the ALT itself, but rather reflects the current landscape of proprietary, closed-source pre-training datasets for most state-of-the-art Large Language Models.** Our ALT is designed to be model-agnostic and is fully applicable to LLMs. Given access to the requisite pre-training data, our framework could be straightforwardly applied to these prominent LLMs. Therefore, overcoming this data requirement—perhaps by exploring large-scale public datasets or advanced data synthesis techniques—remains a promising and important direction for future work, which would unlock the full potential of ALT for LLMs.

**Clarification of the Two-Stage Process.** It is important to clarify that ALT does not directly "shrink" a large model into a smaller one. Instead, it employs a **two-stage process**: (1) First, it *learns* an efficient and dramatically smaller set of inheritable parameters from the large Ans-Net. (2) Second, this compact parameter set is then used to *initialize* a family of variable-sized descendant models. The primary focus of our experiments has been to demonstrate that Des-Nets initialized by ALT exhibit drastically improved training efficiency and downstream performance compared to those training from scratch, when compared at the same scale.

**Cross-Scale Performance Potential.** A key question is whether a smaller model initialized by ALT can achieve performance comparable to a much larger, standardly trained model. Our experiments in the vision domain provide compelling evidence for this potential. For example, a significantly smaller 29.2M ViT model, initialized by ALT and fine-tuned for a mere 5 epochs, achieved a Top-1 accuracy of 80.5% on ImageNet-1K. This performance is nearly competitive to that of a 114.9M ViT model (nearly $4\times$ larger) trained from scratch for a full 150 epochs (80.4% accuracy). This result strongly suggests that ALT's high-quality initialization can enable smaller models to reach performance levels previously only attainable by much larger architectures, and with substantially less fine-tuning effort.

**Determining the Size of Inheritable Parameters.** The total number of inheritable parameters in ALT is the sum of the parameters from the **learngene module** and the **transformation matrices**. A key feature of our ALT framework is that the SVD-based parameterization and dynamic adaptation drastically reduce the transformation matrices parameters. Consequently, the overall scale of the inheritable parameter set is **dominantly determined by the user-defined size of the learngene module itself**. This is evident in our vision experiments: of about 15.3M total inheritable parameters, the learngene module accounts for about 15.0M ($\approx 98\%$), while all the highly-efficient transformation matrices contribute only about 0.3M. **This principle extends directly to LLM scenarios.** When applying ALT to models like LLaMA-7B or 65B, the user would first choose a desired size for the learngene module based on their efficiency requirements. For instance, if one configures a 130M-parameter learngene module (*e.g.*, a 12-layer one), the final total number of inheritable parameters would be only marginally larger (*e.g.*, in the 130–135M range), even for a LLaMA-65B Ans-Net. The size of the inheritable parameters is a flexible, user-controlled design choice.

**Applicability to Decoder Structures without Adjustments.** ALT's core mechanisms do not require any adjustments when applying to the decoder structure. Our ALT method is designed to be **architecture-agnostic** at a macro level because it operates on the linear matrices common to all Transformer models. The core components of a decoder-only model like LLaMA are still linear matrices within Attention or Feed-Forward Network blocks. Crucially, ALT's operations, such as SVD parameterization, GIS calculation, and adaptation strategies (HCA/FGA), are applied directly to the linear matrices within these blocks. Therefore, **ALT can be applied seamlessly to**

Table 11: Tasks and evaluation metrics for the GLUE and SQuAD benchmarks.

| Task | Primary Metric(s) |
|------|-------------------|
| *GLUE Benchmark* | |
| SST-2 | Accuracy (Acc.) |
| MNLI | Accuracy (Acc.) |
| MRPC | F1 / Accuracy (F1/Acc.) |
| QNLI | Accuracy (Acc.) |
| QQP | Accuracy / F1 (Acc./F1) |
| STS-B | Pearson / Spearman Corr. (P./S. Corr.) |
| *Question Answering* | |
| SQuADv1.1 | F1 / Exact Match (F1/EM) |

**linear matrices of a decoder block in exactly the same manner as it is to an encoder block.** No modifications to our SVD parameterization or adaptive strategies are necessary.

## A.12 THE USE OF LARGE LANGUAGE MODELS (LLMS)

In adherence to the ICLR policy on the use of Large Language Models (LLMs), we wish to clarify the role of such tools in the preparation of this paper. We utilized a large language model, specifically Google's Gemini, as a general-purpose writing assistant. Its application was strictly limited to improving the clarity, conciseness, and grammatical correctness of our own original text. The core research ideas, including the formulation of the Gated Importance Score (GIS), the design of the Flat Global Adaptation (FGA) and Hierarchical Component Adaptation (HCA) strategies, as well as the experimental design and analysis of results, are entirely our own.

## A.13 THE RATIONALE AND ABLATION FOR THE INITIAL LEARNGENE STRUCTURE

Our rationale is twofold: ensuring a fair comparison and building upon established empirical findings. We conduct ablation studies to further validate this choice.

**(1) Ensuring a Fair Comparison.** Our primary objective is to demonstrate the superiority of our proposed ALT over existing state-of-the-art methods like LeTs within the Learngene paradigm. Therefore, adhering to the principle of controlled experimental setting, we adopt the identical "minimum effective skeleton" (H6-L8) used in LeTs. This ensures that **the observed performance gains are attributable solely to our novel contributions** (GIS, FGA, HCA, *etc.*), rather than being confounded by a potentially better, but different architectural starting point.

Table 12: Ablation study on varying the initial learngene structure. We train Aux-Nets comprising different learngene structures for 100 epochs on ImageNet-1K and then evaluate the 5-epoch tuning performance of an initialized Des-H8-L12 model.

| Learngene Structure | Inheritable Parameters(M) | Top-1 Acc. (%) |
|---------------------|---------------------------|----------------|
| H4-L6 (Smaller) | 6.0 | 69.3 |
| H6-L8 (Current setting) | 15.3 | 76.0 |
| H8-L8 (Larger) | 26.9 | 77.8 |

**(2) Building on an Established Baseline (LeTs).** The H6-L8 configuration is not arbitrary but is an empirically validated starting point that has emerged from the Learngene paradigm. It represents a widely-accepted trade-off between being compact enough to demonstrate high parameter efficiency and large enough to inherit the knowledge from the Ans-Net. To empirically address your question, we conduct a ablation study on ImageNet-1K by varying the initial learngene structure, evaluate and report the 5-epoch tuning performance of an initialized Des-H8-L12 model. As shown in Tab.12, the smaller H4-L6 one, while more efficient in inheritable parameters, fails to capture sufficient knowledge, leading to a significant drop in the Des-Net performance. Conversely, the larger H8-L10 one offers a performance improvement but at the cost of a substantial increase in

inheritable parameters, which contradicts **our core objective of maximizing inheritable-parameter efficiency**. These results demonstrate that our chosen H6-L8 configuration represents a compelling sweet spot, achieving a balance between Des-Net performance and inheritable parameter efficiency. **It is noteworthy that the learngene structure exhibits inherent flexibility and can be adapted by different users.**

**(3) On the Generalizability of ALT to Different Architectures.** We wish to clarify that our ALT is designed to be **model-agnostic**, and its core mechanisms are not limited to the standard Transformer architecture. The fundamental components of ALT, the Gated Importance Score (GIS), and the adaptation strategies (FGA/HCA)—operate on the linear matrices (*e.g.*, for query/key/value projections and MLP layers) that are common across the Transformer family. Since architectures like the Swin Transformer are also built upon these linear matrices, our ALT is directly applicable.

## A.14 HYPERPARAMETER SENSITIVITY ANALYSIS

Table 13: Ablation studies focusing on the final number of active components $H^{(T)}$ and the orthogonality regularization weight $\beta$. We train Aux-Nets for 100 epochs on ImageNet-1K and then evaluate the 5-epoch tuning performance of an initialized Des-H8-L12 model.

| $H^{(T)}$ | Top-1 Acc. (%) | $\beta$ | Top-1 Acc. (%) |
|---|---|---|---|
| 68 | 75.9 | 0 | 75.6 |
| 136 | **76.0** | 1e-3 | **76.0** |
| 272 | 75.9 | 1e-2 | 75.8 |

Here, we conduct two sets of ablation studies focusing on the final number of active components $H^{(T)}$ and the orthogonality regularization weight $\beta$. Specifically, we train Aux-Nets for 100 epochs on ImageNet-1K with different $H^{(t)}$ and $\beta$ to obtain different inheritable parameter sets, and then evaluate the 5-epoch tuning performance of an initialized Des-H8-L12 model. From Tab.13, we have the following observations.

**(1) Sensitivity to $H^{(T)}$.** The hyperparameter $H^{(T)}$ controls the number of parameters within the SVD-based transformation matrices. From the table above, we observe that different $H^{(T)}$ yield similar performance. This can be attributed to the fact that **these matrices constitute a minimal fraction of the total inheritable parameters (approximately 2%)**, whereas the learngene parameters dominate the vast majority of the inheritable parameters. However, it is crucial to clarify that **this low parameter count does not imply a lack of importance. On the contrary, it highlights the high parameter efficiency of our design.** Despite the compact size, these matrices play a pivotal role in the transformation of learngene parameters to construct the Aux-Net and in the effective initialization of the Des-Net. They function as a lightweight yet critical bridge, enabling effective knowledge transfer.

**(2) Sensitivity to $\beta$.** From the table above, we observe that removing the orthogonality constraint ($\beta = 0$) degrades performance, confirming its necessity for SVD-based parameterization. Moreover, the model's performance remains robust across a range of reasonable values for $\beta$ (1e-3 and 1e-2), indicating that our method is not overly sensitive to this hyperparameter. We attribute this robustness to the fact that the orthogonality regularization is applied to the transformation matrices. Since these matrices constitute only a small fraction of the total inheritable parameters, variations in the regularization strength do not significantly perturb the overall learning process of the inheritable parameters.

## A.15 DETAILED WORKFLOW OF LEARNGENE AND THE LETS METHOD

### A.15.1 THE LEARNGENE PARADIGM

As illustrated in Fig.1(a), the Learngene framework operates on a two-stage process:

**Stage 1 (Training the learngene):** It first learns a single compact parameter set, termed **learngene**, under the guidance of a large well-trained **Ancestry Model (Ans-Net)**.

**Stage 2 (Initialization from learngene):** The learngene is then inherited and transformed to initialize a family of variable-sized **Descendant Models (Des-Nets)**, which are subsequently fine-tuned on downstream tasks.

### A.15.2   THE LETs METHOD

This section provides a detailed summary of the Learnable Transformations (LeTs) method (Xia et al. (2024c)), which serves as the foundation upon which our ALT method is built. The core idea of LeTs is to learn a set of transformation matrices that can adapt a compact **learngene** to initialize variable-sized models. This process is primarily achieved through a three-step workflow: defining the transformations, training an auxiliary model, and initializing descendant models.

**(1) Learnable Transformations.** The LeTs framework begins with a **learngene module**, denoted as $\Theta^{\text{lg}} = [W_1, \dots, W_L]^\top$, which is composed of $L$ learngene matrices. LeTs then defines two types of learnable transformations to alter the dimensions of this module.

- **Width Transformation:** For each learngene matrix $W_l$, LeTs introduces two *dense* learnable matrices, $F_l^{\text{in}}$ and $F_l^{\text{out}}$, to expand its input and output dimensions, respectively. The transformation is applied via concatenation. First, the in-dimension is expanded to create an intermediate matrix $W_l'$:

$$W_l' = \text{Concat}(W_l, F_l^{\text{in}} W_l). \qquad (4)$$

  Then, the out-dimension of $W_l'$ is similarly expanded to produce the final width-transformed matrix:

$$W_l^{\text{wt}} = \text{Concat}\big(W_l', W_l'(F_l^{\text{out}})^\top\big). \qquad (5)$$

- **Depth Transformation:** After all learngene matrices have been width-transformed, the resulting set $\{W_l^{\text{wt}}\}_{l=1}^L$ is partitioned into groups. A depth transformation matrix, $G$, is then used to linearly combine the matrices within each group to form new, deeper layers for a target model. For instance, the $i$-th new layer $W_i^{\text{dt}}$ can be constructed from the $j_0$-th group of learngene matrices as:

$$W_i^{\text{dt}} = \sum_{k=1}^{L_{j_0}} G_{i,k} W_{j_0,k}^{\text{wt}}, \qquad (6)$$

  where $G_{i,k}$ is an entry in the matrix $G$.

**(2) Aux-Net Construction and Training.** LeTs utilizes these transformations to construct an **auxiliary model (Aux-Net)** from the compact learngene. This Aux-Net is then trained under the guidance of a large, pre-trained **Ancestry Model (Ans-Net)**. The training objective is a joint loss function that typically combines a standard task loss (*e.g.*, classification loss $\mathcal{L}_{\text{cls}}$) with a knowledge distillation loss ($\mathcal{L}_{\text{distill}}$), which minimizes the discrepancy between the logits of the Aux-Net and the Ans-Net:

$$\mathcal{L}_{\text{total}} = (1 - \lambda)\mathcal{L}_{\text{cls}} + \lambda\mathcal{L}_{\text{distill}}. \qquad (7)$$

This end-to-end training process simultaneously optimizes both the learngene parameters ($W_l$) and all the transformation matrices ($F_l^{\text{in}}, F_l^{\text{out}}, G$).

**(3) Des-Net Initialization.** Once the Aux-Net training is complete, the well-trained learngene and transformation matrices are used to initialize various descendant models (Des-Nets) with different sizes. To match the specific architectural requirements of a Des-Net, LeTs employs several parameter selection strategies.

- **For Width:** The primary strategy is *continuous selection*, where the first $n$ rows or columns are selected from the learned matrices $F^{\text{in}}$ and $F^{\text{out}}$ to form the smaller transformation matrices required by the Des-Net.
- **For Depth:** The strategy is *sequential selection*, where learngene groups are selected in a predefined order, and the corresponding rows from the learned matrix $G$ are chosen to construct the target number of layers.

After being initialized through this process, the Des-Net is then fine-tuned on downstream tasks.

## A.16 COMPARISON BETWEEN SVD-BASED PARAMETERIZATION AND A LOW-RANK PRODUCT

Here, we discuss our choice of SVD-based parameterization rather than a simpler product of low-rank matrices (*e.g.*, $F = BA$).

**(1) An Empirical-Driven Design Choice.** Our choice of SVD is not arbitrary but is **directly inspired by our initial empirical analysis** of the LeTs transformation matrices. Our investigation begins by utilizing SVD as an analytical tool, which reveals that these matrices exhibit rapid and varying singular value decay. This insight naturally leads us to adopt an SVD-like parameterization ($F = U\Sigma V$), as it directly models the empirically observed structure and forms the foundation for our subsequent component adaptation.

**(2) Corroborating Evidence from Prior Works in Other Domains.** Our choice is further supported by empirical evidence from the Parameter-Efficient Fine-Tuning (PEFT) domain. For instance, AdaLoRa (Zhang et al. (2023)), which employs a SVD-form incremental updates, consistently demonstrated superior performance over the original LoRA (Hu et al. (2022)) which uses the simpler $BA$ decomposition. While the domains differ, this provides strong corroborating evidence that the structural properties of an SVD-based formulation can offer significant advantages over low-rank product form.

Table 14: Comparison between ALT (SVD-based parameterization) and ALT-LoRA (LoRA-based parameterization). We train Aux-Nets for 100 epochs on ImageNet-1K and then evaluate the 5-epoch tuning performance of an initialized Des-H8-L12 model.

| Method | Inheritable parameters (M) | Top-1 Acc. (%) |
|---|---|---|
| ALT-LoRA | 16.0 | 72.2 |
| ALT | 15.3 | **76.0** |

**(4) New Study.** To empirically validate these advantages, we conduct an additional study. We establish a strong baseline, termed "ALT-LoRA", which utilizes LoRA-based parameterization ($F = BA$) to replace the SVD-based one. As shown in Tab.14, our ALT outperforms the low-rank product formulation ALT-LoRA. This provides compelling empirical evidence that the design advantages of the SVD-based parameterization, **particularly its decoupled nature that facilitates our GIS metric and dynamic adaptation process,** are critical for achieving superior performance. The marginally higher inheritable parameters in ALT-LoRA stems from the larger number of retained learnable components.

## A.17 INCORPORATING FINER-GRAINED KNOWLEDGE TRANSFER

Here, we conduct experiments where we enhance our Aux-Net training by incorporating distillation from intermediate layers, specifically from attention maps and hidden states.

**(1) Enhanced Distillation Objective.** We augment our original total training loss, $\mathcal{L}_{\text{total}}$, with two new fine-grained distillation terms:

$$\mathcal{L}_{\text{enhanced}} = \mathcal{L}_{\text{total}} + \gamma\mathcal{L}_{\text{attn-distill}} + \delta\mathcal{L}_{\text{hidden-distill}}, \tag{8}$$

where $\gamma$ and $\delta$ are trade-off coefficients.

**(2) Self-Attention Distillation.** We compute the attention relation matrix $R$ for both Aux-Net and Ans-Net as $R = \text{softmax}(QK^\top/\sqrt{d})$, where $Q, K$ are the query and key matrices. The loss $\mathcal{L}_{\text{attn-distill}}$ is designed as $\left\|R^S - R^T\right\|_2^2$ to minimize the Mean Squared Error (MSE) between their corresponding relation matrices. During empirical experiments, we observe that directly aligning these relation matrices led to training instabilities. Consequently, we omit the attention distillation term in our final training process.

**(3) Hidden-State Distillation.** Let $H^S$ and $H^T$ represent the hidden states of the Aux-Net and Ans-Net, respectively. The hidden-state distillation loss $\mathcal{L}_{\text{hidden-distill}}$ is computed using MSE as $\left\|H^T - \phi(H^S)\right\|_2^2$. It is worth noting that since the feature dimensions of the Aux-Net and Ans-Net

are inherently aligned, the learnable projection layer $\phi(\cdot)$ is not required (or can be regarded as an identity mapping). This objective forces the Aux-Net to regress the Ans-Net's semantic features directly, facilitating effective representation learning without introducing additional parameters.

Table 15: Impact of fine-grained distillation. We train Aux-Nets for 100 epochs on ImageNet-1K and then evaluate the 5-epoch tuning performance of an initialized Des-H8-L12 model.

| Aux-Net Training Method | Top-1 Acc. (%) |
|---|---|
| ALT (Soft Distillation) | 76.0 |
| ALT + Fine-grained Distillation | **76.3** |

**(4) New Experimental Results.** To validate the effectiveness of fine-grained knowledge transfer, we conduct additional experiments comparing our original ALT (utilizing only soft distillation) against the enhanced version, denoted as "ALT + Fine-grained Distillation". Specifically, for the enhanced version, we trained the Aux-Net on ImageNet-1K for 100 epochs, incorporating the fine-grained distillation. As shown in Tab.15, we observe that the enhanced version yields a improvement in the performance of the downstream Des-H8-L12 model. This empirical evidence indicates that incorporating finer-grained intermediate knowledge leads to a higher-quality set of inheritable parameters.

### A.18    THE BREAK-EVEN ANALYSIS OF ALT'S COMPUTATIONAL COST

Table 16: Break-even analysis of ALT's total compute cost versus baselines on ImageNet-1K. Total costs are shown for the specific number of models (N) compared in our experiments.

| Method | Upfront (h) | Avg. FT/Model (h) | Break-even (N) | Total Cost (N models) |
|---|---|---|---|---|
| *(a) Comparison vs. Training from Scratch (Scratch)* | | | | |
| Scratch | 0 | ˜63.5 | ˜7 | N=8: ˜508 h |
| **ALT (ours)** | **˜445** | **˜1.4** | | **N=8: ˜456 h** |
| *(b) Comparison vs. Soft Distillation* | | | | |
| Soft Distillation | 0 | ˜61.0 | ˜8 | N=9: ˜549 h |
| **ALT (ours)** | **˜445** | **˜4.8** | | **N=9: ˜488 h** |

ALT's efficiency stems from its **"learn once, initialize many"** paradigm, where a one-time, upfront computational investment in training the Aux-Net is amortized over the initialization of multiple descendant models, each requiring only a lightweight fine-tuning schedule.

The analysis is based on the following: **(a) Upfront Cost:** For ALT, this is the 445 GPU hours of training the inheritable parameters. For Scratch and Soft Distillation, this is zero. **(b) Fine-Tuning (FT) Cost:** This is the average time to train one Des-Net. Crucially, the fine-tuning schedule for ALT is adapted to the strength of the baseline. To achieve performance competitive with the standard Scratch baseline, a very lightweight 0-5 epoch fine-tuning suffices for most Des-Nets. To match the stronger Soft Distillation baseline, we modestly extended the fine-tuning up to 15 epochs for some Des-Nets. From Tab.16, we have the following observations.

**(1) ALT vs. Training from Scratch.** Training a model from scratch (100-300 epoch) requires a substantial downstream cost (˜63.5h/model). In contrast, **ALT requires only a minimal fine-tuning effort (˜1.4h/model).** The break-even point is reached at approximately **7 models** $(445/(63.5-1.4) \approx 7.2)$. As our direct cost comparison for N=8 models (ranging from 29.2M to 171.6M) shows, ALT's total cost (456h) is already lower than that of training them from scratch (508h).

**(2) ALT vs. Soft Distillation.** Similarly, each new model for Soft Distillation requires a full 100-epoch training schedule (˜61h/model). Even when we use a slightly longer fine-tuning schedule for ALT in this comparison, **its per-model cost (˜4.8h/model) remains an order of magnitude lower.** The break-even point is met at approximately **8 models** $(445/(61.0 - 4.8) \approx 7.9)$. The total cost for training 9 models (ranging from 22.1M to 253.9M) confirms this, with ALT being more efficient (488h vs. 549h). This clearly demonstrates that **ALT's "learn once, initialize many" paradigm is significantly more efficient than repeated, per-model distillation.**

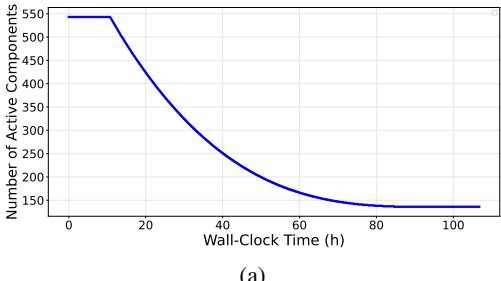 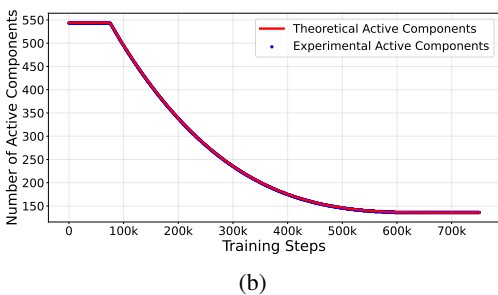

(a)                   (b)

Figure 9: (a) The number of active components vs. the wall time during the 300 epoch Aux-Net training (4 GPUs). (b) The theoretical and experimental number of active components vs. the training step during the 300 epoch Aux-Net training.

**(3) ALT vs. LeTs: A Trade-off between Compute and Parameter Efficiency.** As LeTs shares the same "learn once" paradigm, their downstream costs are identical. LeTs has a lower upfront computational cost (390h vs. 445h for the Aux-Net). Therefore, from a *purely computational cost perspective*, LeTs is slightly more efficient. **However, this comparison overlooks the central goal of our work: inheritable-parameter efficiency**. ALT incurs a marginal ~14% increase in one-time, upfront training compute, but in return, it delivers a **dramatic 3.5× reduction in the size of the inheritable parameter set** (15M vs. 53M). For real-world applications where storage and memory footprint are critical constraints (*e.g.*, edge devices), **this substantial gain in parameter efficiency is often far more valuable than the modest difference in one-time training cost.**

### A.19    THE ANALYSIS OF TRAINING-TIME FOOTPRINT

Here, we provide a comprehensive comparison of peak memory and FLOPs for ALT versus LeTs, and the evolution of active components during Stage-1 300 epoch Aux-Net training of ALT.

Table 17: Comparison of peak memory and FLOPs for the Aux-Net between ALT and LeTs.

| Method | Peak Memory (GB) | FLOPs (G) |
|---|---|---|
| LeTs | ˜41.6 | ˜922.4 |
| **ALT (ours)** | ˜41.2 | ˜939.4 |

**(1) Peak Memory and FLOPs Analysis.** For a fair comparison, both ALT and LeTs are configured to train the Aux-H16-L24 model on ImageNet-1K. We measure the per-GPU peak memory consumption and the FLOPs of the Aux-Net for both methods. We calculate the model's FLOPs using the torchprofile library. From Tab.17, the peak memory and FLOPs for ALT and LeTs are **nearly identical**.

**(2) Evolution of Active Components vs. Wall-Clock Time.** To visualize our dynamic adaptation process, we plot the total number of active components, $H(t)$, as a function of the wall-clock training time. Fig.9(a) shows the number of active components vs. the wall time. Fig.9(b) illustrates how the number of active components decreases over the training process theoretically and experimentally. This demonstrates that **our adaptation mechanism operates as designed, progressively and stably refining the structure of the transformation matrices.**

### A.20    THE ROBUSTNESS UNDER LOW-DATA SCENARIOS

Table 18: Top-1 accuracy (%) of Des-H6-L12 on CIFAR-100 with 300-epoch fine-tuning under low-data fine-tuning settings. The performance under 30% data is averaged across 2 different seeds.

| Training Data | Pre-Fin | ALT (ours) |
|---|---|---|
| 30% | 72.2 | **85.8** |
| 50% | 77.5 | **87.8** |
| 100% (Full) | 86.9 | **89.6** |

We conduct experiments focusing on low-data fine-tuning scenarios, and provide analyses on calibration and stability.

**(1) Performance in Low-Data Fine-tuning Scenarios.** To assess ALT's performance in data-constrained settings, we conduct experiments on subsets of the CIFAR-100 training data, specifically using **30%** and **50%** of the data for 300-epoch fine-tuning. We compare our ALT-initialized model against the Pre-Fin (pre-training on ImageNet-1K + fine-tuning) baseline. Tab.18 shows that **ALT's performance advantage is even more pronounced in low-data regimes**. For example, at 30% data, ALT outperforms Pre-Fin by 13.6%. This confirms that the rich prior knowledge inherited via ALT from the large Ans-Net substantially improves generalization when training data is scarce.

**(2) Calibration and Stability at Small Data Budgets.** We further analyze the experiments on the 30% CIFAR-100 subset to evaluate calibration and stability, as requested.

**Stability:** We run the 30% data experiment for both ALT and Pre-Fin using 2 different random seeds. The standard deviation of the Top-1 accuracy for ALT is $\pm 0.32\%$, which is higher than that for Pre-Fin ($\pm 0.04\%$). Despite the higher standard deviation, the mean performance of ALT is substantially higher, and the intervals (as indicated by the standard deviations) do not overlap.

**Calibration:** We measure the Expected Calibration Error (ECE) [1] on the validation set. At 50% data, the ALT-initialized model achieves an ECE of 0.1373, which is better than the Pre-Fin baseline's ECE of 0.1571. This indicates that models initialized with ALT are not only more accurate but also produce more reliable confidence scores.

In summary, these results provide strong evidence that ALT offers superior robustness in challenging low-data scenarios.

## A.21 PARAMETER COUNTING CONVENTIONS AND STORAGE FOOTPRINT

Table 19: Detailed breakdown of inheritable parameter counts and storage for ALT.

| Metric | ALT (ours) |
|---|---|
| *(1) Pre-pruning Inheritable Parameters (at $t = 0$)* | |
| - Learngene Module | ˜15 M |
| - Initial Transformation Matrices | ˜1.1 M (SVD-based, at $H^{(0)}$) |
| Total Pre-pruning | ˜16.1 M |
| *(2) During Aux-Net Training* | |
| Avg. Active Components | ˜226 |
| *(3) Final Initializer (Post-pruning, at $t = T$)* | |
| Final Inheritable Params | ˜15.3 M |
| Storage Size to Ship (*e.g.*, .pt file) | ˜64.2 MB |

**(1) Pre-pruning Parameter Count.** As shown in Tab.19, the "pre-pruning" count refers to the total inheritable parameters at the beginning of the Aux-Net training ($t = 0$). For ALT, this is the sum of the learngene and our SVD-parameterized matrices at their initial maximum component configuration ($H^{(0)}$). It is noteworthy that even before any dynamic adaptation, ALT's SVD-based parameterization is already significantly more compact than the dense matrices.

**(2) Average Active Components over Training.** This metric represents the time-averaged number of active SVD components throughout the entire Aux-Net training process. We calculate this by averaging the value of $H(t)$.

**(3) Runtime Storage Required to Ship a Reusable Initializer.** The final "shippable" initializer is a self-contained checkpoint file. Crucially, **no additional metadata is required** to reconstruct the per-matrix selections. Our default "continuous selection" strategy for Des-Net initialization is predefined (*i.e.*, it always selects the first $n$ components based on the target dimension), so this information does not need to be stored.

## A.22 EXPERIMENTAL DETAILS

In this section, we describe the experimental details for Fig.2, Fig.3, Fig.4, Tab.1, Tab.2 and Tab.3 in our original paper. All source code for conducting the experiments will be made publicly available as soon as our paper is accepted.

### A.22.1 TRAINING DETAILS FOR AUX-NET

**For vision experiments**, we follow the training setting and hyperparameters provided in DeiT (Touvron et al. (2021)) and LeTs (Xia et al. (2024c)). We configure the learngene module with 6 heads (head dimension is 64) and 8 layers, and transform it to construct Aux-Net with 16 heads and 24 layers (Aux-H16-L24) based on DeiT. We train the Aux-H16-L24 for 300 epochs with 5 warm-up epochs on ImageNet-1K (Deng et al. (2009)). We choose 304M ViT-Large (Dosovitskiy et al. (2021)) as the Ans-Net to employ soft distillation (Hinton et al. (2015)). Overall, the total training loss for the auxiliary model, $\mathcal{L}_{\text{total}}$, is a composite objective comprising three key components: the standard classification loss $\mathcal{L}_{\text{cls}}$, a knowledge distillation loss $\mathcal{L}_{\text{distill}}$, and our proposed orthogonality regularization loss $\mathcal{L}_{\text{ortho}}$. The final objective is defined as:

$$\mathcal{L}_{\text{total}} = (1 - \alpha)\mathcal{L}_{\text{cls}} + \alpha\mathcal{L}_{\text{distill}} + \beta\mathcal{L}_{\text{ortho}}, \tag{9}$$

where $\mathcal{L}_{\text{cls}}$ is the cross-entropy loss, and $\mathcal{L}_{\text{distill}}$ measures the discrepancy between the student and teacher models' outputs. The orthogonality term,

$$\mathcal{L}_{\text{ortho}} = \sum_k \left( \|\boldsymbol{U}^{(k)\top}\boldsymbol{U}^{(k)} - \boldsymbol{I}\|_F^2 + \|\boldsymbol{V}^{(k)}\boldsymbol{V}^{(k)\top} - \boldsymbol{I}\|_F^2 \right), \tag{10}$$

encourages the matrices $\boldsymbol{U}^{(k)}$ and $\boldsymbol{V}^{(k)}$ to maintain orthogonality. The hyperparameters $\alpha$ and $\beta$ control the trade-off between these different objectives. $\alpha$ is set to 1.0 and $\beta$ is set to $1e - 3$. Specifically, we use the AdamW (Loshchilov & Hutter (2018)) optimizer with weight decay 0.05 and a cosine scheduler, where batch size is set to 128. $H^{(0)}$ is set to 544 and $H^{(T)}$ is set to 136.

**For language experiments**, we follow the training setting and hyperparameters provided in (Tan & Bansal (2020); Wang et al. (2023a)) for BERT-based Aux-Net (Devlin et al. (2019)). For BERT, the learngene module is configured with 6 heads and 8 layers, and transformed to construct an Aux-Net with 16 heads and 24 layers. We train Aux-H16-L24 for 100K steps with 2K warm-up steps on English Wikipedia corpus. We choose BERT-Large(335M) (Devlin et al. (2019)) as the Ans-Net to employ soft distillation. Specifically, we use the AdamW (Loshchilov & Hutter (2018)) optimizer with weight decay 0.1 and a linear scheduler, where batch size for each GPU is set to 128, the trade-off for distillation and orthogonality is set to 0.5 and $1e - 3$. $H^{(0)}$ is set to 544 and $H^{(T)}$ is set to 136. For GPT2, the learngene module has 6 heads and 12 layers, and is transformed to construct an Aux-Net with 25 heads and 48 layers, with GPT2-XLarge (1.5B) (Radford et al. (2019)) as the Ans-Net. For GPT2-based Aux-Net (Radford et al. (2019)), we follow the training setting and hyperparameters provided in (Karpathy (2022)). We train Aux-H25-L48 for 50K steps with 1K warm-up steps on OpenWebText dataset (Gao et al. (2020); Gokaslan et al. (2019)). We choose GPT2-XLarge (Radford et al. (2019)) as the Ans-Net to employ soft distillation. Specifically, we use the AdamW (Loshchilov & Hutter (2018)) optimizer with weight decay 0.1 and a cosine scheduler, where batch size for each GPU is set to 12, the trade-off for distillation and orthogonality is set to 0.5 and $1e - 3$. $H^{(0)}$ is set to 800 and $H^{(T)}$ is set to 400. All models are implemented by PyTorch (Paszke et al. (2019)), and trained on NVIDIA H100 GPUs, NVIDIA A800 GPUs and NVIDIA RTX 3090 GPUs. GPU hours are approximately measured based on H100 GPUs or A800 GPUs and have been appropriately converted.

### A.22.2 EXPERIMENTAL DETAILS FOR FIG.2 IN OUR ORIGINAL PAPER

We report the numerical results of Fig.2 in Tab.4. For Scratch, we train all the Des-Nets for 100 epochs (more epochs for larger models) with 5 warm-up epochs on ImageNet-1K (Deng et al. (2009)) with timm default initialization (Paszke et al. (2019)), where we use the AdamW (Loshchilov & Hutter (2018)) optimizer with weight decay 0.05, batch size 128 and a cosine scheduler following (Touvron et al. (2021)). For LeTs (Xia et al. (2024c)), to keep fair comparison, we also configure the learngene module with 6 heads (head dimension is 64) and 8 layers, and transform it with *dense* transformation matrices to construct Aux-H16-L24. The training setting of Aux-H16-L24 for LeTs and ALT is same.

For LeTs and ALT, we initialize Des-Nets with continuous selection and sequential selection strategy, and fine-tune them for 5 epochs. We use the AdamW optimizer with weight decay 0.05, batch size 128 and a cosine scheduler for all Des-Nets. We also inherit the parameters of patch projection and classification head to initialize Des-Nets. As shown in and Tab.4, ALT consistently outperforms Scratch after 5 epoch tuning (in some cases after 1 epoch), while achieving around $2\times$ reduction in training costs for 19 models ($\sim 947$ GPU hours *vs.* $\sim 472$ GPU hours). When trained from scratch on ImageNet-1K, the Des-H16-L24 underperforms relative to smaller models, likely due to the limited number of training samples in ImageNet-1K.

### A.22.3 EXPERIMENTAL DETAILS FOR FIG.3 IN OUR ORIGINAL PAPER

**Calculation for Number of Inheritable Parameters in ALT.** A key metric for evaluating the efficiency of our ALT is the total number of *inheritable parameters*. It is important to clarify how this quantity is calculated in the context of our dynamic component adaptation. Our SVD-based parameterization decomposes a transformation matrix $\boldsymbol{F}^{(k)}$ into a series of learnable components. Our dynamic adaptation mechanism discards unessential components by setting their corresponding learnable values $\sigma_i^{(k)}$ to zero. From a mathematical and practical standpoint, any component for which $\sigma_i^{(k)} = 0$ makes no contribution to the final transformation matrix, as its product with the corresponding learnable vectors will always be a zero matrix. Consequently, for the purpose of storage, inheritance, and deployment, these components—including the zero-valued $\sigma_i^{(k)}$ and its associated vectors—can be entirely discarded without any loss of information. Therefore, when we report the final number of inheritable parameters, we **only include the parameters of the active components** (*i.e.*, those for which the final adapted $\sigma_i^{(k)} \neq 0$). This calculation accurately reflects the true storage and memory footprint of the compact parameter set that is inherited by the Des-Nets, and it is consistent with standard practices for parameter counting in network pruning.

In Fig.3a, we compare the inheritable parameters between Learngene methods on vision domain. Grad-LG (Wang et al. (2022)) selects the last 3 layers from the 12-layer Ans-Net. TLEG (Xia et al. (2024b)) linearly expands 2 learngene layers to construct the 12-layer Aux-Net. SWS (Xia et al. (2024a)) expands 5 learngene layers to construct the 16-layer Aux-Net. Since TLEG and SWS both use LeViT-384(39M) (Graham et al. (2021)) as their Ans-Net, we base their "Ans-Net parameters" on the size of their corresponding Aux-Nets, ensuring a fair comparison. LeTs (Xia et al. (2024c)) transfers substantially larger dense transformation parameters. In contrast, ALT significantly reduces this overhead and inherits about 15M parameters, which achieves a $\mathbf{20\times}$ reduction over the 304M ViT-Large Ans-Net.

For all methods and all downstream image classification datasets in Fig.3b-3i, we fine-tune (train) the Des-Nets whose backbones are Des-H6-L12 or Des-H12-L12 for 300 epochs. "Pre-Fin" pre-trains the Des-Nets on ImageNet-1K for 100 epochs and then fine-tunes them on downstream datasets. We use the AdamW (Loshchilov & Hutter (2018)) optimizer with batch size 128 or 256, and a cosine scheduler. For Cars-196 (Krause et al. (2013)), both Scratch and Grad-LG fail to converge, likely due to the limited number of training samples and unsatisfactory initialization quality.

For all semantic segmentation datasets in Fig.3j-3o, we follow the training setting of (Strudel et al. (2021)). We fine-tune the Des-Nets on ADE20K (Zhou et al. (2019)), Pascal Context (Mottaghi et al. (2014)) and Cityscapes (Cordts et al. (2016)) for 64, 256 and 216 epochs, respectively. For object detection dataset COCO2017 (Lin et al. (2014)) in Fig.3p, we follow the training setting of (Fang et al. (2021)). We fine-tune the initialized Des-Nets on COCO2017 (Lin et al. (2014)) for 100 epochs.

In Fig.3q, we present the comparison between ALT and knowledge distillation method (Soft-distillation), more discussions are presented in Sec.A.10. In Fig.3r, IMwLM (Xu et al. (2024)) uses one 300ep-pretrained model with 12 heads and 18 layers (129.1M) to initialize models of different sizes. Specifically, we use the consecutive selection strategy and select the first $N$ layers to initialize target models as introduced in IMwLM (Xu et al. (2024)), where $N$ represents the layer number of target models. In Fig.3s, for Matformer, we refer to the results presented in Fig.4(a) of their original paper (Devvrit et al. (2023)). In Tab.8, we present the comparison between ALT and LeTs under the linear-probing protocol.

### A.22.4 EXPERIMENTAL DETAILS FOR FIG.4 IN OUR ORIGINAL PAPER

In Fig.4a, we train and inherit about 39M parameters to initialize BERT-based Des-Nets. For Scratch, we train Des-H8-L12 for 50K steps with 1K warm-up steps on English Wikipedia corpus with timm default initialization (Paszke et al. (2019)) following the protocols of (Tan & Bansal (2020); Wang et al. (2023a)), where we use the AdamW (Loshchilov & Hutter (2018)) optimizer with weight decay 0.1, batch size 128 and a linear scheduler. ALT achieves around $2\times$ reduction in training steps compared to training from scratch with the Des-H8-L12 (54.0M).

In Fig.4b, we train and inherit about 62M parameters to initialize GPT2-based Des-H16-L24 (354.3M). For Scratch, we train Des-H16-L24 for 100K steps with 20 warm-up steps on the OpenWebText datasetwith timm default initialization (Paszke et al. (2019)) following the protocols of (Karpathy (2022); Gao et al. (2020); Gokaslan et al. (2019)), where we use the AdamW (Loshchilov & Hutter (2018)) optimizer with weight decay 0.1, batch size 12 and a cosine scheduler. ALT achieves around $2\times$ reduction in training steps compared to training from scratch with the Des-H16-L24 (354.3M).

### A.22.5 EXPERIMENTAL DETAILS FOR TAB.1 IN OUR ORIGINAL PAPER

For all GLUE and SQuADv1.1 experiments, we follow the standard training and evaluation protocols established by the HuggingFace Transformers library Wolf et al. (2019). The specific datasets and their corresponding evaluation metrics are summarized in Tab.11. Specifically, the standard pretrain-finetune (Pre-Fin) baseline involved pretraining the Des-H8-L12 and Des-H12-L12 model for 50K steps on English Wikipedia corpus, followed by fine-tuning on each downstream dataset. For ALT, we initialized the Des-H8-L12 and Des-H12-L12 model with ALT and then trained it for 50K steps on English Wikipedia corpus before fine-tuning. As shown in Tab.1, ALT matches or outperforms the standard pretraining and finetuning protocol across all tasks. The above results demonstrate that leveraging ALT for initialization provides a substantially better starting point, which not only accelerates the pre-training process but also boosts the final performance on downstream tasks.

### A.22.6 EXPERIMENTAL DETAILS FOR TAB.2 IN OUR ORIGINAL PAPER

To rigorously evaluate and dissect the contributions of our core methodological innovations, we conducted a comprehensive ablation study. This study was designed to systematically answer three key questions: (1) Is dynamic component adaptation necessary? (2) Is our proposed Gated Importance Score (GIS) superior to simpler alternatives? (3) How do our FGA and HCA strategies perform? The results are presented in Tab.2, and the analysis is twofold, focusing first on the performance of intermediate auxiliary model (Aux-Net) and then on the performance of final initialized descendant models (Des-Nets).

**Experimental Setup and Baselines.** All ablation experiments were conducted on the ImageNet-1K dataset. To isolate the impact of each component, we established two primary baselines:

- **ALT(Fixed):** A static, non-adaptive baseline. In this configuration, each transformation matrix is assigned a fixed, predefined number of active components (*i.e.*, a fixed rank) which remains unchanged throughout the entire 300-epoch training of the Aux-Net. This baseline serves to quantify the benefits of the dynamic adaptation process itself.

- **ALT(Add):** A dynamic baseline that replaces our core GIS metric. This version implements the same dynamic adaptation framework as our full method but utilizes the simpler *additive* importance score proposed in AdaLoRA (Zhang et al. (2023)) to guide component selection. This baseline allows for a direct, fair comparison to evaluate the effectiveness of our GIS formulation.

Our proposed methods, **ALT(GIS)** and **ALT(HCA)**, utilize the full GIS metric, employing the Flat Global Adaptation and Hierarchical Component Adaptation strategies, respectively.

**Analysis of Auxiliary Model (Aux-Net) Performance.** The first stage of our analysis focuses on the direct performance of the Aux-Net after its 100-epoch training phase, as this reflects the quality of the learned inheritable parameters before they are used for initialization. The results in Tab.2 (left half) reveal a clear hierarchy of performance. First, the performance gap between ALT(Add) and the static ALT(Fixed) baseline unequivocally demonstrates the importance of *dynamically* adapting the structure of the transformation matrices during training. Second, and more critically, our GIS-based

approaches (both FGA and HCA variants) consistently outperform ALT(Add). The training hours of ALT(GIS) and ALT(HCA) is similar (the gap is less than 0.5h). This validates the superiority of our proposed **Gated Importance Score**, suggesting it provides a more accurate and effective signal for identifying essential components compared to the simpler additive metric.

**Analysis of Final Descendant Model (Des-Net) Performance.** The ultimate measure of success is the performance of the descendant models initialized by the inheritable parameters. The right half of Tab.2 presents the performance of various Des-Nets after they have been initialized and fine-tuned. The results show that our full approach, ALT, consistently and significantly outperforms the ALT(Add/300) baseline across all Des-Net configurations.

This superiority provides the most compelling evidence for our ALT. It demonstrates that the benefits observed at the Aux-Net stage—derived from the combination of our superior GIS metric and sophisticated adaptation strategies like HCA—translate directly into a higher-quality set of inheritable parameters. These parameters, in turn, provide a substantially better initialization point for downstream models, ultimately leading to improved final performance.

### A.22.7 EXPERIMENTAL DETAILS FOR TAB.3 IN OUR ORIGINAL PAPER

In Tab.3, we evaluate ALT using various parameter selection strategies under 5 epoch tuning of Des-H8-L12 following LeTs. Specifically, we consider several magnitude-wise selection strategies including $L_2$-norm $\uparrow$, $L_2$-norm $\downarrow$, where $L_2$-norm $\uparrow$ means selecting rows/columns from transformation matrices whose $L_2$-norm ranks top, and so on. We also compare different learngene group selection schemes. As an illustrative example, we partition eight learngene matrices into four groups of two layers each. The sequence "$1, 1, 1, 2, 2, 2, 3, 3, 3, 4, 4, 4$" indicates that group $1, 2, 3$, and $4$ are each applied three times in the initialization process. Moreover, we ablate each Transformer module during ALT initialization for Des-H6-L12 (5 epoch tuning): Multi-Head Self-Attention (MSA), Multi-Layer Perceptron (MLP), Layer Normalization (LN). Tab.3 reveals that omitting MSA or MLP, especially MLP, causes severe performance degradation, indicating that initializing all modules is necessary.

### A.23 COMPARISON WITH ALTERNATIVE MODEL INITIALIZATION PARADIGMS

This section contextualizes our ALT approach by comparing it with other major paradigms for obtaining and initializing neural network models: downloading pretrained online models, meta-learning, and existing methods within the Learngene framework itself.

**ALT versus Pretrained Online Models.** A common practice for obtaining models of various sizes is to download individual checkpoints from public repositories like HuggingFace (Wolf et al. (2019)) or timm (Paszke et al. (2019)). While convenient, this approach has notable drawbacks: each model represents a significant, independent pre-training cost, and users must download and store a multitude of separate checkpoints.

ALT offers a fundamentally more efficient and flexible paradigm. By learning a single, highly-compact set of inheritable parameters, ALT obviates the need for storing numerous large checkpoints. This compact set can then be used to initialize a wide spectrum of descendant models at a fine granularity, unconstrained by predefined architectures. This provides superior customization and portability, especially in resource-constrained environments or when rapid prototyping of different model sizes is required.

**ALT versus Meta-Learning.** Meta-learning, or learning-to-learn, also focuses on producing robust model initializations. However, meta-learning techniques typically optimize an initialization for a *specific target architecture* to facilitate rapid adaptation to a distribution of related tasks. The goal is fast task-adaptation, not architectural flexibility.

ALT, by contrast, addresses the orthogonal problem of initializing a *family of variable-sized model architectures* for a given task domain. It learns one compact parameter set and a transformation mechanism to apply this set across architectural configurations. This design directly supports the Learngene framework's core objective of achieving high inheritable-parameter efficiency across diverse model initialization scenarios, a goal seldom targeted by conventional meta-learning.

**ALT's Contribution within the Learngene Framework.** While prior methods within the Learngene framework have developed various parameter transformation strategies, their primary focus has often

been on the transformation mechanism itself, rather than on the efficiency of the inheritable parameter set. Consequently, these methods, while effective, often result in a set of inheritable parameters that is only moderately more compact than the original ancestry model. This limits the portability and broad applicability envisioned by the Learngene framework.

Our approach re-evaluates this paradigm through the lens of inheritable-parameter efficiency. By integrating a highly efficient SVD-based parameterization with our dynamic adaptation strategies (HCA and FGA), ALT achieves a dramatic reduction in the size of the inheritable parameter set, unlocking substantial gains in both portability and flexibility. We believe that prioritizing the compactness and efficiency of the inheritable "gene" itself, as demonstrated by ALT, represents a valuable and promising direction for future Learngene research.

### A.24    LIMITATIONS AND FUTURE WORK

While our work demonstrates the significant potential of the ALT framework, we acknowledge several limitations that also present exciting avenues for future research.

First, the hyperparameter configurations for both the auxiliary model (Aux-Net) training and the subsequent descendant model (Des-Net) fine-tuning were not exhaustively optimized. Our current schedules were selected based on standard practices, but we believe that a more systematic hyperparameter search, potentially using automated methods, could unlock further gains in both performance and efficiency. This represents a straightforward direction for future practical improvements.

Second, our current implementation of ALT relies solely on soft distillation to transfer knowledge from the ancestry model. While effective, this approach may not capture the full richness of the Ans-Net's internal knowledge. A promising line of future work is to explore more sophisticated distillation objectives. Incorporating techniques such as intermediate feature matching or attention transfer could potentially lead to a higher-fidelity inheritable parameter set, further enhancing the quality of the final initialization.

Finally, the empirical validation in this paper has focused on vision and standard-scale language modeling tasks. A natural and important next step is to scale and extend the ALT framework to more complex, multi-modal domains and to the latest generation of large-scale language models (LLMs), such as the LLaMA series (Touvron et al. (2023)). Successfully applying ALT in these challenging scenarios would further validate its generality and practical utility as a paradigm for efficient model initialization across a wide spectrum of model scales and modalities.

### A.25    BROADER IMPACTS STATEMENT

Our work focuses on the fundamental challenge of improving the parameter efficiency of model initialization. The primary positive societal impact lies in democratizing access to large-scale models by lowering computational and storage barriers, which can foster broader innovation and lead to reduced energy consumption in AI development. We have used only publicly available, standard academic datasets and do not foresee direct negative societal impacts, as our method is a general-purpose efficiency enhancement.

