# OpenReview forum: "ALT: Adaptive Low‑Rank Transformation for Learngene-based Transformer Initialization"
_ICLR.cc/2026/Conference — Submitted to ICLR 2026_

### Official Review · Reviewer_MpXW · 2025-10-23

**Soundness:** 2
**Presentation:** 2
**Contribution:** 2
**Rating:** 2
**Confidence:** 5

**Summary:**

The paper proposes ALT (Adaptive Low-rank Transformation) for the Learngene paradigm, which aims to learn a compact, reusable parameter set (“learngene”) from a large, well‑trained ancestry model and then transform it to initialize many descendant models of varying widths/depths. The key observation is that, in prior Learngene approaches (e.g., LeTs), the width transformation matrices dominate the inherited parameter budget. ALT replaces dense transformation matrices with explicit SVD parameterizations F=U Σ VF=U\,\Sigma\,VF=UΣV and introduces a Gated Importance Score (GIS) to rank SVD components by combining a sensitivity-based score of the singular value with the averaged sensitivities of the corresponding singular vectors. Two pruning/adaptation strategies are proposed: Flat Global Adaptation (FGA) (global top‑H across all matrices) and Hierarchical Component Adaptation (HCA) (inter‑matrix apportionment + intra‑matrix selection). The method is trained in two stages: (1) train an Aux‑Net (with SVD-parameterized transforms) via distillation from a large teacher; (2) inherit the compact learngene + transforms to initialize many descendants. Experiments on ImageNet‑1K (and downstream classification, segmentation, detection) and on BERT/GPT‑2 pretraining + GLUE/SQuAD report 20–25× fewer inherited parameters than the teacher while maintaining or improving performance vs. baselines, and reduced steps to target loss vs. training from scratch.

**Strengths:**

Clear bottleneck diagnosis & targeted design. The paper carefully identifies that most inherited parameters in LeTs are in the transformation matrices (≈38M out of ≈53M in a representative setup), motivating a low‑rank reparameterization; the SVD plots show sharp spectral decay and module/layer‑dependent patterns, supporting adaptive rank selection.

Methodologically coherent. The explicit SVD parameterization, orthogonality regularization, and per‑component GIS (linking singular value “energy” and vector “orientation”) form a consistent mechanism to enable dynamic pruning via FGA/HCA; the HCA capacity‑aware apportionment is spelled out with pseudo‑code.

Breadth of empirical scope. Results span classification (ImageNet‑1K + CIFAR/Food‑101/Cars‑196), segmentation (ADE20K, Pascal Context, Cityscapes), detection (COCO2017), and language (BERT on Wikipedia, GPT‑2 on OpenWebText, GLUE/SQuAD). Baselines include Scratch, Pre‑Fin, Grad‑LG, TLEG, SWS, LeTs, and comparisons vs. IMwLM/MatFormer and soft distillation.

Practical efficiency narrative. For vision, inheritable parameters drop to ~15M (≈20× vs. ViT‑Large 304M). For language, ~39M for BERT (8× vs. BERT‑Large; 20× excluding embeddings) and ~62M for GPT‑2 (25× vs. 1.5B; 60× excluding embeddings). The paper also argues the “learn once, initialize many” amortization, showing a break‑even example over nine models.

Ablations & module insights. The paper ablates ALT(Fixed) vs ALT(Add) (AdaLoRA-style) vs ALT(GIS) with FGA/HCA; it also probes selection strategies and shows that omitting MLP initialization devastates performance, underscoring where initialization matters most.

**Weaknesses:**

Novelty relative to adaptive low‑rank PEFT may be overstated. While the goal (portable initialization across architectures) differs from PEFT, the mechanism—SVD‑parameterized low‑rank components with a learned importance score and progressive pruning—is close in spirit to methods like AdaLoRA/PiSSA/MiLoRA. The paper positions GIS as “multiplicative” vs. additive metrics, but lacks deeper justification (theory or controlled studies) showing that multiplicativity per se drives the gains beyond careful tuning. The ablation (ALT(Add) vs ALT(GIS)) is helpful but could be confounded by hyperparameters and schedules.

GIS definition lacks clarity and could be brittle. The sensitivity term uses EMA of a per‑parameter importance and an uncertainty estimate; exact computation, stability, and scaling are not sufficiently detailed in the main paper (e.g., normalization across heterogeneous U/V shapes, invariance to re‑scalings of U/V/Σ, choice of wU,wVw_U,w_VwU​,wV​, and the impact of orthogonality losses). The notation reuses I(⋅)I(\cdot)I(⋅) for both indicator and sensitivity in different places, which hampers clarity.

Data/teacher dependency undermines portability claims. Stage‑1 requires access to (a proxy of) the original pretraining distribution and a large teacher model for distillation. The authors acknowledge difficulty scaling to modern LLMs due to closed pretraining corpora. This dependence could significantly limit real‑world adoption in privacy‑ or license‑constrained settings, weakening the “universal portable initializer” narrative.

Compute story is mixed. ALT’s Aux‑Net costs ≈445 GPU‑hours vs. 390 for LeTs—higher one‑time cost. The amortization study is thin: for nine students the total is only modestly better than soft distillation; for 1–2 students, ALT may be costlier overall. A formal break‑even analysis (as a function of #students, sizes, and tuning epochs) is missing.

Fairness of comparisons.
Some downstream comparisons fine‑tune descendants for 300 epochs, while ALT emphasizes 5–15‑epoch efficiency in other places; the paper needs a consistent budget comparison per setting (same epochs/augs/opt).
Pre‑Fin uses specific schedules/augmentations; whether ALT’s descendants also use equally strong training recipes is not entirely transparent.
For language, the GPT‑2 result is shown as loss curves; consolidated final perplexity numbers and multi‑seed statistics are not reported.

Limited statistical rigor. There is no discussion of variance across seeds, confidence intervals, or significance tests. Given many 0.2–0.5% improvements, error bars are essential to support claims.
Potential selection bias in width initialization. The continuous selection policy “take the first n rows/cols” of reconstructed FFF could bias performance (e.g., if learned structure is not aligned with the first coordinates). The paper notes several selection strategies but still defaults to a fixed continuous policy during initialization; more robust, learned selection or ordering may further change the relative advantage.

Counting inherited parameters excludes zeroed components post‑hoc. While reasonable for storage, it obscures the training-time memory/compute footprint (since dormant components were still carried and updated before pruning). The method’s true training efficiency vs. dense baselines is therefore less clear without detailed FLOPs/memory traces during Stage‑1.

Theoretical underpinnings are light. The paper relies on empirical singular value decay to motivate low rank. There is no analysis of when GIS‑guided pruning preserves initialization quality, nor bounds on the degradation induced by component removal across modules/layers.

Scope of LLM validation is narrow. Results are limited to GPT‑2‑XL and BERT‑Large on public corpora; no evidence is provided on more modern architectures (e.g., decoder‑only with RMSNorm/rotary attention) or instruction‑tuned settings. The paper itself notes data access as a barrier, but this weakens the generality claim.

Clarity and presentation issues.
Some figures (e.g., long SVD spectra with indices up to 1.5k) lack explicit matrix shapes, making it hard to interpret rank percentages across modules.
Notation overload (e.g., III ) and occasional grammar/formatting issues slow down comprehension.

**Questions:**

GIS specifics & stability. How exactly are the sensitivity and uncertainty terms computed (batching, normalization, temperature, EMA decay)? How sensitive are results to wU,wVw_U,w_VwU​,wV​ and these EMAs? Please provide ablations and a discussion of invariance to re‑scaling U,V,ΣU,V,\SigmaU,V,Σ.

Orthogonality regularization. What is the overhead (time/memory) of the orthogonality loss (Eq. 5) across all matrices? Does performance degrade if you stop enforcing orthogonality partway through training (or use spectral parameterization with implicit orthogonality)?

Break‑even analysis. For a given teacher, dataset, and hardware, how many descendant models (and of what sizes) are needed before ALT’s total compute is lower than (a) scratch, (b) soft distillation, and (c) LeTs? Please include a table with explicit assumptions.

Seed variance. What are mean ± std across multiple seeds for key tables/figures (e.g., Fig. 2, Tab. 4–7, Tab. 1)? Several gains are small; variance is needed to judge significance.

Selection policy. You default to continuous selection of rows/columns when reconstructing FFF for a given width. Have you tried re‑ordering bases (e.g., via learned permutations or sensitivity‑aware submatrix selection) and does this change the outcomes or parameter counts?

Training‑time footprint. Please report training‑time peak memory/FLOPs for Stage‑1, and the evolution of active components H(t)H(t)H(t) vs. wall clock. How do these compare to LeTs and to a dense Aux‑Net without SVD?

Language results. Can you report final perplexities (not just curves) for GPT‑2 experiments and provide task‑level GLUE/SQuAD variances? Also, how does ALT fare on modern LLM blocks (e.g., RoPE, SwiGLU, RMSNorm) or instruction‑tuned checkpoints?

Robustness. How does ALT behave under distribution shift or low‑data finetuning scenarios relative to Pre‑Fin/IMwLM? Any evidence on calibration or stability at very small budgets?

Ablations on GIS design. Beyond additive vs. multiplicative, have you compared (i) using only σi\sigma_iσi​, (ii) only vector scores, (iii) using quantile‑based aggregation in HCA, (iv) different decay schedules for H(t)H(t)H(t)? A compact grid would clarify which pieces matter most.

Counting conventions. You exclude zeroed components when reporting “inheritable parameters.” Please also report (i) the pre‑pruning parameter count, (ii) the average active components over training, and (iii) the runtime storage required to ship a reusable initializer (including any metadata needed to reconstruct per‑matrix selections).

---

> ### Author Response · Authors · 2025-11-21
> **Response to Reviewer #MpXW(1/10)**
>
> **Q1: On GIS Specifics, Hyperparameter Sensitivity, and Scale Invariance.**
>
> Thank you for these questions. Here, we provide a detailed explanation of its computation, new ablation studies on its key hyperparameters, and a discussion on its scaling properties.
>
> **(1) Computation of Sensitivity and Uncertainty.**
> We referenced PLATON[1] for the single parameter importance function $s(\cdot)$ but did not elaborate on its computation. The function $s(\cdot)$ is designed to be an importance metric that considers both the sensitivity of a parameter and the uncertainty of that sensitivity estimation. The process is as follows:
>
>
>
> **1) Sensitivity ($I^{(t)}$):** At each training iteration $t$, we first compute the sensitivity for each parameter $\theta_j$ as the magnitude of the gradient-weight product: $I_j^{(t)} = |\theta_j^{(t)} \cdot \nabla_{\theta_j} \mathcal{L}(\boldsymbol{\theta}^{(t)})|$.
>
> **2) Smoothed Sensitivity ($\overline{I}^{(t)}$):** We compute a smoothed sensitivity $\overline{I}_j^{(t)}$ using Exponential Moving Average (EMA): $\overline{I}_j^{(t)} = \beta_1 \overline{I}_j^{(t-1)} + (1-\beta_1) I_j^{(t)}$.
>
> **3) Uncertainty Estimation ($U^{(t)}$):** We quantify the uncertainty of the sensitivity estimation by its local temporal variation: $U_j^{(t)} = |I_j^{(t)} - \overline{I}_j^{(t-1)}|$.
>
> **4) Smoothed Uncertainty ($\overline{U}^{(t)}$):** This uncertainty measure is also smoothed via another EMA with a decay rate $\beta_2$: $\overline{U}_j^{(t)} = \beta_2 \overline{U}_j^{(t-1)} + (1-\beta_2) U_j^{(t)}$.
>
> **5) Final Importance Function $s(\cdot)$:** The final base importance for a single parameter is the product of the smoothed sensitivity and the smoothed uncertainty: $s(\theta_j^{(t)}) = \overline{I}_j^{(t)} \cdot \overline{U}_j^{(t)}$.
>
> **(2) Hyperparameter Sensitivity Analysis.**
> We conduct new ablation studies to analyze the sensitivity of ALT to ($w_U$, $w_V$) and ($\beta_1$, $\beta_2$). We train Aux-Nets for 100 epochs on ImageNet-1K and then evaluate the 5-epoch tuning performance of an initialized Des-H8-L12 model.
>
> **Table:** Sensitivity to ($w_U$, $w_V$) and ($\beta_1$, $\beta_2$).
>
> | ($w_U$, $w_V$) | (0.25, 0.75) | (0.5, 0.5) | ($\beta_1$, $\beta_2$) | (0.85, 0.85) | (0.95, 0.95) |
> | :---: | :---: | :---: | :---: | :---: | :---: |
> | Top-1 Acc. (%) | 75.9 | 76.0 | Top-1 Acc. (%) | 76.0 | 75.9 |
>
> We adopt different weighting trade-offs between the importance of learnable vectors $\boldsymbol{U}$ and $\boldsymbol{V}$ in the GIS formulation. For simplicity, we fix $w_U+w_V=1$. The results in the above table show that averagely considering both vector components yields better performance. We also evaluate the impact of the EMA decay rates and find the performance is stable across a range of decay rates. The above results show that **ALT is not overly sensitive to these two hyperparameters**.
>
> **(3) Discussion on Invariance to Re-scaling.**
> We explain that in our actual training, the scaling ambiguity is implicitly but mitigated by key mechanisms: Our training objective includes a regularization term $\mathcal{L}_{\text{ortho}}$ that explicitly penalizes deviations from orthogonality. This constraint encourages the column/row vectors of $\boldsymbol{U}$ and $\boldsymbol{V}$ to have a norm close to 1, which helps to mitigate arbitrary re-scaling.
>
> **References**
>
> [1] ``PLATON: Pruning Large Transformer Models with Upper Confidence Bound of Weight Importance.'', ICML 2022.

---

> ### Author Response · Authors · 2025-11-21
> **Response to Reviewer #MpXW(2/10)**
>
> **Q2: On the Overhead and Necessity of Orthogonality Regularization.**
>
> Thank you for questions regarding the cost and the necessity of orthogonality regularization term.
>
> **(1) Overhead Analysis of the Orthogonality Loss.**
> The computational overhead of the orthogonality loss, $\mathcal{L}_{\text{ortho}}$, is minimal by design. This is because the core operations of this term---matrix multiplications ($\boldsymbol{U}^\top\boldsymbol{U}$, $\boldsymbol{V}\boldsymbol{V}^\top$) and norm calculations---are performed on the small matrices ($\boldsymbol{U}, \boldsymbol{V}$). The dimensions of these matrices are determined by the rank $r$, which is orders of magnitude smaller than the common model dimensions. **Consequently, the cost of this term is negligible compared to the forward and backward passes of the entire Aux-Net.**
>
> **(2) Analysis of the Orthogonality Constraint.**
> To investigate the importance of enforcing orthogonality, we conduct an ablation study comparing our ALT against a version where the regularization is disabled ($\beta=0$). As shown in the Tab.13 of the revision, we observe that removing the orthogonality constraint degrades performance, confirming its necessity for SVD-based parameterization. Moreover, the model's performance remains robust across a range of reasonable values for $\beta$ (1e-3 and 1e-2), indicating that our method is not overly sensitive to this hyperparameter. We attribute this robustness to the fact that the orthogonality regularization is applied to the transformation matrices. **Since these matrices constitute only a small fraction of the total inheritable parameters, variations in the regularization strength do not significantly perturb the overall learning process of the inheritable parameters.**

---

> ### Author Response · Authors · 2025-11-21
> **Response to Reviewer #MpXW(3/10)**
>
> **Q3: On the Break-even Analysis of ALT's Computational Cost.**
>
> **Table:** Break-even analysis of ALT's total compute cost versus baselines on ImageNet-1K. Total costs are shown for the specific number of models (N) compared in our experiments.
>
> | Method | Upfront (h) | Avg. FT/Model (h) | Break-even (N) | Total Cost (N models) |
> | :--- | :---: | :---: | :---: | :---: |
> | *(a) Comparison vs. Training from Scratch (Scratch)* | | | | |
> | Scratch | 0 | ~63.5 | ~7 | N=8: ~508 h |
> | **ALT (ours)** | **~445** | **~1.4** | | **N=8: ~456 h** |
> | *(b) Comparison vs. Soft Distillation* | | | | |
> | Soft Distillation | 0 | ~61.0 | ~8 | N=9: ~549 h |
> | **ALT (ours)** | **~445** | **~4.8** | | **N=9: ~488 h** |
>
>
>
> Thank you for this question. ALT's efficiency stems from its **``learn once, initialize many''** paradigm, where a one-time, upfront computational investment in training the Aux-Net is amortized over the initialization of multiple Des-Nets, each requiring only a lightweight fine-tuning schedule.
>
> The analysis is based on the following:
> **(a) Upfront Cost:** For ALT, this is the 445 GPU hours of training the inheritable parameters. For Scratch and Soft Distillation, this is zero.
> **(b) Fine-Tuning (FT) Cost:** This is the average time to train one Des-Net. Crucially, the fine-tuning schedule for ALT is adapted to the strength of the baseline. To achieve performance competitive with the standard Scratch baseline, a very lightweight 0-5 epoch fine-tuning suffices for most Des-Nets. To match the stronger Soft Distillation baseline, we modestly extended the fine-tuning up to 15 epochs for some Des-Nets.
>
> **(1) ALT vs. Training from Scratch.**
> Training a model from scratch (100-300 epoch) requires a substantial downstream cost (63.5h/model). In contrast, **ALT requires only a minimal fine-tuning effort (1.4h/model).** The break-even point is reached at approximately **7 models** ($445 / (63.5 - 1.4) \approx 7.2$). As our direct cost comparison for N=8 models (ranging from 29.2M to 171.6M) shows, ALT's total cost (456h) is already lower than that of training them from scratch (508h).
>
> **(2) ALT vs. Soft Distillation.**
> Similarly, each new model for Soft Distillation requires a full 100-epoch training schedule (61h/model). Even when we use a slightly longer fine-tuning schedule for ALT in this comparison, **its per-model cost (4.8h/model) remains an order of magnitude lower.** The break-even point is met at approximately **8 models** ($445 / (61.0 - 4.8) \approx 7.9$). The total cost for training 9 models (ranging from 22.1M to 253.9M) confirms this, with ALT being more efficient (488h vs. 549h). This clearly demonstrates that **ALT's ``learn once, initialize many'' paradigm is significantly more efficient than repeated, per-model distillation.**
>
> **(3) ALT vs. LeTs: A Trade-off between Compute and Parameter Efficiency.**
> As LeTs shares the same ``learn once'' paradigm, their downstream costs are identical. LeTs has a lower upfront computational cost (390h vs. 445h for the Aux-Net). Therefore, from a *purely computational cost perspective*, LeTs is slightly more efficient. **However, this comparison overlooks the central goal of our work: inheritable-parameter efficiency**. ALT incurs a marginal ~14% increase in one-time, upfront training compute, but in return, it delivers a **dramatic 3.5× reduction in the size of the inheritable parameter set** (15M vs. 53M). For real-world applications where storage and memory footprint are critical constraints (*e.g.*, edge devices), **this substantial gain in parameter efficiency is often far more valuable than the modest difference in one-time training cost.**

---

> ### Author Response · Authors · 2025-11-21
> **Response to Reviewer #MpXW(4/10)**
>
> **Q4: On the Analysis of Seed Variance.**
>
> Due to the significant computational cost of our experiments, we are unable to re-run all experiments with multiple seeds. However, to address your question, we prioritize and conduct a new set of experiments for the comparison: ALT versus the pretraining and finetuning baseline, to present the effectiveness of initialization with ALT. We report the mean and standard deviation on GLUE/SQuAD across 3 seeds for Des-H8-L12.
>
> **Table:** Mean and standard deviation on GLUE/SQuAD across 3 seeds for Des-H8-L12.
>
> | Task | Metric | Pre-Fin Baseline | ALT (ours) |
> | :---: | :---: | :---: | :---: |
> | MRPC | F1/Acc. | 87.53 $\pm$ 1.22 / 82.35 $\pm$ 1.47 | **89.06 $\pm$ 0.38 / 85.13 $\pm$ 0.38** |
> | STS-B | P./S. Corr. | 82.52 $\pm$ 0.60 / 82.19 $\pm$ 0.45 | **85.66 $\pm$ 0.92 / 85.36 $\pm$ 0.85** |
> | SQuADv1.1 | F1/EM | 80.00 $\pm$ 0.05 / 70.16 $\pm$ 0.43 | **80.97 $\pm$ 0.11 / 71.53 $\pm$ 0.34** |
>
> The results in the table above clearly show that, **when accounting for seed variance, our ALT method consistently and significantly outperforms Pre-Fin.** Specifically, the mean performance of ALT is substantially higher, and the confidence intervals (as indicated by the standard deviations) do not overlap. For instance, on MRPC, ALT achieves a mean accuracy of 85.13% compared to Pre-Fin's 82.35%, a gain of +2.78%. This provides strong evidence that the improvements by our ALT are significant.
>
> **Q5: On the Selection Policy for Des-Net Initialization.**
>
> Thank you for this question regarding the strategy for selecting sub-matrices during Des-Net initialization. First, we would like to clarify that the core novelty of our work lies in **learning** a highly-efficient inheritable parameter set (via GIS and our adaptation strategies), rather than in the subsequent step of **using** this set to initialize Des-Nets. For the latter, we adopt the established and effective strategies from LeTs to **ensure a focused and fair comparison of the inheritable parameters themselves**.
>
> You raise an excellent point about comparing our default strategy with alternatives like ''sensitivity-aware submatrix selection''. **We would like to respectfully point out that we have already conducted this analysis in our original paper in Sec.4.3, under ''Parameter Selection Strategies'' with results presented in Tab.3.** In that section, we evaluated several ``magnitude-wise selection strategies,'' which are conceptually identical to the ''sensitivity-aware selection'' suggested by you. Specifically, we tested selection based on the $L_2$-norm of the rows/columns of the transformation matrices (termed '$L_2$-norm $\uparrow$'), which re-ranks components based on their vector magnitude (a proxy for sensitivity) before selection. As reported in Tab.3, the results showed that these more complex, magnitude-wise selection strategies yield similar performance compared to our simpler, default continuous selection strategy (*e.g.*, 79.1% for continuous vs. 79.1% for '$L_2$-norm $\uparrow$' on Des-H8-L12). Given these existing results, we present that the default continuous selection is a well-justified choice.
>
> **Q6: On the Analysis of Training-Time Footprint.**
>
> Thank you for these questions regarding the detailed training-time resource consumption of our ALT. To address this, we provide a comprehensive comparison of peak memory and FLOPs for ALT versus LeTs, and the evolution of active components during Stage-1 Aux-Net training of ALT.
>
> **Table:** Comparison of peak memory and FLOPs for the Aux-Net between ALT and LeTs.
>
> | Method | Peak Memory (GB) | FLOPs (G) |
> | :---: | :---: | :---: |
> | LeTs | ~41.6 | ~922.4 |
> | **ALT (ours)** | ~41.2 | ~939.4 |
>
> **(1) Peak Memory and FLOPs Analysis.**
> For a fair comparison, both ALT and LeTs are configured to train the Aux-H16-L24 model on ImageNet-1K. We measure the per-GPU peak memory consumption and the FLOPs of the Aux-Net for both methods. We calculate the model's FLOPs using the torchprofile library. From the table above, the peak memory and FLOPs for ALT and LeTs are **nearly identical**.
>
> **(2) Evolution of Active Components vs. Wall-Clock Time.**
> To visualize our dynamic adaptation process, we plot the total number of active components, $H(t)$, as a function of the wall-clock training time. Fig.9(a) in the revision shows the number of active components vs. the wall time. Fig.9(b) in the revision illustrates how the number of active components decreases over the training process theoretically and experimentally. This demonstrates that **our adaptation mechanism operates as designed, progressively and stably refining the structure of the transformation matrices.**

---

> ### Author Response · Authors · 2025-11-21
> **Response to Reviewer #MpXW(5/10)**
>
> **Q7: On Fairness of Comparisons, Language Model Results and Applicability to Modern LLM components.**
>
> Thank you for these questions regarding the fairness of comparisons, details of our language model experiments and the forward-looking applicability of ALT.
>
> **(1) Fairness of Comparisons.**
> Regarding the comment **"Some downstream comparisons fine‑tune descendants for 300 epochs, while ALT emphasizes 5–15‑epoch efficiency in other places; the paper needs a consistent budget comparison per setting,"** we would like to provide a clarification. First, the **"300 epochs"** refers to the standard training setting for downstream image classification datasets; both ALT and the baseline Pre-Fin utilize this identical 300-epoch setting, under which ALT consistently achieves superior performance. Second, the **"5–15 epochs"** claim highlights our efficiency on ImageNet-1K fine-tuning, where ALT reaches convergence significantly faster than the **100 epochs** required for training from Scratch, thereby drastically reducing total costs. Adopting different epoch schedules for different datasets/tasks is reasonable and standard practice, and these details are explicitly documented in **Sec.A.22** of the paper. Regarding the concern that **"Pre‑Fin uses specific schedules/augmentations; whether ALT’s descendants also use equally strong training recipes is not entirely transparent,"** we confirm that both Pre-Fin and ALT employ **identical training settings** without any method-specific schedules or unfair augmentations, ensuring a strictly fair comparison.
>
> **(2) Final Perplexities.**
> For our GPT-2 experiments on OpenWebText, the final validation loss for the scratch baseline and ALT was 3.17, but ALT achieves around **2×** reduction in training steps compared to the scratch baseline.
>
> **(3) GLUE/SQuAD Variances.**
> We report our BERT experiments on some GLUE tasks and SQuADv1.1 under **3 different random seeds**. The table in the response to Q4 presents the mean and standard deviation for a representative subset of tasks. The results confirm that **when accounting for seed variance, ALT consistently provides better performance.**
>
> **(4) Applicability to Modern LLM components (RoPE, SwiGLU, RMSNorm).**
> Our ALT framework is fully compatible with modern LLM architectural components without requiring any modification to its core mechanisms. ALT operates on the linear matrices within the attention and FFN blocks. Modern components like RoPE, SwiGLU, and RMSNorm do not alter this fundamental structure:
> **RoPE (Rotary Position Embedding)** is applied *after* the query and key projection matrices have been computed and thus does not affect their parameterization.
> **SwiGLU** replaces the standard FFN with a different set of linear projections. ALT can be seamlessly applied to these new linear layers.
> **RMSNorm** is a variant of Layer Normalization. Since our ALT treats normalization layers as fixed parts of the Aux-Net, replacing LayerNorm with RMSNorm has no impact on ALT.
> Therefore, **ALT is inherently generalizable to these modern architectural designs.**
>
> **(5) Applicability to Instruction-Tuned Checkpoints.**
> You raise a point regarding instruction-tuned models. It is important to clarify that **ALT and instruction-tuning are orthogonal techniques that operate at different stages of a model's lifecycle.** ALT is an **initialization method** designed to improve the efficiency and effectiveness of the **pre-training** phase. An instruction-tuned checkpoint, by contrast, is the result of a **downstream fine-tuning** phase applied to an already-trained base model. One cannot use an initialization method to ``initialize'' an already fully trained model. However, the two techniques are **highly complementary**. The correct application would be a two-step process: (a) Use ALT to initialize and then pre-train a model. (b) Then, apply instruction-tuning to this ALT-trained model. We hypothesize that starting instruction-tuning from a better model produced efficiently by ALT, would lead to better final instruction-following capabilities.

---

> ### Author Response · Authors · 2025-11-21
> **Response to Reviewer #MpXW(6/10)**
>
> **Q8: On the Robustness under Low-Data Scenarios.**
>
> Thank you for these questions about the robustness of our ALT. To address them, we conduct new experiments focusing on low-data fine-tuning scenarios, and provide new analyses on calibration and stability.
>
> **Table:** Top-1 accuracy (%) of Des-H6-L12 on CIFAR-100 with 300-epoch fine-tuning under low-data fine-tuning settings. The performance under 30% data is averaged across 2 different seeds.
>
> | Training Data | Pre-Fin | ALT (ours) |
> | :---: | :---: | :---: |
> | 30% | 72.2 | **85.8** |
> | 50% | 77.5 | **87.8** |
> | 100% (Full) | 86.9 | **89.6** |
>
>
>
> **(1) Performance in Low-Data Fine-tuning Scenarios.**
> To assess ALT's performance in data-constrained settings, we conduct experiments on subsets of the CIFAR-100 training data, specifically using **30%** and **50%** of the data for 300-epoch fine-tuning. We compare our ALT-initialized model against the Pre-Fin (pre-training on ImageNet-1K + fine-tuning) baseline. The above table shows that **ALT's performance advantage is even more pronounced in low-data regimes**. For example, at 30% data, ALT outperforms Pre-Fin by 13.6%. This confirms that the rich prior knowledge inherited via ALT from the large Ans-Net substantially improves generalization when training data is scarce.
>
> **(2) Calibration and Stability at Small Data Budgets.**
> We further analyze the experiments on the 30% CIFAR-100 subset to evaluate calibration and stability, as requested.
>
> **Stability:**
> We run the 30% data experiment for both ALT and Pre-Fin using 2 different random seeds. The standard deviation of the Top-1 accuracy for ALT is $\pm$0.32%, which is higher than that for Pre-Fin ($\pm$0.04%). Despite the higher standard deviation, the mean performance of ALT is substantially higher, and the intervals (as indicated by the standard deviations) do not overlap.
>
> **Calibration:**
> We measure the Expected Calibration Error (ECE) [1] on the validation set. At 50% data, the ALT-initialized model achieves an ECE of 0.1373, which is better than the Pre-Fin baseline's ECE of 0.1571. This indicates that models initialized with ALT are not only more accurate but also produce more reliable confidence scores.
>
> In summary, these new results provide strong evidence that ALT offers superior robustness in challenging low-data scenarios.
>
> **References**
>
> [1] ``On Calibration of Modern Neural Networks.'', ICML 2017.

---

> ### Author Response · Authors · 2025-11-27
> **Response to Reviewer #MpXW(7/10)**
>
> **Q9: On Ablation Studies for GIS Design and Adaptation Strategy.**
>
> Following your guidance, we have conducted all the suggested experiments. We train Aux-Nets for 100 epochs on ImageNet-1K and then evaluate the 5-epoch tuning performance of an initialized Des-H8-L12 model. The results are summarized in the table below, as requested.
>
> **Table:** Ablation studies on GIS components, HCA aggregation, and the $H(t)$ decay schedule.
>
> | Experiment Setup | Configuration Detail | Top-1 Acc. (%) |
> | :---: | :---: | :---: |
> | *(1) GIS Component Analysis* | | |
> | | GIS (using only value score, $s(\sigma_i)$) | 75.4 |
> | | GIS (using only vector scores) | 75.5 |
> | | **Full GIS (ours)** | **76.0** |
> | *(2) HCA Aggregation Function* | | |
> | | HCA with Quantile (75%) Aggregation | 76.0 |
> | | **HCA with Mean Aggregation (ours)** | 76.0 |
> | *(3) Decay Schedule for $H(t)$* | | |
> | | Linear Decay Schedule | 75.9 |
> | | **Cubic Decay Schedule (ours)** | **76.0** |
>
> **(1) On GIS Components.**
> Our ablation study demonstrates that utilizing either the learnable singular value scores or the vector-based scores alone results in performance degradation. This empirical evidence confirms that both components are indispensable and complementary: the singular values ($\sigma_i$) encode the signal ''magnitude'' (importance), while the vectors capture the structural ``orientation''. Combining both signals enables the model to leverage a complete view of information, thereby maximizing downstream performance.
>
> **(2) On HCA Aggregation Strategies.**
> We conduct an ablation study to compare the standard ''mean'' aggregation against the ``75%-quantile'' strategy within the HCA module. The empirical results reveal that the quantile-based method yields performance comparable to the mean-based method, with negligible differences in downstream Des-Net performance. This observation suggests that our proposed **ALT is relatively insensitive to the specific choice of the aggregation function.** It demonstrates intrinsic robustness, capable of maintaining consistent effectiveness across different strategies without requiring meticulous hyper-parameter tuning.
>
> **(3) On the $H(t)$ Decay Schedule.**
> We conduct an ablation study to compare the proposed cubic decay schedule against a linear decay schedule. The empirical results show that the performance difference on the downstream Des-Net is negligible, which suggests that our proposed **ALT is insensitive to the specific choice of the decay function.** It further corroborates the method's stability, indicating that it can maintain consistent effectiveness across different scheduling strategies without the need for complex hyper-parameter tuning.

---

> ### Author Response · Authors · 2025-11-27
> **Response to Reviewer #MpXW(8/10)**
>
> **Q10: On Parameter Counting Conventions and Storage Footprint.**
>
> **Table:** Detailed breakdown of inheritable parameter counts and storage for ALT.
>
> | Metric | ALT (ours) |
> | :---: | :---: |
> | *(1) Pre-pruning Inheritable Parameters (at $t=0$)* | |
> | - Learngene Module | ~15 M |
> | - Initial Transformation Matrices | ~1.1 M (SVD-based, at $H^{(0)}$) |
> | Total Pre-pruning | ~16.1 M |
> | *(2) During Aux-Net Training* | |
> | Avg. Active Components | ~226 |
> | *(3) Final Initializer (Post-pruning, at $t=T$)* | |
> | Final Inheritable Parameters | ~15.3 M |
> | Storage Size to Ship (*e.g.*, .pt file) | ~64.2 MB |
>
> **(1) Pre-pruning Parameter Count.**
> As shown in the table above, the ``pre-pruning'' count refers to the total inheritable parameters at the beginning of the Aux-Net training ($t=0$). For ALT, this is the sum of the learngene and our SVD-parameterized matrices at their initial maximum component configuration ($H^{(0)}$). It is noteworthy that even before any dynamic adaptation, ALT's SVD-based parameterization is already significantly more compact than the dense matrices.
>
> **(2) Average Active Components over Training.**
> This metric represents the time-averaged number of active SVD components throughout the entire Aux-Net training process. We calculate this by averaging the value of $H(t)$.
>
> **(3) Runtime Storage Required to Ship a Reusable Initializer.**
> The final ''shippable'' initializer is a self-contained checkpoint file. Crucially, **no additional metadata is required** to reconstruct the per-matrix selections. Our default ``continuous selection'' strategy for Des-Net initialization is predefined (*i.e.*, it always selects the first $n$ components based on the target dimension), so this information does not need to be stored.

---

> ### Author Response · Authors · 2025-11-27
> **Response to Reviewer #MpXW(9/10)**
>
> **Q11: On Data/Teacher Dependency, Portability, Generality of ALT.**
>
> Thank you for these questions. ALT's Stage-1 training requires access to the pre-training dataset (or a close proxy) and the teacher model. We acknowledge this dependency in Sec.A.11. However, we would like to clarify the intended user and application scenario for ALT, in which **this dependency is not a limitation but rather a natural prerequisite that highlights our method's unique value.**
>
>
>
> **(1) Discussion on Data/Teacher Dependency: Reframing the Application Scenario.**
> For the developers and owners of large foundational models (*e.g.*, AI research labs, large enterprises). For these stakeholders, the central challenge is often: ``Given our massive, proprietary, and computationally expensive Ans-Net, how can we most efficiently leverage its knowledge to generate and deploy a diverse family of smaller, specialized models for various internal or external needs?'' In this context, **access to the pre-training data and the teacher model is a given premise, not a practical obstacle.** ALT operates under this premise to provide a novel solution that was previously unavailable.
>
> **(2) Discussion on Portability: ALT as a ``Knowledge Condensation and Decoupling'' Tool.**
> For these model owners, ALT offers a powerful method for **``knowledge condensation and decoupling.''** It enables them to perform a **one-time** knowledge extraction process, condensing the core capabilities of their massive Ans-Net into a single, highly-compact, and portable set of inheritable parameters. Once this compact parameter set is created, it can be easily stored and distributed. Any number of Des-Nets can then be initialized rapidly and efficiently *without ever needing to access the original multi-billion parameter Ans-Net again*. This process allows the model owner to share the compact initializer with downstream users or teams *without exposing their core intellectual property*: the full-scale model and its massive, sensitive pre-training dataset.
>
> In summary, while the data dependency does constrain who can create an ALT initializer, **it simultaneously enhances its value for those who can.** By transforming a large, inaccessible model into a small and powerful initializer, ALT acts as a critical enabler for the broad and safe dissemination of knowledge from foundational models.
>
> **(3) Discussion on the Generality of ALT.**
> Regarding the concern about the generality of ALT, we wish to emphasize that our core contribution is proposing a **novel, efficient, and model-agnostic** initialization framework. The applicability of ALT is not constrained by specific architectural components. While we lack access to the massive proprietary pretraining datasets required to train inheritable parameters for certain state-of-the-art architectures, **this represents a constraint on resources, not on the methodology.** Practitioners or organizations possessing such pretraining data can readily leverage our method to generate inheritable parameters for modern LLMs. Therefore, we posit that the generality of ALT is **not compromised** by the architectures tested in the paper, as the underlying mechanism remains effective across different model configurations.

---

> ### Author Response · Authors · 2025-11-27
> **Response to Reviewer #MpXW(10/10)**
>
> **Q12: On the Development Logic and Theoretical Outlook of ALT.**
>
> **(1) An Empirically-Driven Approach.**
> We agree with you that our work is primarily **empirically-driven**. Our methodology, observing an empirical phenomenon (the low-rank nature of transformation matrices), formulating a hypothesis, designing a solution (ALT), and validating it through extensive experiments, is **a common and effective pipeline for making progress in the complex, non-linear domain of Transformer models.** Many advances in this field have followed a similar path.
>
> **(2) On When GIS-Guided Adaptation Preserves Quality.**
> While we do not offer a formal proof, the design of the Gated Importance Score (GIS) provides a heuristic answer. Our core hypothesis is that initialization quality is best preserved by retaining components that are deemed important by multiple learning signals. Our key innovation is the multiplicative formulation of GIS itself. As demonstrated in our ablation studies (*e.g.*, ALT-GIS vs. ALT-Add), this multiplicative combination is empirically superior to simpler additive metrics. We posit that this is because the gating mechanism, where the ''energy'' (learnable value) term modulates the ``orientation'' (learnable vectors) term, provides a more effective and holistic measure of a component's true contribution. Therefore, by selectively retaining components with high GIS scores, our framework ensures that the most structurally and functionally significant parts of the transformation are preserved, leading to a high-quality initialization.
>
> **(3) On Theoretical Outlook of ALT.**
> We appreciate your suggestion regarding the theoretical guarantees of ALT. However, we respectfully submit that **establishing formal guarantees for a dynamic, data-dependent adaptation process within deep, non-linear Transformers remains a significant and open research challenge.** It is worth noting that even in more well-established field of model pruning, the predominant paradigm for state-of-the-art methods **relies on empirical heuristics rather than rigorous theoretical bounds**, such as [1],[2],[3], and *etc.*. Developing a solid theoretical guarantee of such method involves complex intersections of optimization theory and generalization bounds, which is widely regarded as an independent and active research domain in itself. While we agree that establishing a theoretical guarantee for GIS and our proposed strategies is a valuable direction, we respectfully note that a comprehensive theoretical proof falls outside the primary scope of this paper. We intend to investigate these promising theoretical directions in our future work.
>
> **References**
>
> [1] ``Width & Depth Pruning for Vision Transformers.", AAAI 2022.
>
> [2] ``Isomorphic Pruning for Vision Models.'', ECCV 2024.
>
> [3] ``Variance-Based Pruning for Accelerating and Compressing Trained Networks.'', ICCV 2025.

---

### Official Review · Reviewer_whCg · 2025-10-28

**Soundness:** 3
**Presentation:** 3
**Contribution:** 3
**Rating:** 6
**Confidence:** 4

**Summary:**

Focusing on the issue of inefficient parameter inheritance in the Learngene framework, this paper proposes the ALT method. Through analysis of LeTs, it is observed that the transformation matrices are over-parameterized. After confirming their low-rank characteristics via SVD, ALT parameterizes the transformation matrices in the form of SVD and introduces GIS to quantify component importance. Two adaptive strategies, FGA and HCA, are designed to optimize the matrices. The method is executed in two stages: training an Aux-Net to optimize parameters, and inheriting parameters to initialize Des-Nets. Experiments demonstrate that in both vision (20× compression compared to the 304M ViT-Large) and language (25× compression compared to the 1.5B GPT2-XLarge) domains, ALT achieves state-of-the-art results in parameter efficiency, downstream performance, and training efficiency.

**Strengths:**

1. Precise problem identification and solid theoretical foundation: It directly addresses the core bottleneck of "over-parameterized transformation matrices", systematically validating the low-rank characteristics of matrices through SVD, thereby providing a rigorous basis for low-rank parameterization and dynamic adaptation.
2. Practical technological innovation and strong adaptability: GIS accurately measures component importance by leveraging the multiplicative properties of SVD, while FGA and HCA are designed to accommodate "extreme compression" and "structural adaptation" scenarios respectively, meeting diverse initialization requirements for Des-Nets.

**Weaknesses:**

Limited knowledge transfer: The approach merely employs soft distillation to transfer knowledge from the Ans-Net's output layer, without utilizing fine-grained information such as intermediate features and attention mechanisms, thereby failing to fully exploit the potential of the Ans-Net.

**Questions:**

When adapting to 7B/65B LLaMA models, how is the scale of inherited parameters determined? Does the decoder structure require adjustments to ALT's SVD parameterization or adaptive strategies?

---

> ### Author Response · Authors · 2025-11-21
> **Response to Reviewer #whCg(1/2)**
>
> **Q1: On Incorporating Finer-Grained Knowledge Transfer.**
>
> Thank you for this constructive suggestion. We agree that exploring finer-grained distillation is a highly promising direction. To validate this, we conduct new experiments where we enhance our Aux-Net training by incorporating distillation from intermediate layers, specifically from attention maps and hidden states.
>
> **(1) Enhanced Distillation Objective.**
> We augment our original total training loss, $\mathcal{L}_{\text{total}}$, with two additional distillation terms defined as Eq.(8) in the revision.
>
> **(2) Self-Attention Distillation.**
> We compute the attention relation matrix $\boldsymbol{R}$ for both Aux-Net and Ans-Net as $\boldsymbol{R} = \operatorname{softmax}(\boldsymbol{Q}\boldsymbol{K}^\top / \sqrt{d})$, where $\boldsymbol{Q}, \boldsymbol{K}$ are the query and key matrices. The loss $\mathcal{L}_{\text{attn-distill}}$ is designed as $\left\| \boldsymbol{R}^S - \boldsymbol{R}^T \right\|^2_2$ to minimize the Mean Squared Error (MSE) between their corresponding relation matrices. During empirical experiments, we observe that directly aligning these relation matrices led to training instabilities. Consequently, we omit the attention distillation term in our final training process.
>
> **(3) Hidden-State Distillation.**
> Let $\boldsymbol{H}^S$ and $\boldsymbol{H}^T$ represent the hidden states of the Aux-Net and Ans-Net, respectively. The hidden-state distillation loss $\mathcal{L}_{\text{hidden-distill}}$ is computed using MSE as $\left\| \boldsymbol{H}^T - \phi(\boldsymbol{H}^S) \right\|^2_2.$ It is worth noting that since the feature dimensions of the Aux-Net and Ans-Net are inherently aligned, the learnable projection layer $\phi(\cdot)$ is not required (or can be regarded as an identity mapping). This objective forces the Aux-Net to regress the Ans-Net's semantic features directly, facilitating effective representation learning without introducing additional parameters.
>
> **Table:** Impact of fine-grained distillation. We train Aux-Nets for 100 epochs on ImageNet-1K and then evaluate the 5-epoch tuning performance of an initialized Des-H8-L12 model.
>
> | Aux-Net Training Method         | Top-1 Acc. (%) |
> | :------------------------------ | :------------: |
> | ALT (Soft Distillation)         |      76.0      |
> | ALT + Fine-grained Distillation |    **76.3**    |
>
> **(4) New Experimental Results.**
> To validate the effectiveness of fine-grained knowledge transfer, we conduct additional experiments comparing our original ALT (utilizing only soft distillation) against the enhanced version, denoted as ``ALT + Fine-grained Distillation''. Specifically, for the enhanced version, we trained the Aux-Net on ImageNet-1K for 100 epochs, incorporating the fine-grained distillation. As shown in the table above, we observe that the enhanced version yields a improvement in the performance of the downstream Des-H8-L12 model. This empirical evidence indicates that incorporating finer-grained intermediate knowledge leads to a higher-quality set of inheritable parameters.

---

> ### Author Response · Authors · 2025-11-27
> **Response to Reviewer #whCg(2/2)**
>
> **Q2: On Applying ALT to Decoder-based LLMs.**
>
> Thank you for this insightful question regarding the scalability of ALT to large, decoder-only language models like LLaMA. We are happy to confirm that our **ALT is directly applicable without requiring adjustments to its core mechanisms**. And we have added these discussion into Sec.A.11 of our original paper.
>
> **(1) Determining the Size of Inheritable Parameters.**
> The total number of inheritable parameters in ALT is the sum of the parameters from the **learngene module** and the **transformation matrices**. A key feature of our ALT framework is that the SVD-based parameterization and dynamic adaptation drastically reduce the transformation matrices parameters. Consequently, the overall scale of the inheritable parameter set is **dominantly determined by the user-defined size of the learngene module itself**. This is evident in our vision experiments: of about 15.3M total inheritable parameters, the learngene module accounts for about 15.0M ($\approx 98$\%), while all the highly-efficient transformation matrices contribute only about 0.3M. **This principle extends directly to LLM scenarios.** When applying ALT to models like LLaMA-7B or 65B, the user would first choose a desired size for the learngene module based on their efficiency requirements. For instance, if one configures a 130M-parameter learngene module (*e.g.*, a 12-layer one), the final total number of inheritable parameters would be only marginally larger (*e.g.*, in the 130--135M range), even for a LLaMA-65B Ans-Net. The size of the inheritable parameters is a flexible, user-controlled design choice.
>
> **(2) Applicability to Decoder Structures without Adjustments.**
> ALT's core mechanisms do not require any adjustments when applying to the decoder structure. Our ALT method is designed to be **architecture-agnostic** at a macro level because it operates on the linear matrices common to all Transformer models. The core components of a decoder-only model like LLaMA are still linear matrices within Attention or Feed-Forward Network blocks. Crucially, ALT's operations, such as SVD parameterization, GIS calculation, and adaptation strategies (HCA/FGA), are applied directly to the linear matrices within these blocks. Therefore, **ALT can be applied seamlessly to linear matrices of a decoder block in exactly the same manner as it is to an encoder block.** No modifications to our SVD parameterization or adaptive strategies are necessary.

---

### Official Review · Reviewer_jR65 · 2025-10-31

**Soundness:** 2
**Presentation:** 3
**Contribution:** 2
**Rating:** 4
**Confidence:** 2

**Summary:**

The paper presents a new approach to improve upon the learngene framework. The learngene framework focuses on learning initialization parameters for different architectures from a single pre-trained checkpoint. The paper observes that a key bottleneck of the learngene framework is the size of the transformation matrices and finds that they are low rank. The paper proposes to parameterize these matrices using dynamic low-rank ones and adjust the rank during the training process.

**Strengths:**

1. The paper presents an extensive range of experiments on different Transformer architectures and datasets using data from different domains.
2. The paper provides detailed analysis experiments showcasing the efficacy of their approach.

**Weaknesses:**

1. Since the paper builds on the learngene framework, it should have a section explaining the detailed workflow of learngene and LeTs. For readers unfamiliar with these frameworks, it is quite difficult to understand section 3.1 and several other parts of the paper.

2. It is unclear why the paper chose to go with the parameterization defined in Eq. 1. From the motivation, the paper aim to have a low-rank parameterization, which could be easily achieved using a product of low-rank matrices. This would remove the need to have regularization parameters to ensure the $U$ and $V$ are orthogonal. It would also be possible to have derive a dynamic adaptation version of this framework by writing the overall matrix as a sum of low-rank matrices (with different ranks). The dynamic adaptation could focus on selecting the constants in this setting.

3. Because of the weakness 1, it is unclear what loss function is being optimized while performing dynamic component adaptation in Section 3.3.

**Questions:**

Please respond to the questions in the above sections.

---

> ### Author Response · Authors · 2025-11-21
> **Response to Reviewer #jR65(1/3)**
>
> **Q1: On the Detailed Workflow of Learngene and LeTs.**
>
> Thank you for this feedback. We agree that a more detailed explanation of the Learngene framework and the LeTs method is important for readers unfamiliar with this background.
>
> **First, we would like to gently point out that we have provided an overview of these concepts in our original paper.** Specifically, Fig.1(a) and 1(b) illustrate the general paradigm, and the second paragraph of our Introduction as well as the Learngene paragraph in our Related Work section describe the basic concepts. To make our paper more self-contained, we provide a more detailed description below, and we have added these details to the Sec.A.15 of appendix in the revision.
>
>
>
> **(1) The Learngene Paradigm.**
> As illustrated in Fig.1(a), the Learngene framework operates on a two-stage process:
>
> **Stage 1 (Training the learngene):** It first learns a single compact parameter set, termed **learngene**, under the guidance of a large well-trained **Ancestry Model (Ans-Net)**.
>
> **Stage 2 (Initialization from learngene):** The learngene is then inherited and transformed to initialize a family of variable-sized **Descendant Models (Des-Nets)**, which are subsequently fine-tuned on downstream tasks.
>
> **(2) The LeTs Method.**
> LeTs is a state-of-the-art method of the Learngene paradigm that introduces **learnable transformation matrices** to perform learngene transformation. It leverages the ``Aux-Net Strategy" (illustrated in Fig.1(b)):
>
> **Auxiliary Model (Aux-Net):** LeTs constructs an Aux-Net, whose parameters are generated by transforming the compact learngene. This transformation is performed by a set of learnable matrices.
>
> **Transformation Dimensions:** The transformation in LeTs operates along two dimensions:
>
> 1) *Width Transformation:* It uses **dense** matrices, $\boldsymbol{F}^{\mathrm{in}}$ and $\boldsymbol{F}^{\mathrm{out}}$, to increase the width (*e.g.*, hidden dimension) of the learngene matrices. **This is the primary bottleneck that our ALT addresses.**
>
> 2) *Depth Transformation:* It uses another matrix, $\boldsymbol{G}$, to linearly combine the width-transformed learngene matrices to construct variable-depth models.
>
> **Stage 1 (Training the learngene):** The Aux-Net is trained under the guidance of the Ans-Net, typically via knowledge distillation. This process simultaneously optimizes both the learngene itself and the learnable transformation matrices ($\boldsymbol{F}$ and $\boldsymbol{G}$).
>
> **Stage 2 (Initialization from learngene):** Once the Aux-Net is trained, the well-trained transformation matrices are used to initialize Des-Nets of various sizes, employing selection strategies (*e.g.*, continuous selection) to meet specific architectural requirements.

---

> ### Author Response · Authors · 2025-11-21
> **Response to Reviewer #jR65(2/3)**
>
> **Q2: On the Choice of SVD-based Parameterization.**
>
> Thank you for this question regarding our choice of the SVD-based parameterization (Eq.(1)) rather than a simpler product of low-rank matrices (*e.g.*, $\boldsymbol{F} = \boldsymbol{B}\boldsymbol{A}$). We would like to clarify that **our methodological starting point was not a prior goal to have ``a low-rank parameterization'' but rather was the direct result of an empirical investigation into the properties of the LeTs method.** Our design process was empirically driven, proceeding from analysis to formulation.
>
> **(1) Analysis via SVD Reveals Key Insights.**
> **As detailed in the third paragraph of our Introduction,** our investigation began by utilizing Singular Value Decomposition (SVD) as an analytical tool to explore the internal structure of the well-trained (and dense) transformation matrices from LeTs. This analysis was motivated by the established principle that matrices with rapidly decaying singular values possess a strong low-rank structure. **Our analysis, presented in Fig.1(c), yielded two critical insights that directly motivated our entire approach:**
>
> **Insight 1 (Rapid Decay):** The singular values of these matrices decay rapidly, confirming that they are highly overparameterized and can be effectively approximated with a low-rank parameterization.
>
> **Insight 2 (Varying Patterns):** The decay patterns are not the same, where they vary significantly across different matrices (both by layer and module type).
>
> **(2) From Insights to Methodological Design.**
> These two empirical observations naturally and logically led to our core design choices:
>
> **SVD-based Parameterization (Eq.(1)):**
> Based on *Insight 1*, choosing an SVD-based parameterization ($\boldsymbol{F} = \boldsymbol{U\Sigma V}$) was not merely one option among many, but **the most natural choice**. It directly models the low-rank structure that our analysis revealed. **As we state in original paper (Line 97-98)**, ``Specifically, inspired by the rapid singular-value decay of these transformation matrices, we explicitly parameterize in the form of SVD, ...''.
>
> **Dynamic Component Adaptation:**
> In *Insight 2*, the varying decay patterns indicated an opportunity for a granular and adaptive approach. It is because of our SVD-based parameterization, which decouples a matrix into distinct components, that we are able to exploit these diverse patterns by dynamically adapting the number of active components for each matrix. **As we state in original paper (Line 99-101)**, ''..., which naturally enables us to define the triplet of a learnable value and its associated vectors as a cohesive learnable component, forming the basis for our subsequent adaptation. Next, exploiting the diverse decay patterns, we dynamically adapt the number of active components within each parameterized matrix.''
>
> In summary, **our choice of the parameterization in Eq.(1) is not an arbitrary selection but a direct and logical consequence of our initial empirical analysis.** And it also serves as the base for our entire dynamic adaptation process.
>
> **(3) Corroborating Evidence from Prior Works in Other Domains.**
> Our choice is further supported by empirical evidence from the Parameter-Efficient Fine-Tuning (PEFT) domain. For instance, AdaLoRA[1], which employs a SVD-form incremental updates, consistently demonstrated superior performance over the original LoRA[2], which uses the simpler $\boldsymbol{B}\boldsymbol{A}$ decomposition. While the domains differ, this provides strong corroborating evidence that the structural properties of an SVD-based formulation can offer significant advantages over low-rank product form.
>
> **Table:** Comparison between ALT (SVD-based parameterization) and ALT-LoRA (LoRA-based parameterization). We train Aux-Nets for 100 epochs on ImageNet-1K and then evaluate the 5-epoch tuning performance of an initialized Des-H8-L12 model.
>
> | Method   | Inheritable parameters (M) | Top-1 Acc. (%) |
> | :------- | :------------------------: | :------------: |
> | ALT-LoRA |          16.0            |      72.2      |
> | ALT      |            15.3            |    **76.0**    |
>
>
> **(4) New Study.**
> We establish a strong baseline, termed ``ALT-LoRA'', which utilizes LoRA-based parameterization ($\boldsymbol{F}=\boldsymbol{B}\boldsymbol{A}$) to replace the SVD-based one. As shown in the table above, our ALT outperforms the low-rank product formulation ALT-LoRA. This provides compelling empirical evidence that the design advantages of the SVD-based parameterization, **particularly its decoupled nature that facilitates our GIS metric and dynamic adaptation process,** are critical for achieving superior performance. The marginally higher inheritable parameters in ALT-LoRA stems from the larger number of retained learnable components.
>
> **References**
>
> [1] ``AdaLoRA: Adaptive Budget Allocation for Parameter-Efficient Fine-Tuning.'', ICLR 2023.
>
> [2] ``LoRA: Low-rank adaptation of large language models.'', ICLR 2022.

---

> ### Author Response · Authors · 2025-11-21
> **Response to Reviewer #jR65(3/3)**
>
> **Q3: On the Loss Function for Dynamic Component Adaptation.**
>
> Thank you for the question regarding the loss function that guides our dynamic component adaptation process. To clarify, you are correct that Sec.3.3 describes the dynamic adaptation process. **The loss function that drives this entire process is the same joint training objective used to train our Aux-Net.** This is because the dynamic adaptation is not a separate optimization process, but an integrated mechanism that operates during the training of the Aux-Net.
>
> We provided the description and definition of this loss function in the original paper in subsequent sections. We would like to guide you to the specific locations: A general description of this joint objective is provided in **Sec.3.4, ``Overall Pipeline of ALT'' (Line 300-304)**, where we state: **``The Aux-Net is then trained with a joint objective that minimizes both the task loss, the output discrepancy between Ans-Net and Aux-Net, and one orthogonality regularization loss.''**. The formulation of this total training loss, $\mathcal{L}_{\text{total}}$, is detailed in **Sec.A.22.1, ``Training Details for Aux-Net''**, and is given by **Eq.(9)**.
>
> To summarize, the ``Dynamic Component Adaptation'' mechanism works by reading the gradients computed from $\mathcal{L}_{\text{total}}$, calculating the Gated Importance Scores (GIS), and then refining the transformation matrix accordingly. To improve the reading flow of our paper, we have added a forward reference in Sec.3.3 of the revision, explicitly pointing the reader to the sections where the loss function is detailed.

---

### Official Review · Reviewer_7Pxt · 2025-11-01

**Soundness:** 3
**Presentation:** 2
**Contribution:** 2
**Rating:** 6
**Confidence:** 3

**Summary:**

This paper addresses a critical limitation in the Learngene framework: the low efficiency of inheritable parameters. Existing methods (e.g., LeTs) suffer from overparameterized transformation matrices, achieving only 3–6× parameter compression and poor portability. To tackle this, the authors propose ALT (Adaptive Low-Rank Transformation), a novel Learngene approach that leverages Singular Value Decomposition (SVD) for parameterization and dynamic component adaptation. Key innovations include: 1) a SVD-inspired Gated Importance Score (GIS) that integrates component magnitude and orientation to quantify importance; 2) two adaptation strategies—Flat Global Adaptation (FGA) and Hierarchical Component Adaptation (HCA)—to dynamically optimize transformation matrix structures. Comprehensive experiments across vision (with ViT-Large as the Ancestry model, achieving 20× compression) and language (with GPT2-XLarge as the Ancestry model, achieving 25× compression) domains demonstrate that ALT outperforms baselines (Scratch, LeTs, Pre-Fin) on downstream tasks (e.g., ImageNet-1K classification, ADE20K segmentation, GLUE benchmarks) while reducing training steps by approximately 2×. This work confirms that ALT significantly enhances inheritable-parameter efficiency without compromising initialization quality.

**Strengths:**

1. Precise Problem Identification: The authors accurately pinpoint the core bottleneck of existing Learngene methods—overparameterized transformation matrices (accounting for >70% of inherited parameters in LeTs). By analyzing the low-rank properties of these matrices via SVD (rapid singular value decay), they provide a solid theoretical foundation for ALT’s design, avoiding the pitfall of "searching for a solution without a clear problem."

2. Novel and Rigorous Methodology: The proposed GIS metric effectively captures the multiplicative interaction between singular values (energy and structural direction), offering greater accuracy than additive scores (e.g., those in AdaLoRA). The two adaptation strategies (FGA, HCA) cater to distinct needs: FGA ensures global elitism in component selection, while HCA adapts to the modular structure of Transformers through a two-stage decision process (inter-matrix allocation + intra-matrix selection), demonstrating strong algorithmic depth.

3. Comprehensive and Rigorous Experiments: The work validates ALT across diverse tasks (classification, segmentation, detection in vision; pre-training, QA, reasoning in language) and model scales (from Des-H6-L10 to Des-H16-L24). Ablation studies systematically verify the necessity of dynamic adaptation, GIS, and HCA, while comparisons with competitive baselines (LeTs, IMwLM, MatFormer, knowledge distillation) highlight ALT’s dual advantages in efficiency and performance.
High Practical Value: ALT achieves 20–25× parameter compression, drastically reducing storage and computational costs for model initialization. Its "train once, initialize many times" paradigm is particularly valuable for resource-constrained scenarios (e.g., edge devices, mobile platforms) and rapid prototyping of variable-sized models, addressing real-world pain points of existing Learngene methods.

**Weaknesses:**

1. Lack of Sufficient Justification for the Priori "Minimum Effective Skeleton" of Learngene, Leading to Strong Subjectivity: The initial structure of Learngene (the "minimum effective skeleton") in the paper is set via fixed priors: "6 heads (head dimension 64) + 8 layers" for vision tasks, and "6 heads + 8 layers (BERT)/12 layers (GPT-2)" for language tasks. The authors only describe this as an "empirically derived minimum effective scale" but provide no clear rationale for the choices: e.g., why 6 heads instead of 5 or 7? Why 8 layers as the minimum effective number for vision tasks instead of 7 or 9? This strong priori setting poses two issues: first, there are no ablation experiments to verify how different initial skeletons (e.g., varying head counts or layer numbers) affect Learngene’s knowledge inheritance ability and subsequent parameter compression efficiency, making it impossible to confirm if the current setting is optimal or suboptimal. Second, it remains unclear whether this priori skeleton is applicable when ALT is transferred to other architectures (e.g., Swin Transformers, MoE models). Such subjectivity may limit ALT’s generalizability and reduce the method’s interpretability and reproducibility.

2. Insufficient Comparison with State-of-the-Art Low-Rank and Compression Methods: Although the paper distinguishes ALT from Parameter-Efficient Fine-Tuning (PEFT) methods (e.g., LoRA, AdaLoRA) by their objectives (initialization vs. task adaptation), it lacks direct empirical comparisons with these methods within the Learngene framework. For example, it does not verify how ALT performs against "LoRA-based transformation matrices" in terms of parameter efficiency and downstream accuracy. Additionally, ALT is not compared with other model compression techniques (e.g., structured pruning) for inheritable parameter learning, weakening the argument for ALT’s superiority in efficiency.

3. Lack of Hyperparameter Sensitivity Analysis: Key hyperparameters of ALT (e.g., initial/final component counts H (0) /H (T), orthogonality regularization weight β,) are set to fixed values (e.g., H (0) =544, β=1e−3) without testing how variations in these hyperparameters affect model performance. For example, how does adjusting H (T) balance parameter compression ratio and downstream performance? Such analysis is critical for guiding the practical deployment of ALT, and its absence undermines the method’s usability.

4. Incomplete Discussion of Des-Net Initialization Strategies: The paper adopts LeTs’ initialization strategies but does not compare them with alternative strategies (e.g., random selection). It also fails to explain why continuous selection is optimal.
Insufficient Analysis of Performance Differences Across Des-Net Scales: Experimental results show that ALT’s performance gains vary across Des-Net sizes (e.g., 0.7% improvement over LeTs for Des-H6-L10, 0.9% for Des-H12-L16). However, the authors do not analyze the reasons for this variation.

5. Insufficient Analysis of Performance Differences Across Des-Net Scales: Experimental results show that ALT’s performance gains vary across Des-Net sizes (e.g., 0.7% improvement over LeTs for Des-H6-L10, 0.9% for Des-H12-L16). However, the authors do not analyze the reasons for this variation.

**Questions:**

Please refer to the weaknesses section

---

> ### Author Response · Authors · 2025-11-21
> **Response to Reviewer #7Pxt (1/4)**
>
> **Q1: On the Rationale and Ablation for the Initial Learngene Structure.**
>
> We appreciate your question about the choice of initial learngene structure. Our rationale is twofold: **ensuring a fair comparison** and **building upon established empirical findings**. We have also conducted new ablation studies.
>
> **(1) Ensuring a Fair Comparison.**
> Our primary objective is to demonstrate the superiority of our proposed ALT over existing state-of-the-art methods like LeTs within the Learngene paradigm. Therefore, adhering to the principle of controlled experimental setting, we adopt the identical ``minimum effective skeleton'' (H6-L8) used in LeTs. This ensures that **the observed performance gains are attributable solely to our novel contributions** (GIS, FGA, HCA, *etc.*), rather than being confounded by a potentially better, but different architectural starting point.
>
> **Table:** Ablation study on varying the initial learngene structure. We train Aux-Nets comprising different learngene structures for 100 epochs on ImageNet-1K and then evaluate the 5-epoch tuning performance of an initialized Des-H8-L12 model.
>
> | Learngene Structure     | Inheritable Parameters(M) | Top-1 Acc. (%) |
> | :---------------------- | :-----------------------: | :------------: |
> | H4-L6 (Smaller)         |            6.0            |      69.3      |
> | H6-L8 (Current setting) |           15.3            |      76.0      |
> | H8-L8 (Larger)          |           26.9            |      77.8      |
>
> **(2) Building on an Established Baseline (LeTs).**
> The H6-L8 configuration is not arbitrary but is an empirically validated starting point that has emerged from the Learngene paradigm. It represents a widely-accepted trade-off between being compact enough to demonstrate high parameter efficiency and large enough to inherit the knowledge from the Ans-Net. To empirically address your question, we conduct a ablation study on ImageNet-1K by varying the initial learngene structure, evaluate and report the 5-epoch tuning performance of an initialized Des-H8-L12 model.
>
> As shown in the table above, the smaller H4-L6 one, while more efficient in inheritable parameters, fails to capture sufficient knowledge, leading to a significant drop in the Des-Net performance. Conversely, the larger H8-L10 one offers a performance improvement but at the cost of a substantial increase in inheritable parameters, which contradicts **our core objective of maximizing inheritable-parameter efficiency**. These results demonstrate that our chosen H6-L8 configuration represents a compelling sweet spot, achieving a balance between Des-Net performance and inheritable parameter efficiency. **It is noteworthy that the learngene structure exhibits inherent flexibility and can be adapted by different users.**
>
> **(3) On the Generalizability of ALT to Different Architectures.**
> We wish to clarify that our ALT is designed to be **model-agnostic**, and its core mechanisms are not limited to the standard Transformer architecture. The fundamental components of ALT, the Gated Importance Score (GIS), and the adaptation strategies (FGA/HCA)—operate on the linear matrices (*e.g.*, for query/key/value projections and MLP layers) that are common across the Transformer family. Since architectures like the Swin Transformer are also built upon these linear matrices, our ALT is directly applicable.

---

> ### Author Response · Authors · 2025-11-21
> **Response to Reviewer #7Pxt (2/4)**
>
> **Q2: On the Comparison with Low-Rank and Compression Methods.**
>
> Thank you for recognizing the conceptual differences between our ALT and standard PEFT/pruning methods. Here, we empirically compare ALT against strong methods from PEFT and pruning.
>
> **(1) Comparison with PEFT.**
> Conceptually, ALT and most PEFT methods (*e.g.*, LoRA[1]) **operate at fundamentally different stages of the model lifecycle and address distinct problems**: PEFT methods are designed to efficiently fine-tune a pre-existing large pretrained model. They achieve parameter efficiency by only updating a small subset of parameters while keeping the majority of the original pretrained weights frozen. This inherently **requires loading and maintaining the entire pretrained model** during the fine-tuning process. This dependency limits their flexibility in scenarios where the target device or environment cannot accommodate the full large model. For example, if a pretrained model has 100M parameters, but the target device can only support deploying and tuning models of up to 50M parameters, PEFT methods cannot be directly applied, as they assume access to the full 100M model. In contrast, ALT focuses on **model initialization** under diverse resource constraints. ALT learns and inherits a highly compact parameter set (significantly $\ll$ 100M parameters). This compact set enables us to directly initialize a suitable model of any desired size (*e.g.*, a 40M-parameter instance) on a target device, without ever needing to load a larger pretrained model. This is critical for scenarios with strict memory or computational limits, where deploying a large pretrained model for PEFT is simply infeasible.
>
> **Table:** Comparison between LoRA[1], GPS[2], SNF[3] and ALT on CIFAR100. We refer the results reported in Tab.4 of [4].
>
> |    Model    | LoRA | GPS  | SNF  |   ALT    |
> | :---------: | :--: | :--: | :--: | :------: |
> | Des-H12-L12 | 67.1 | 81.1 | 84.0 | **90.4** |
>
> Experimentally, we compare ALT against LoRA[1], GPS[2] and SNF[3] on CIFAR100 with Des-H12-L12 model (86M) and refer the results reported in Tab.4 of [4]. As shown in the table above, ALT achieves better performance. Notably, LoRA requires the entire 86M pretrained parameters to initialize the downstream model. In contrast, ALT only requires its highly compact parameter set (15.3M) to initialize.
>
> Crucially, **ALT and PEFT methods are orthogonal and can be used in sequence**. After initializing a model via ALT, one can still apply any PEFT method on the ALT-initialized Des-Net.
>
> **(2) Comparison with Pruning.**
> Conceptually, pruning aims to obtain a single compact model as an endpoint. In contrast, ALT, embodying a ``learn once, initialize many'' paradigm, learns a highly-compact, reusable parameter set for initializing variable-sized models. Furthermore, pruning often results in irregular model architectures (*e.g.*, varying widths per layer), which may complicate deployment, whereas ALT always initializes standard, regular Transformer architectures.
>
> **Table:** Comparison between WDPruning[5], VBP[6] and ALT. ''wo KD'' and ''w/ KD'' mean without and with knowledge distillation respectively.
>
> |  Method   | Params(M) |         Main Process         | Top-1 Acc.(%) |
> | :-------: | :-------: | :--------------------------: | :-----------: |
> | WDPruning |   55.3    |  100 epoch pruning (wo KD)   |     80.8      |
> |    VBP    |   55.4    | 10 epoch fine-tuning (w/ KD) |     80.7      |
> |    ALT    | **51.5**  | 5 epoch fine-tuning (wo KD)  |   **80.9**    |
>
> Experimentally, we compare ALT against WDPruning[5] and VBP[6] (results are referred in their original paper): pruning ImageNet-1K pretrained Des-Nets to a target size. As shown in the table above, ALT achieves **better performance and obtains smaller model**. Both WDPruning and VBP require the entire pretrained parameters to initialize the pre-pruning model and then prune the model to target ones. Besides, VBP needs 10-epoch fine-tuning with knowledge distillation. In contrast, ALT only requires its reusable and compact parameter set to initialize target model and simply fine-tunes it for 5 epochs.
>
> **References**
>
> [1] ``Lora: Low-rank adaptation of large language models.'', ICLR 2021.
>
> [2] ``Gradient-based parameter selection for efficient fine-tuning.'', CVPR 2024.
>
> [3] ``Adapting shortcut with normalizing flow: An efficient tuning framework for visual recognition.'', CVPR 2023.
>
> [4] ``Parameter-efficient fine-tuning for pre-trained vision models: A survey.'', arXiv 2024.
>
> [5] ``Width & Depth Pruning for Vision Transformers.", AAAI 2022.
>
> [6] ``Variance-Based Pruning for Accelerating and Compressing Trained Networks.'', ICCV 2025.

---

> ### Author Response · Authors · 2025-11-21
> **Response to Reviewer #7Pxt (3/4)**
>
> **Q3: On the Hyperparameter Sensitivity Analysis.**
>
> To address the additional sensitivity analysis, we have conducted two new sets of ablation studies focusing on the final number of active components $H^{(T)}$ and the orthogonality regularization weight $\beta$. Specifically, we train Aux-Nets for 100 epochs on ImageNet-1K with different $H^{(t)}$ and $\beta$ to obtain different inheritable parameter sets, and then evaluate the 5-epoch tuning performance of an initialized Des-H8-L12 model.
>
> **Table:** Ablation studies focusing on the final number of active components $H^{(T)}$ and the orthogonality regularization weight $\beta$. We train Aux-Nets for 100 epochs on ImageNet-1K and then evaluate the 5-epoch tuning performance of an initialized Des-H8-L12 model.
>
> | $H^{(T)}$ | Top-1 Acc. (%) | $\beta$ | Top-1 Acc. (%) |
> | :-------: | :------------: | :-----: | :------------: |
> |    68     |      75.9      |    0    |      75.6      |
> |    136    |    **76.0**    |  1e-3   |    **76.0**    |
> |    272    |      75.9      |  1e-2   |      75.8      |
>
> **(1) Sensitivity to $H^{(T)}$.**
> The hyperparameter $H^{(T)}$ controls the number of parameters within the SVD-based transformation matrices. From the table above, we observe that different $H^{(T)}$ yield similar performance. This can be attributed to the fact that **these matrices constitute a minimal fraction of the total inheritable parameters (approximately 2%)**, whereas the learngene parameters dominate the vast majority of the inheritable parameters. However, it is crucial to clarify that **this low parameter count does not imply a lack of importance. On the contrary, it highlights the high parameter efficiency of our design.** Despite the compact size, these matrices play a pivotal role in the transformation of learngene parameters to construct the Aux-Net and in the effective initialization of the Des-Net. They function as a lightweight yet critical bridge, enabling effective knowledge transfer.
>
> **(2) Sensitivity to $\beta$.**
> From the table above, we observe that removing the orthogonality constraint ($\beta=0$) degrades performance, confirming its necessity for SVD-based parameterization. Moreover, the model's performance remains robust across a range of reasonable values for $\beta$ (1e-3 and 1e-2), indicating that our method is not overly sensitive to this hyperparameter. We attribute this robustness to the fact that the orthogonality regularization is applied to the transformation matrices. Since these matrices constitute only a small fraction of the total inheritable parameters, variations in the regularization strength do not significantly perturb the overall learning process of the inheritable parameters.
>
> **Q4: On the Discussion of Des-Net Initialization Strategies.**
>
> Thank you for the insightful feedback regarding the initialization strategies for Des-Nets. First, we would like to clarify that the core novelty of our work lies in learning a highly-efficient inheritable parameter set (via GIS and our adaptation strategies), rather than in the subsequent step of using this set to initialize Des-Nets. For the latter, we adopt the established and effective strategies from LeTs to **ensure a focused and fair comparison of the inheritable parameters themselves**. Here, we elaborate on the rationale and provide further empirical evidence. The choice of continuous selection over random selection is motivated by the desire to preserve the learned connectivity information encoded in the transformation matrices. To empirically validate this, we compare our default continuous selection against random selection for initializing a Des-H8-L12 model.
>
> **Table:** Comparison between different initialization strategies. We train Aux-Nets for 300 epochs on ImageNet-1K and then evaluate the 5-epoch tuning performance of an initialized Des-H8-L12 model.
>
> | Initialization Strategy | Top-1 Acc. (%) |
> | :---------------------: | :------------: |
> |    Random Selection     |      78.7      |
> |  Continuous Selection   |    **79.1**    |
> |   L2-norm $\uparrow$    |      79.1      |
> |  L2-norm $\downarrow$   |      78.8      |
>
> The table above clearly demonstrates that continuous selection outperforms random selection, which confirms that **preserving the learned connectivity** is crucial for a high-quality initialization. Furthermore, as you correctly pointed out the need for comparison, **we would like to highlight that in Tab.3 of our original paper, we already compared several alternative selection strategies.** Specifically, we evaluated magnitude-based selection methods, which are also based on an importance principle. The results showed that these strategies achieve similar, strong performance, and all are superior to a random selection.

---

> ### Author Response · Authors · 2025-11-21
> **Response to Reviewer #7Pxt (4/4)**
>
> **Q5: On the Analysis of Performance Gains Across Des-Net Sizes.**
>
> Thank you for this insightful observation regarding the variance in ALT's performance gains across different Des-Net sizes. Our analysis suggests that this phenomenon is not an anomaly specific to ALT, but rather a general characteristic in the context of variable-sized model initialization. We support this claim with two lines of evidence.
>
> **(1) First, this variance is consistent within the Learngene framework itself.**
> As can be observed from results (*e.g.*, in Tab.4 of our paper), the performance gain of the baseline method, LeTs, over training from scratch also varies across different Des-Net sizes. For instance, the improvement is 2.4% for Des-H6-L10 and 0.6% for Des-H8-L10. This indicates that the effectiveness of Learngene-based initialization is intrinsically linked to the size and capacity of the target Des-Net architecture.
>
> **(2) Second, this trend is broadly observed across different initialization paradigms beyond Learngene.**
> For example, a similar pattern is evident in Matformer[1] (see Fig.4(a) of their original paper), where the performance improvement of Matformer over its baseline also varies differently with the size of the sub-model being supported. This suggests that the interplay between the quality of an initialization and the final performance of the target model is a complex, non-linear function of the model's intrinsic properties (*e.g.*, number of parameters, depth, width).
>
> To summarize, we hypothesize that the magnitude of the performance gain reflects how sensitive a model of a particular size is to a high-quality initialization. For certain model sizes (*i.e.*, ``sweet spots''), a superior starting point can dramatically alter the optimization trajectory, leading to a more substantial performance improvement. For other sizes, the impact of initialization might be partially affected by other factors, such as the model's inherent capacity limits or the complexity of the optimization landscape. **Therefore, observing a variation in performance gains is an expected outcome that reflects this complex interaction.**
>
> **References**
>
> [1] ``MatFormer: Nested Transformer for Elastic Inference.'', NeurIPS 2024.

---

### Meta-Review · Area_Chair_gW58 · 2026-01-07

**Summary:**

This paper observes the key bottleneck of the learngene framework, i.e., the size of the transformation matrices and finds that they are low rank. To address this issue, this paper proposes to parameterize these matrices using dynamic low-rank ones and adjust the rank during the training process. Four reviewers pointed out several critical concerns regarding different aspects of this paper, including but not limited to:

1. The novelty is relatively limited. The core idea, i.e., SVD‑parameterized low‑rank components with a learned importance score and progressive pruning, is close to methods like AdaLoRA/PiSSA/MiLoRA.
2. Lack of sufficient justification for the priori "Minimum Effective Skeleton" of learngene leads to strong subjectivity. There are no ablation experiments to verify how different initial skeletons affect Learngene’s knowledge inheritance ability and subsequent parameter compression efficiency. Second, it remains unclear whether this priori skeleton is applicable when ALT is transferred to other architectures. Such subjectivity may limit ALT’s generalizability and reduce the method’s interpretability and reproducibility.
3. The comparison with state-of-the-art low-rank and compression methods is insufficient. It lacks direct empirical comparisons with these methods within the Learngene framework. ALT is not compared with other model compression techniques (e.g., structured pruning) for inheritable parameter learning.
4. The hyperparameter sensitivity analysis is missing, such as the initial/final component counts H (0) /H (T), orthogonality regularization weight.
5. The discussion of Des-Net initialization strategies is incomplete. Why continuous selection is optimal is not well explained.
6. The analysis of performance differences across des-net scales is insufficient.
7. The approach merely employs soft distillation to transfer knowledge from the Ans-Net's output layer, without utilizing fine-grained information such as intermediate features and attention mechanisms, thereby failing to fully exploit the potential of the Ans-Net.
8. Data/teacher dependency undermines portability claims. The application to real‑world adoption in privacy‑ or license‑constrained settings could be significantly limited.
9. Compute story is mixed. ALT may be costlier overall. A formal break‑even analysis (as a function of #students, sizes, and tuning epochs) is missing.
10. There is no discussion of variance across seeds, confidence intervals, or significance tests.
11. The presentation needs to be improved. GIS definition lacks clarity and could be brittle.

The rebuttal addressed some concerns, such as ablations on hyperparameter sensitivity. But the current version needs a thorough revision based on the reviewers' comments and cannot be accepted.

**Reviewer Concerns:**

The 2nd, 4th, 5th, 6th, 7th concerns are partially addressed by the rebuttal, while the others are still outstanding.

**Reviewer Scores:**

None.

---

### Decision · Program_Chairs · 2026-01-26

Reject